# Nonconvex Decentralized Stochastic Bilevel Optimization under heavy-tailed noise

## Abstract

Existing decentralized stochastic optimization methods assume the lower-level loss function is strongly convex and the stochastic gradient noise has finite variance. These strong assumptions typically are not satisfied in real-world machine learning models. To address these limitations, we develop a novel decentralized stochastic bilevel optimization algorithm for the nonconvex bilevel optimization problem under heavy-tailed noise. Specifically, we develop a normalized stochastic variance-reduced bilevel gradient descent algorithm, which does not rely on any clipping operation. Moreover, we establish its convergence rate by innovatively bounding interdependent gradient sequences under heavy-tailed noise for nonconvex decentralized bilevel optimization problems. As far as we know, this is the first decentralized bilevel optimization algorithm with rigorous theoretical guarantees under heavy-tailed noise. The extensive experimental results confirm the effectiveness of our algorithm in handling heavy-tailed noise.

## 1 Introduction

Stochastic bilevel optimization consists of two levels of optimization subproblems, where the upper-level subproblem depends on the optimal solution of the lower-level subproblem. It has received a surge of attention in recent years because it lays the optimization foundation for a series of machine learning models, such as model-agnostic meta-learning (Finn et al., 2017), hyperparameter optimization (Franceschi et al., 2018; Pedregosa, 2016), imbalanced data classification (Yang, 2022), reinforcement learning (Shen et al., 2024; Li et al., 2024a), large language models (Shen et al., 2025; Li et al., 2024b), etc. To facilitate stochastic bilevel optimization for distributed machine learning models, where data are distributed across different workers, a series of decentralized stochastic bilevel optimization algorithms have been developed in recent years. Specifically, in a decentralized setting, each device computes stochastic gradients based on its local training data to update the variables of both the upper-level and lower-level subproblems, and then communicates these updates with neighboring workers in a peer-to-peer manner.

Compared to traditional single-level optimization problems, a unique challenge in decentralized stochastic bilevel optimization lies in computing the stochastic hypergradient, that is, the stochastic gradient of the upper-level loss function with respect to its variable. This challenge is caused by the unique characteristic of bilevel optimization: the upper-level subproblem relies on the optimal solution of the lower-level subproblem, which requires the global Hessian inverse matrix. To address this challenge, three categories of decentralized stochastic bilevel optimization algorithms (Yang et al., 2022b; Gao et al., 2023; Chen et al., 2022a;b; Zhang et al., 2023; Kong et al., 2024; Zhu et al., 2024; Lu et al., 2022; Liu et al., 2022b; 2023a; Wang et al., 2024; Qin et al., 2025) have been developed. The first category, such as Yang et al. (2022b), uses the Neumann series expansion approach to approximate the Hessian inverse on each device and then communicates it between workers, suffering from high communication costs. The second category, such as Zhang et al. (2023); Zhu et al. (2024), estimates the Hessian-inverse-vector product by solving an auxiliary quadratic optimization problem with gradient descent on each device and then communicating this estimator, which helps reduce communication costs. However, both the first and second categories incur significant computational overhead due to the need to compute second-order Hessian information. The third category, such as Wang et al. (2024), addresses this challenge by reformulating the decentralized stochastic bilevel problem as a single-level optimization problem and then solving it with only first-order gra-

dients. By avoiding the computation of second-order gradients, this category significantly reduces computational overhead.

However, existing decentralized stochastic bilevel optimization algorithms suffer from significant limitations. First, these algorithms require the loss function of the lower-level subproblem to be strongly convex. This strong assumption is not satisfied by most practical machine learning models, such as deep neural networks, which are inherently nonconvex. Second, they assume the stochastic noise in the gradient has finite variance. However, existing studies (Şimşekli et al., 2019; Zhang et al., 2020) have demonstrated that this bounded variance assumption does not hold for the commonly used deep neural networks. In practice, the stochastic noise often follows a heavy-tailed distribution. Hence, these practical scenarios make existing algorithmic designs and theoretical foundations for decentralized bilevel optimization ineffective. It is therefore necessary to develop new decentralized stochastic bilevel optimization algorithms that can accommodate a broader range of machine learning models and provide solid theoretical guarantees. To this end, the goal of this paper is to develop an efficient decentralized stochastic bilevel optimization algorithm for *nonconvex* bilevel problems under *heavy-tailed noise*, with rigorous theoretical guarantees. Since the first-order methods in the aforementioned third category offer low computational overhead and communication costs, this paper focuses on the first-order method.

For standard single-level optimization problems in the single-machine setting, a commonly used approach to handling heavy-tailed noise is Clipped SGD (Zhang et al., 2020), which mitigates the effect of heavy-tailed noise by clipping the norm of the stochastic gradient below a predefined threshold. Nevertheless, tuning the clipping threshold can be challenging. Recently, several works (Liu & Zhou, 2025; Hübler et al., 2024; Sun et al., 2024) have shown that the gradient normalization technique is sufficient to guarantee the convergence of stochastic gradient descent in-expectation under heavy-tailed noise without assuming bounded gradients. For instance, Hübler et al. (2024) proves that the batched normalized SGD (batched-NSGD) can converge in-expectation for a smooth nonconvex minimization problem under heavy-tailed noise, while Sun et al. (2024) achieves a similar conclusion for NSGD using a stronger assumption, the individual Lipschitz smoothness. Additionally, Liu & Zhou (2025) established the in-expectation convergence rate of the batched normalized stochastic gradient descent with momentum (batched-NSGDM) algorithm under heavy-tailed noise by innovatively bounding the accumulated noise from an online learning perspective.

Since the aforementioned approaches focus solely on single-level optimization in a single-machine setting, they are not applicable to decentralized stochastic bilevel optimization problems. In practice, this setting presents several unique challenges, outlined as follows.

1. In bilevel optimization, **multiple gradients interact with one another**. Each of these gradients is affected by the heavy-tailed noise, which in turn impacts convergence. Therefore, it is challenging to control all of them and establish a convergence rate under heavy-tailed noise.

2. In the decentralized setting, **the consensus error with respect to gradients is also affected by heavy-tailed noise**. It remains unclear how to design algorithms and analyses that effectively control this noise to ensure convergence.

3. The aforementioned first-order method for bilevel optimization requires advanced gradient estimators, such as the variance-reduced gradient, to avoid the quite slow convergence rate under the finite variance assumption, as shown in Kwon et al. (2023a). However, **no existing work for both single-level and bilevel problems has demonstrated that the advanced gradient estimator can ensure convergence under heavy-tailed noise without assuming bounded gradients**.

In summary, it is challenging to achieve a fast convergence rate for the first-order gradient-based decentralized bilevel optimization algorithm under heavy-tailed noise. To address these unique challenges, we develop a novel decentralized normalized stochastic gradient with variance reduction algorithm to solve Eq. (1). This algorithm **only requires normalized first order gradients**, making it more efficient and effective in handling heavy-tailed noise, which is lacking in existing second-order-based methods. Importantly, our algorithm demonstrates when gradient normalization should be applied in decentralized bilevel optimization. **To the best of our knowledge, this is the first algorithm capable of handling heavy-tailed noise in bilevel optimization**. We further establish the convergence rate of the developed algorithm under heavy-tailed noise. Specifically, to address challenges arising from the interaction between gradients of different variables, we explicitly char-

acterize their interdependence by innovatively handling the optimization subproblems associated with each variable. In addition, we provide a novel analysis of the consensus errors related to these gradients, which are also influenced by heavy-tailed noise. **To the best of our knowledge, this is the first work to bound interdependent gradient sequences under heavy-tailed noise in bilevel optimization**. Finally, the established convergence rate clearly illustrates how the properties of a decentralized system influence overall convergence, and extensive experimental results validate the effectiveness of the proposed algorithm in handling heavy-tailed noise.

## 2 RELATED WORK

### 2.1 DECENTRALIZED STOCHASTIC BILEVEL OPTIMIZATION

Decentralized stochastic bilevel optimization enables the decentralized optimization framework for bilevel optimization problems. Due to the two-level characteristics of this problem, there are some unique challenges for computation and communication compared to the decentralization of traditional single-level optimization problems. Specifically, the hypergradient on each worker relies on the global Jacobian matrix and the inverse of the global Hessian matrix. Directly communicating or computing them on each worker can result in a large communication and computation overhead, such as Yang et al. (2022b); Chen et al. (2022a) in the aforementioned first category, which communicates Jacobian or Hessian matrix in each iteration. To avoid this issue, Zhang et al. (2023) developed the first single-loop decentralized algorithm, which computes and communicates the Hessian-inverse-vector product to reduce both computation and communication overhead. This approach has also been applied to the full gradient method (Dong et al., 2023), stochastic gradient (Zhu et al., 2024), and the momentum-based method (Kong et al., 2024). However, these methods require to compute the second-order Jacobian and Hessian matrix, which can incur large memory and computation overhead for high-dimensional problems. To avoid computing second-order gradients, in the single-machine setting, Shen & Chen (2023); Kwon et al. (2023b;a); Chen et al. (2024) propose converting the bilevel optimization problem into a single-level optimization problem via the penalty approach and then only the first-order gradient is needed to solve it, which can save computation overhead significantly. Based on this reformulation, Wang et al. (2024) developed a decentralized first-order method, which only requires the standard stochastic gradient. Therefore, its practical computational time is much smaller than the second-order gradient based method. However, Wang et al. (2024) still suffers from some limitations. On the one hand, it can only handle the strongly-convex lower-level loss function, which is also a limitation of all aforementioned decentralized methods (Yang et al., 2022b; Gao et al., 2023; Chen et al., 2022a;b; Zhang et al., 2023; Kong et al., 2024; Zhu et al., 2024; Lu et al., 2022; Liu et al., 2022b; 2023a; Wang et al., 2024). On the other hand, Wang et al. (2024) suffers from a quite slow convergence rate, $O(1/T^{1/7})$, where $T$ is the number of iterations, while the first-order method Kwon et al. (2023a) in the single-machine setting can achieve a convergence rate of $O(1/T^{1/5})$. Finally, it is worth noting that all existing bilevel optimization methods, including both single-machine and decentralized settings, assume that the stochastic noise in the gradient has finite variance. Therefore, these algorithms cannot handle heavy-tailed noise.

### 2.2 STOCHASTIC OPTIMIZATION UNDER HEAVY-TAILED NOISE

Some recent works (Zhang et al., 2020) have shown that the finite variance assumption is too restrictive for modern machine learning models. In practice, commonly used deep neural networks, such as image classification models (Simsekli et al., 2019; Battash et al., 2024) and attention-based models (Zhang et al., 2020; Ahn et al., 2023), have stochastic gradients whose noise follows a heavy-tailed distribution. This observation has sparked the recent interest (Zhang et al., 2020; Cutkosky & Mehta, 2021; Liu et al., 2023b; Nguyen et al., 2023; Liu et al., 2024; Liu & Zhou, 2025; Hübler et al., 2024; Sun et al., 2024; Gorbunov et al., 2023) in the study of stochastic optimization under heavy-tailed noise. For example, Zhang et al. (2020) established the in-expectation convergence rate of Clipped SGD for strongly convex and nonconvex loss functions. As discussed earlier, Clipped SGD requires a clipping threshold, which introduces more difficulties for tuning the optimizer. Therefore, some recent efforts (Liu & Zhou, 2025; Hübler et al., 2024; Sun et al., 2024) have been made to get rid of the clipping operation, while keeping the normalization operation. For example, Sun et al. (2024) established the in-expectation convergence rate of normalized SGD based on a strong assumption

of the individual Lipschitz smoothness. Hübler et al. (2024) also achieved this result without using this strong assumption in the cost of a large batch size. However, extending the convergence rate of normalized SGD to normalized SGD with momentum is not trivial. Sun et al. (2024) addressed this problem by assuming a bounded stochastic gradient. Based on this assumption, Sun et al. (2024) further established the in-expectation convergence rate of normalized SGD with variance reduction. Nevertheless, such a strong assumption is easily violated in practice. Recently, Liu & Zhou (2025) developed an innovative approach from the online learning perspective and successfully addressed this issue, establishing the in-expectation convergence rate of normalized SGD without relying on the bounded stochastic gradient assumption. However, it remains unclear whether the approach in Liu & Zhou (2025) can be applied to the normalized SGD with variance reduction.

In the distributed setting, the heavy-tailed noise has been less studied, although Gürbüzbalaban et al. (2024) has shown that noise in the decentralized setting tends to have heavier tails than in the centralized setting. Moreover, existing distributed methods for handling heavy-tailed noise (Sadiev et al., 2023; Yang et al., 2022a; Lee et al., 2025) still rely on the gradient clipping technique. Therefore, it remains unclear whether the gradient normalization technique without assuming bounded gradients works in the decentralized setting. Furthermore, to the best of our knowledge, gradient normalization without clipping has not yet been explored for decentralized bilevel optimization or decentralized minimax optimization under heavy-tailed noise. Thus, it is important to fill this gap.

## 3 PROBLEM SETUP

### 3.1 PROBLEM DEFINITION

In this paper, we assume that there are $K$ workers, indexed by $k \in \{1, 2, \cdots, K\}$, which form a communication graph and perform peer-to-peer communication within it. These workers collaboratively optimize a nonconvex decentralized stochastic bilevel optimization problem, defined as:

$$\min_{x \in \mathbb{R}^{d_1}, y \in y^*(x)} \frac{1}{K} \sum_{k=1}^{K} f^{(k)}(x, y) \qquad s.t. \quad y^*(x) = \arg\min_{y \in \mathbb{R}^{d_2}} \frac{1}{K} \sum_{k=1}^{K} g^{(k)}(x, y). \tag{1}$$

In Eq. (1), $f(x, y) = \frac{1}{K} \sum_{k=1}^{K} f^{(k)}(x, y)$ is the global upper-level loss function, where $f^{(k)}(x, y) = \mathbb{E}[f^{(k)}(x, y; \xi^{(k)})]$ is the local one on the $k$-th worker and $\xi^{(k)}$ denotes random samples on that worker. Additionally, $g(x, y) = \frac{1}{K} \sum_{k=1}^{K} g^{(k)}(x, y)$ is the global lower-level loss function, where $g^{(k)}(x, y) = \mathbb{E}[g^{(k)}(x, y; \zeta^{(k)})]$ is the lower-level one on the $k$-th worker and $\zeta^{(k)}$ represents the corresponding random samples. Unlike existing decentralized bilevel optimization methods (Yang et al., 2022b; Gao et al., 2023; Chen et al., 2022a;b; Zhang et al., 2023; Kong et al., 2024; Zhu et al., 2024; Lu et al., 2022; Liu et al., 2022b; 2023a; Wang et al., 2024), which assume that $g(x, y)$ is strongly convex with respect to $y$, we assume that $g(x, y)$ is a nonconvex loss function with respect to $y$, but satisfies the Polyak-Lojasiewicz (PL) condition with respect to $y$ for any given $x$.

### 3.2 MINIMAX REFORMULATION

Because $g(x, y)$ is nonconvex with respect to $y$, the second-order-based method, which relies on the Hessian inverse with respect to $y$ of $g(x, y)$, is not applicable to Eq. (1). Hence, we employ the first-order-based method to solve it. Specifically, Kwon et al. (2023a) shows that the lower-level subproblem in Eq. (1) can be converted into a constraint: $g(x, y) \leq \min_{z \in \mathbb{R}^{d_y}} g(x, z)$, and then it can be converted into a minimax optimization problem based on the penalty method, which is defined as follows:

$$\min_{x \in \mathbb{R}^{d_1}, y \in \mathbb{R}^{d_2}} \max_{z \in \mathbb{R}^{d_2}} \frac{1}{K} \sum_{k=1}^{K} f^{(k)}(x, y) + \frac{1}{\delta} \left( \frac{1}{K} \sum_{k=1}^{K} g^{(k)}(x, y) - \frac{1}{K} \sum_{k=1}^{K} g^{(k)}(x, z) \right), \tag{2}$$

where $\delta > 0$ denotes the penalty parameter. With this reformulation, we only need to compute the first-order gradient with respect to $x$, $y$, and $z$ to update them.

To solve Eq. (2) and measure its approximation for Eq. (1), we introduce the following functions:

$$\Phi(x) = \min_{y \in y^*(x)} \frac{1}{K} \sum_{k=1}^{K} f^{(k)}(x, y), \ \Phi_\delta(x) = \min_{y \in \mathbb{R}^{d_2}} \max_{z \in \mathbb{R}^{d_2}} \frac{1}{\delta} \frac{1}{K} \sum_{k=1}^{K} h_\delta^{(k)}(x, y) - \frac{1}{\delta} \frac{1}{K} \sum_{k=1}^{K} g^{(k)}(x, z), \tag{3}$$

where $h_\delta^{(k)}(x,y) = \delta f^{(k)}(x,y) + g^{(k)}(x,y)$ and $h_\delta(x,y) = \frac{1}{K}\sum_{k=1}^K h_\delta^{(k)}(x,y)$. Chen et al. (2024) shows that $\Phi_\delta(x)$ can approximate $\Phi(x)$ well, including both their loss functions and gradients, by controlling the penalty parameter $\delta$, which is shown in Appendix C.1. Importantly, $\min_{x\in\mathbb{R}^{d_1}} \Phi_\delta(x)$ is tractable compared to $\min_{x\in\mathbb{R}^{d_1}} \Phi(x)$. With the minimax reformulation, in the single-machine setting, Kwon et al. (2023a) shows that the convergence rate when using first-order stochastic gradients is $O(1/T^{1/7})$ and can be improved to $O(1/T^{1/5})$ when using first-order stochastic variance-reduced gradients. Note that this reformulation for nonconvex bilevel optimization cannot achieve the $O(1/T^{1/3})$ convergence rate as the single-level method when using variance-reduced gradients. In fact, it is still an open problem to achieve that convergence rate. **The purpose of this paper is not to bridge this gap. Instead, our goal is to design a decentralized algorithm to solve Eq. (2) under heavy-tailed noise and then formally show how its solution solves Eq. (1).** Note that there are currently no decentralized minimax optimization methods capable of handling heavy-tailed noise without gradient clipping. Moreover, due to the penalty term, establishing the convergence rate is significantly more challenging than in existing single-level or standard minimax methods. Therefore, *solving Eq. (2) as a mean to solve Eq. (1) under heavy-tailed noise requires new algorithm design and convergence analysis.*

### 3.3 Assumptions

To solve Eq. (1), we introduce some commonly used assumptions, which have been used in existing nonconvex bilevel optimization methods, such as Kwon et al. (2024); Chen et al. (2024).

**Assumption 3.1.** *Let $z = (x,y) \in \mathbb{R}^{d_x} \times \mathbb{R}^{d_y}$, then the upper-level function $f^{(k)}(z)$ and lower-level function $g^{(k)}(z)$ on the $k$-th worker, and the penalty function $h_\delta(z)$ satisfy the following conditions:*

1. *For any $z_1$ and $z_2$, $\mathbb{E}[\|\nabla f^{(k)}(z_1;\xi) - \nabla f^{(k)}(z_2;\xi)\|] \le L_f\|z_1 - z_2\|$ where the constant $L_f > 0$; $\|\nabla_2 f^{(k)}(x,y)\| \le C_f$ where the constant $C_f > 0$; $\mathbb{E}[\|\nabla^2 f^{(k)}(z_1;\xi) - \nabla^2 f^{(k)}(z_2;\xi)\|] \le \ell_f\|z_1 - z_2\|$ where the constant $\ell_f > 0$.*

2. *For any $z_1$ and $z_2$, $\mathbb{E}[\|\nabla g^{(k)}(z_1;\zeta) - \nabla g^{(k)}(z_2;\zeta)\|] \le L_g\|z_1 - z_2\|$ where the constant $L_g > 0$; $\mathbb{E}[\|\nabla^2 g^{(k)}(z_1;\xi) - \nabla^2 g^{(k)}(z_2;\xi)\|] \le \ell_g\|z_1 - z_2\|$ where the constant $\ell_g > 0$.*

3. *$g(x,y)$ satisfies the $\mu$-PL with respect to $y$ where the constant $\mu > 0$; $h_\delta(x,y)$ satisfies the $\mu$-PL with respect to $y$.*

**Assumption 3.2.** *(**heavy-tailed noise**) All first-order and second-order gradients are the unbiased estimators for the corresponding deterministic gradients. Moreover, there exist $s \in (1,2]$ and $\sigma > 0$ such that $\mathbb{E}[\|\nabla f^{(k)}(z;\xi) - \nabla f^{(k)}(z)\|^s] \le \sigma^s$ and $\mathbb{E}[\|\nabla g^{(k)}(z;\xi) - \nabla g^{(k)}(z)\|^s] \le \sigma^s$ for any $z = (x,y) \in \mathbb{R}^{d_x} \times \mathbb{R}^{d_y}$.*

**Assumption 3.3.** *For the adjacency matrix $E = [e_{ij}] \in \mathbb{R}_+^{K \times K}$ of the communication graph, $e_{ij} > 0$ indicates that the $i$-th worker and the $j$-th worker are connected. Otherwise, $e_{ij} = 0$. In addition, $\mathcal{N}_k = \{j | e_{kj} > 0\}$ denotes the neighboring workers of the $k$-th worker. Moreover, it satisfies the following conditions:*

1. *$E^T = E$, $E\mathbf{1} = \mathbf{1}$, $\mathbf{1}^T E = \mathbf{1}^T$, where $\mathbf{1} \in \mathbb{R}^K$ is the vector of all ones.*

2. *Its eigenvalues can be ordered by magnitude as: $|\lambda_K| \le |\lambda_{K-1}| \le \cdots \le |\lambda_2| < |\lambda_1| = 1$.*

*By denoting $\lambda = |\lambda_2|$, the spectral gap is $1 - \lambda$.*

**Notations.** In this paper, we define $\ell = \max\{L_f, L_g, \ell_f, \ell_g\}$, denote the condition number by $\kappa = \ell/\mu$, and represent the gradient with respect to the $i$-th variable with $\nabla_i$.

## 4 Decentralized Normalized Stochastic Gradient Descent Ascent with Variance Reduction Algorithm

### 4.1 Algorithm Design

To solve the reformulated Eq. (2), we developed a novel decentralized normalized stochastic gradient descent ascent with variance reduction (D-NSVRGDA) algorithm, which is presented in Algorithm 1. Specifically, we use the normalized variance-reduced gradient to update three variables: $x$,

---

**Algorithm 1** D-NSVRGDA

---

**Input:** $\eta_x > 0, \eta_y > 0, \eta_z > 0, \gamma_x > 0, \gamma_y > 0, \gamma_z > 0.$

$\quad \mathbb{I}_{t>0} = 1$ when $t > 0$. Otherwise, $\mathbb{I}_{t>0} = 0$. The batch size is $B_0$ when $t = 0$. Otherwise, it is $O(1)$.

$\quad$ Initialization on the $k$-th worker: $x_0^{(k)} = x_0, y_0^{(k)} = y_0, z_0^{(k)} = z_0$,

1: **for** $t = 0, \cdots, T-1$, the $k$-th worker **do**

2: $\quad$ Variance gradient estimators:

$\quad\quad u_{1,t}^{(k)} = (1 - \gamma_x)(u_{1,t-1}^{(k)} - \nabla_1 f^{(k)}(x_{t-1}^{(k)}, y_{t-1}^{(k)}; \xi_t^{(k)}))\mathbb{I}_{t>0} + \nabla_1 f^{(k)}(x_t^{(k)}, y_t^{(k)}; \xi_t^{(k)}),$

$\quad\quad u_{2,t}^{(k)} = (1 - \gamma_x)(u_{2,t-1}^{(k)} - \nabla_1 g^{(k)}(x_{t-1}^{(k)}, y_{t-1}^{(k)}; \zeta_t^{(k)}))\mathbb{I}_{t>0} + \nabla_1 g^{(k)}(x_t^{(k)}, y_t^{(k)}; \zeta_t^{(k)}),$

$\quad\quad u_{3,t}^{(k)} = (1 - \gamma_x)(u_{3,t-1}^{(k)} - \nabla_1 g^{(k)}(x_{t-1}^{(k)}, z_{t-1}^{(k)}; \zeta_t^{(k)}))\mathbb{I}_{t>0} + \nabla_1 g^{(k)}(x_t^{(k)}, z_t^{(k)}; \zeta_t^{(k)}),$

$\quad\quad v_{1,t}^{(k)} = (1 - \gamma_y)(v_{1,t-1}^{(k)} - \nabla_2 f^{(k)}(x_{t-1}^{(k)}, y_{t-1}^{(k)}; \xi_t^{(k)}))\mathbb{I}_{t>0} + \nabla_2 f^{(k)}(x_t^{(k)}, y_t^{(k)}; \xi_t^{(k)}),$

$\quad\quad v_{2,t}^{(k)} = (1 - \gamma_y)(v_{2,t-1}^{(k)} - \nabla_2 g^{(k)}(x_{t-1}^{(k)}, y_{t-1}^{(k)}; \zeta_t^{(k)}))\mathbb{I}_{t>0} + \nabla_2 g^{(k)}(x_t^{(k)}, y_t^{(k)}; \zeta_t^{(k)}),$

$\quad\quad w_{1,t}^{(k)} = (1 - \gamma_z)(w_{1,t-1}^{(k)} - \nabla_2 g^{(k)}(x_{t-1}^{(k)}, z_{t-1}^{(k)}; \zeta_t^{(k)}))\mathbb{I}_{t>0} + \nabla_2 g^{(k)}(x_t^{(k)}, z_t^{(k)}; \zeta_t^{(k)}),$

3: $\quad$ Combine gradient estimators together for each variable:

$\quad\quad u_t^{(k)} = u_{1,t}^{(k)} + \frac{1}{\delta}(u_{2,t}^{(k)} - u_{3,t}^{(k)}), \quad v_t^{(k)} = v_{1,t}^{(k)} + \frac{1}{\delta}v_{2,t}^{(k)}, \quad w_t^{(k)} = \frac{1}{\delta}w_{1,t}^{(k)},$

4: $\quad$ Gradient tracking:

$\quad\quad \tilde{p}_t^{(k)} = (p_{t-1}^{(k)} - u_{t-1}^{(k)})\mathbb{I}_{t>0} + u_t^{(k)}, \quad p_t^{(k)} = \sum_{j \in \mathcal{N}_k} e_{kj}\tilde{p}_t^{(j)},$

$\quad\quad \tilde{q}_t^{(k)} = (q_{t-1}^{(k)} - v_{t-1}^{(k)})\mathbb{I}_{t>0} + v_t^{(k)}, \quad q_t^{(k)} = \sum_{j \in \mathcal{N}_k} e_{kj}\tilde{q}_t^{(j)},$

$\quad\quad \tilde{r}_t^{(k)} = (r_{t-1}^{(k)} - w_{t-1}^{(k)})\mathbb{I}_{t>0} + w_t^{(k)}, \quad r_t^{(k)} = \sum_{j \in \mathcal{N}_k} e_{kj}\tilde{r}_t^{(j)},$

5: $\quad$ Updating:

$\quad\quad \tilde{x}_{t+1}^{(k)} = x_t^{(k)} - \eta_x \frac{p_t^{(k)}}{\|p_t^{(k)}\|}, \quad x_{t+1}^{(k)} = \sum_{j \in \mathcal{N}_k} e_{kj}\tilde{x}_{t+1}^{(j)},$

$\quad\quad \tilde{y}_{t+1}^{(k)} = y_t^{(k)} - \eta_y \frac{q_t^{(k)}}{\|q_t^{(k)}\|}, \quad y_{t+1}^{(k)} = \sum_{j \in \mathcal{N}_k} e_{kj}\tilde{y}_{t+1}^{(j)},$

$\quad\quad \tilde{z}_{t+1}^{(k)} = z_t^{(k)} - \eta_z \frac{r_t^{(k)}}{\|r_t^{(k)}\|}, \quad z_{t+1}^{(k)} = \sum_{j \in \mathcal{N}_k} e_{kj}\tilde{z}_{t+1}^{(j)},$

6: **end for**

---

$y$, and $z$. More specifically, in Step 3 of Algorithm 1, we compute the variance-reduced gradient estimator for three variables as follows:

$$u_t^{(k)} = u_{1,t}^{(k)} + \frac{1}{\delta}\left(u_{2,t}^{(k)} - u_{3,t}^{(k)}\right), \quad v_t^{(k)} = v_{1,t}^{(k)} + \frac{1}{\delta}v_{2,t}^{(k)}, \quad w_t^{(k)} = \frac{1}{\delta}w_{1,t}^{(k)}, \tag{4}$$

In Eq. (4), $u_{1,t}^{(k)}$, $u_{2,t}^{(k)}$, and $u_{3,t}^{(k)}$ are the variance-reduced gradient estimator for $\nabla_1 f^{(k)}(x_t^{(k)}, y_t^{(k)})$, $\nabla_1 g^{(k)}(x_t^{(k)}, y_t^{(k)})$, and $\nabla_1 g^{(k)}(x_t^{(k)}, z_t^{(k)})$, respectively. Similarly, $v_{1,t}^{(k)}$ and $v_{2,t}^{(k)}$ estimate $\nabla_2 f^{(k)}(x_t^{(k)}, y_t^{(k)})$, $\nabla_2 g^{(k)}(x_t^{(k)}, y_t^{(k)})$, respectively, while $w_{1,t}^{(k)}$ is used to estimate $\nabla_2 g^{(k)}(x_t^{(k)}, z_t^{(k)})$. All these gradient estimators are computed using the STORM method (Cutkosky & Orabona, 2019), as described in Step 2 of Algorithm 1, where $\gamma_x \in (0,1)$, $\gamma_y \in (0,1)$, and $\gamma_z \in (0,1)$ are three hyperparameters.

Then, our algorithm uses the gradient tracking approach to communicate these gradient estimators, which is shown in Step 4. For example, $p_t^{(k)} = \sum_{j \in \mathcal{N}_k} e_{kj}\tilde{p}_t^{(j)}$ represents the aggregation of gradient estimators $\tilde{p}_t^{(j)}$ from the neighboring workers $\mathcal{N}_k$ of the $k$-th worker. Finally, in Step 5, our algorithm uses the normalized gradient estimator to update the variables. For example, the $k$-th worker uses the normalized gradient estimator to update its local variable $x$ and communicates the updated variable $\tilde{x}_{t+1}^{(k)}$ as follows:

$$\tilde{x}_{t+1}^{(k)} = x_t^{(k)} - \eta_x \frac{p_t^{(k)}}{\|p_t^{(k)}\|}, \quad x_{t+1}^{(k)} = \sum_{j \in \mathcal{N}_k} e_{kj}\tilde{x}_{t+1}^{(j)}, \tag{5}$$

where $\eta_x > 0$ denotes the learning rate for variable $x$, $\frac{p_t^{(k)}}{\|p_t^{(k)}\|}$ denotes the normalized gradient estimator, and the second equation represents the aggregation of updated variables $\tilde{x}_{t+1}^{(j)}$ from the neighboring workers $\mathcal{N}_k$ of the $k$-th worker. The other two variables are updated in the same way.

In Algorithm 1, we use only the normalized gradient estimator to update variables, without employing gradient clipping. To the best of our knowledge, **this is the first decentralized algorithm for**

**bilevel optimization under heavy-tailed noise that does not rely on gradient clipping**. Furthermore, we believe that our algorithm can also be applied to standard minimax optimization under heavy-tailed noise, for which a decentralized algorithm without gradient clipping is also lacking.

## 4.2 CONVERGENCE RATE

Based on Assumptions 3.1-3.3, we establish the theoretical convergence rate of Algorithm 1 in the following theorem.

**Theorem 4.1.** *Given Assumptions 3.1-3.3, by setting the coefficient as $\gamma_x = \gamma_y = \gamma_z = \min\left\{1, O\left(\frac{K^{\frac{1}{2s+1}}}{T^{\frac{2s}{2s+1}}\sigma^{\frac{3s}{(2s+1)(s-1)}}}\right)\right\}$, the learning rate as $\eta_x = O\left(\frac{1-\lambda}{\kappa^5\ell}\frac{K^{\frac{1}{2s+1}}}{T^{\frac{2s}{2s+1}}\sigma^{\frac{4-s}{2(2s+1)(s-1)}}}\right)$, $\eta_y = \eta_x\frac{4(\delta L_f + L_g)}{\mu}$, $\eta_z = \eta_x\frac{4L_g}{\mu}$, the batch size in the first step as $B_0 = O\left(K^{\frac{2s}{2s+1}}T^{\frac{2s}{2s+1}}\sigma^{\frac{s(4s-1)}{(2s+1)(s-1)^2}}\right)$, the batch size in other steps as $O(1)$, and the penalty parameter as $\delta = O\left(\frac{1}{\kappa^3\ell}\frac{1}{K^{\frac{s-1}{2s+1}}T^{\frac{s-1}{2s+1}}}\right)$, we can obtain the following convergence rate for Algorithm 1:*

$$\frac{1}{T}\sum_{t=0}^{T-1}\mathbb{E}[\|\nabla\Phi(\bar{x}_t)\|] \leq O\left(\frac{\kappa^5\ell}{1-\lambda}\frac{\sigma^{\frac{4-s}{2(2s+1)(s-1)}}}{K^{\frac{1}{2s+1}}T^{\frac{1}{2s+1}}}\right) + O\left(\frac{1}{K^{\frac{s-1}{2s+1}}T^{\frac{s-1}{2s+1}}}\right)$$
$$+ O\left(\frac{\kappa^4\ell\sigma^{\frac{2s-2}{2s+1}}}{K^{\frac{s-1}{2s+1}}T^{\frac{s-1}{2s+1}}}\right) + O\left(\frac{\ell\sigma^{\frac{2}{2s+1}}}{K^{\frac{1}{2s+1}}T^{\frac{1}{2s+1}}}\right). \tag{6}$$

From Theorem 4.1, we can obtain the following conclusions.

1. Because $s \in (1, 2]$, the convergence rate is dominated by $O\left(\frac{1}{K^{\frac{s-1}{2s+1}}T^{\frac{s-1}{2s+1}}}\right)$ in terms of the number of ierations $T$. On the one hand, the spectral gap $1 - \lambda$ affects only the high-order term of the convergence rate. On the other hand, the factor $K^{\frac{s-1}{2s+1}}$ in the dominated term indicates the linear speed up with respect to the number of workers. To the best of our knowledge, this is the first work achieving the linear speed up convergence rate for nonconvex decentralized bilevel optimization under heavy-tailed noise.

2. The convergence rate in Theorem 4.1 can recover the finite-variance setting. Specifically, when $s = 2$, $K = 1$, and not considering other factors, our convergence rate is $O\left(\frac{1}{T^{\frac{1}{5}}}\right)$, which is same as the convergence rate in the single-machine setting in Kwon et al. (2023a).

## 4.3 PROOF SKETCH

Establishing the convergence rate for Algorithm 1 is significantly more challenging than for existing methods (Liu & Zhou, 2025; Hübler et al., 2024) that address single-level problems in a single-machine setting. The main difficulties arise from: 1) **the interaction between gradients with respect to three variables due to the bilevel structure**, and 2) **the consensus error introduced by the decentralized setting**. On the other hand, both challenges are compounded by heavy-tailed noise, which makes the analysis more difficult than that in existing decentralized bilevel optimization methods that rely on the finite variance assumption. **In Appendix B, we provide a proof sketch to demonstrate how these challenges are addressed. In Appendix C, we provide the detailed proof of Theorem 4.1**.

## 5 EXPERIMENT

In our experiments, we evaluate our algorithm on two machine learning applications: hyperparameter optimization and model pruning. Due to space constraints, we present only the results on two synthetic datasets related to hyperparameter optimization here. **Additional experimental results on real-world datasets for both hyperparameter optimization and model pruning are provided in Appendix A**.

## 5.1 HYPERPARAMETER OPTIMIZATION

To validate the performance of D-NSVRGDA, we consider a nonconvex hyperparameter optimization problem, with the corresponding loss function defined in Eq. (7). Specifically, in the lower-level optimization subproblem, we optimize the weights of a two-layer fully connected neural network. Although this is a nonconvex optimization problem, existing work has shown that it can satisfy the Polyak-Łojasiewicz (PL) condition under the overparameterized regime. In the upper-level optimization subproblem, we optimize the hyperparameters that are used to regularize the neural network weights. Formally, it is defined as below:

$$
\min_{x=\{x_1,x_2\}} \frac{1}{K} \sum_{k=1}^{K} \mathcal{L}^{(k)}(y^*(x); \mathcal{D}_{vl}^{(k)})
$$

$$
s.t.\ y^*(x) = \arg\min_{y=\{y_1,y_2\}} \frac{1}{K} \sum_{k=1}^{K} \mathcal{L}^{(k)}(y; \mathcal{D}_{tr}^{(k)}) + \mathcal{R}_1(x) + \mathcal{R}_2(x) ,
$$

(7)

where $y_1 = [y_{1,pq}] \in \mathbb{R}^{d_1 \times d_2}$ is the weight of the first layer, $y_2 = [y_{2,pq}] \in \mathbb{R}^{d_2 \times d_3}$ is the weight of the second layer, $x_1 = [x_{1,q}] \in \mathbb{R}^{d_2}$ and $x_2 = [x_{2,q}] \in \mathbb{R}^{d_3}$ are hyperparameters for the regularization term: $\mathcal{R}_1(x) = \frac{1}{d_2} \sum_{q=1}^{d_2} \exp(x_{1,q}) \frac{1}{d_1} \sum_{p=1}^{d_1} y_{1,pq}^2$ and $\mathcal{R}_2(x) = \frac{1}{d_3} \sum_{q=1}^{d_3} \exp(x_{2,q}) \frac{1}{d_2} \sum_{p=1}^{d_2} y_{2,pq}^2$. In our experiments, $d_1$ is set to the number of input features, $d_2$ is set to 20, and $d_3$ is set to 1 for binary classification.

### 5.1.1 SYNTHETIC DATASET I

We use a synthetic dataset to allow full control over the heavy-tailed noise. Specifically, we generate a binary classification training dataset via $y = \text{sgn}(Xw + \alpha\xi)$, where $X \in \mathbb{R}^{10,000 \times 100}$ is drawn from a standard Gaussian distribution, $w \in \mathbb{R}^{100}$ is also drawn from a standard Gaussian distribution, the noise $\xi \in \mathbb{R}^{10,000}$ is drawn from a heavy-tailed Cauchy distribution, and $\alpha > 0$ is a scalar for controlling the contribution of heavy-tailed noise. These training samples are evenly distributed to eight workers. We then use the same approach to generate the validation and testing set that have the same number of samples.

Since all existing decentralized bilevel optimization algorithms require a strongly convex lower-level loss function, there are no baseline methods applicable to the *nonconvex* bilevel optimization problem in Eq. (7). Therefore, in our experiment, we primarily investigate the effect of gradient normalization in handling heavy-tailed noise. Specifically, we remove the normalization step in Algorithm 1 to create its variant, denoted as D-SVRGDA. In addition, we incorporate gradient clipping into D-SVRGDA to obtain the second baseline method, D-SVRGDA-Clip. We then compare the performance of D-NSVRGDA with that of D-SVRGDA and D-SVRGDA-Clip. For all algorithms, we use identical hyperparameters. In detail, the learning rate is set to 0.001, the coefficient for momentum is set to 0.9, and the penalty parameter is set to 0.3. As for D-SVRGDA-Clip, we use different clipping threshold to fully demonstrate its performance. Additionally, there are eight workers, which are connected into a LINE graph. The batch size on each worker is set to 32.

Moreover, we compare our algorithm with methods developed for nonconvex bilevel problems in the single-machine setting under the bounded-variance assumption: F$^2$BSA (Kwon et al., 2023a). We consider both single-loop and double-loop variants of F$^2$BSA under the decentralized setting. For the double-loop approach, the inner-loop iterations are set to one, five, and ten, which we denote as D-F$^2$BSA-1, D-F$^2$BSA-5, and D-F$^2$BSA-10, respectively. For the single-loop approach, which also uses STORM variance reduction technique, we denote it by D-F$^2$BSA-VR. The learning rates of these baselines are set according to Corollaries 5.2 and 5.5 of Kwon et al. (2023a).

Figure 1 shows the upper-level loss function value and the test accuracy of all methods on different datasets that are generated with different levels of heavy-tailed noise. In detail, we use $\alpha = \{0.2, 0.1, 0.05\}$ for generating three datasets. Both the loss function value and the test accuracy in Figure 1 confirm the effectiveness of our algorithm D-NSVRGDA in accommodating different levels of heavy-tailed noise compared to D-SVRGDA. In addition, we can find that D-SVRGDA-Clip is heavily affected by the clipping threshold $\tau$. Therefore, D-SVRGDA-Clip is much more difficult to tune than our method. Furthermore, the double-loop approach (D-F$^2$BSA) depends heavily on the number of inner-loop iterations; increasing it may improve performance but incurs substantial

computational overhead. Though the single-loop approach with variance reduction (D-F$^2$BSA-VR) performs better than the other baselines, it still remains inferior to our method.

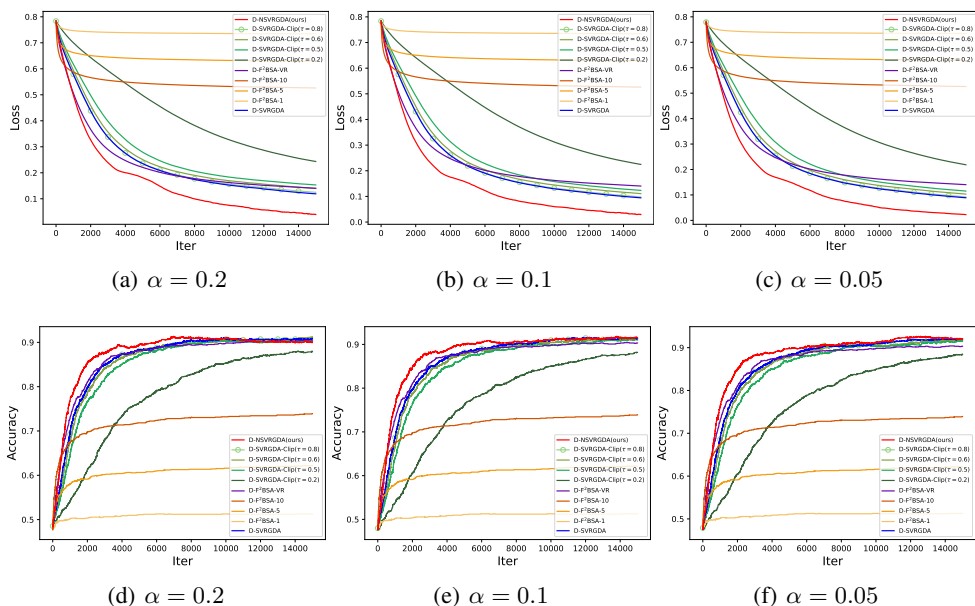

Figure 1: The upper-level loss function value and test accuracy on different datasets that are generated with different levels of heavy-tailed noise. (Add new baselines: different variants of D-F$^2$BSA)

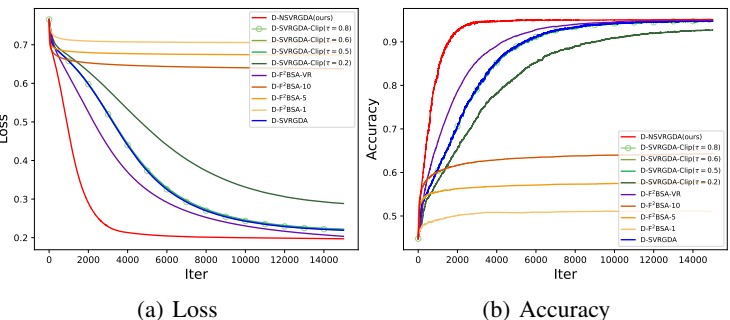

Figure 2: The upper-level loss function value and test accuracy on the second synthetic dataset. (Add new baselines: different variants of D-F$^2$BSA)

### 5.1.2 SYNTHETIC DATASET II

In this experiment, we introduce a new synthetic dataset to simulate heavy-tailed noise in language data. Specifically, in natural language, some words appear much more frequently than others, which actually follow a heavy-tailed distribution. To simulate this phenomenon, we split features into the common and rare features. Specifically, following Lee et al. (2025), we assume $10\%$ features are the common ones, $X_{\text{common}}$, which are drawn from a Bernoulli distribution with the probability being 0.9, and $90\%$ are the rare ones, $X_{\text{rare}}$, which are drawn from a Bernoulli distribution with probability 0.1. Then, the generated samples are represented by $X = [X_{\text{common}}, X_{\text{rare}}]$. Then, we use the same method to generate $w$, $\xi$, and $y$ as the first synthetic dataset, where $\alpha$ is 0.1. Moreover, the total number of features is 100, and the number of samples in the training, validation, and testing sets is 10,000. The other settings are the same as those of the first synthetic dataset. Figure 2 shows the upper-level loss function value and the test accuracy of all methods. Both the loss function value and the test accuracy in Figure 2 further confirm the effectiveness of our algorithm D-NSVRGDA in accommodating heavy-tailed noise compared to other baselines.

## 6 CONCLUSION

Heavy-tailed noise is common in practical machine learning models, yet it has not been studied in the context of decentralized bilevel optimization. To bridge this gap, our paper developed the first decentralized bilevel optimization algorithm to handle heavy-tailed noise in machine learning models that can be formulated as the bilevel optimization problem. Moreover, our paper provided ta theoretical convergence rate for our algorithm under heavy-tailed noise. To the best of our knowledge, this is the first theoretical result for nonconvex decentralized bilevel optimization under heavy-tailed noise. Finally, the extensive experiments validate the effectiveness of the proposed algorithm in handling heavy-tailed noise.

**Ethics statement**   This work complies with the ICLR Code of Ethics. It does not involve human subjects or personal data, and all datasets used are publicly available benchmarks. The research is primarily algorithmic and theoretical and does not pose foreseeable risks to fairness, privacy, or security.

**Reproducibility statement**   We provide the problem setup and assumptions in Section 3, the algorithm design and theoretical analyses in Section 4. A proof sketch and the main proof are included in Appendix B- C. Experimental details are given in Section 5 and Appendix A. The full source code will be released upon acceptance.

**The Use of Large Language Models (LLMs)**   LLMs were employed solely for polishing the writing to enhance the clarity of presentation. They were not involved in the conception of the research process.

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

# A MORE EXPERIMENTS

## A.1 HYPERPARAMETER OPTIMIZATION ON REAL-WORLD DATASETS

In this experiment, we evaluate the performance of D-NSVRGDA on three real-world datasets: a9a, covtype, and IMDB, all of which are available from LIBSVM [1]. The experimental settings, including the communication graph, the batch size, the learning rate, and the penalty parameter, are the same as those in the first two experiments.

Figure 3 shows the upper-level loss function value and the test accuracy of D-NSVRGDA and other baselines on three real-world datasets. Similar to the first two experiments, both the loss function value and the test accuracy in Figure 3 further confirm the effectiveness of our algorithm D-NSVRGDA. In particular, IMDB is a text dataset whose features naturally follow a heavy-tailed distribution, and our algorithm demonstrates significant improvement over the baseline.

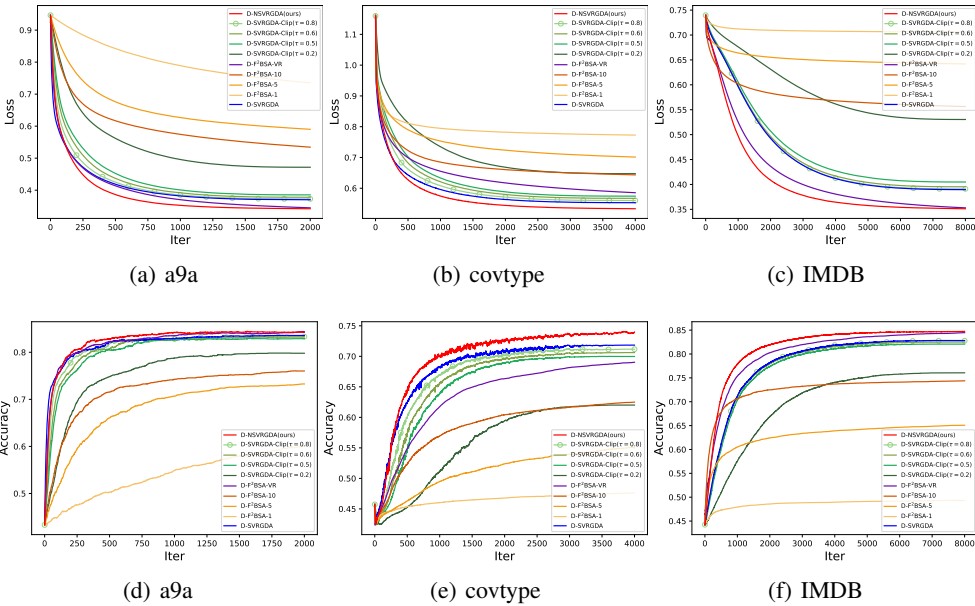

(a) a9a      (b) covtype      (c) IMDB

(d) a9a      (e) covtype      (f) IMDB

Figure 3: The upper-level loss function value and test accuracy on real-world datasets for hyperparameter optimization task. (Add new baselines: different variants of D-F$^2$BSA)

In addition, we provide further experiments under different communication graphs (Figure 4) and different hyperparameter settings (Figure 5). Figure 4 shows that our algorithm consistently outperforms the baselines across different graph topologies. Figure 5 demonstrates that the convergence rate improves with a larger learning rate and a smaller penalty parameter $\delta$.

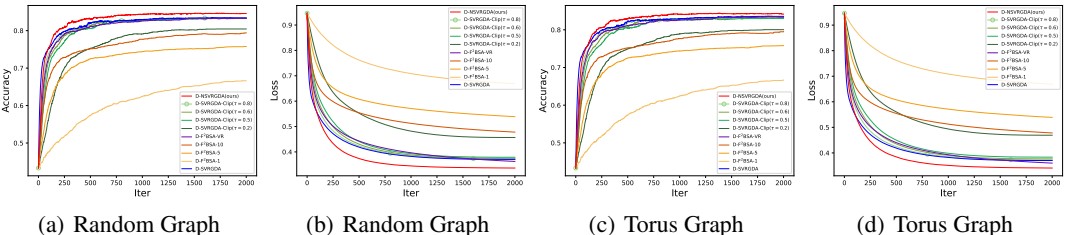

(a) Random Graph    (b) Random Graph    (c) Torus Graph    (d) Torus Graph

Figure 4: The upper-level loss function value and test accuracy under different graphs on a9a dataset.

## A.2 HYPERPARAMETER OPTIMIZATION ON NONCONVEX-STRONGLY-CONVEX BILEVEL OPTIMIZATION PROBLEM

---

[1] https://www.csie.ntu.edu.tw/~cjlin/libsvmtools/datasets/

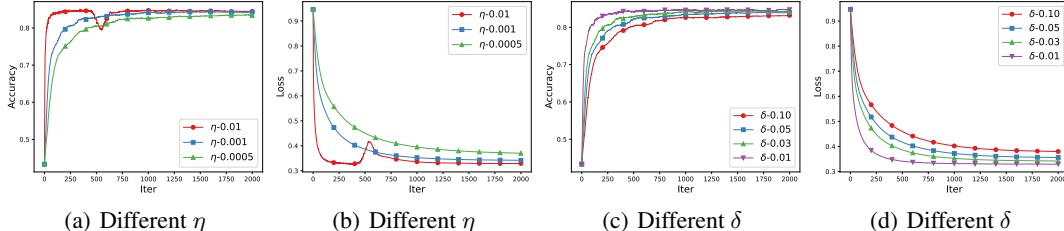

(a) Different $\eta$       (b) Different $\eta$       (c) Different $\delta$       (d) Different $\delta$

Figure 5: The upper-level loss function value and test accuracy under different hyperparameters on a9a dataset.

To provide a more comprehensive comparison with existing decentralized bilevel baselines, we further conduct experiments on a hyperparameter optimization task under a nonconvex–strongly-convex bilevel formulation. Specifically, we focus on the hyperparameter optimization task, where the classifier is a logistic regression model. Then, the lower-level optimization problem is to learn the weight of the logistic regression model, and the upper-level optimization problem is to learn the coefficient of the regularization term like Eq. (7). Since strong convexity implies the PL condition, our method can be directly applied in this setting. In addition to our variant with gradient clipping, we compare against the following representative baselines: DSBO (Chen et al., 2022a), MA-DSBO (Chen et al., 2022b), Gossip-DSBO (Yang et al., 2022b), VRDBO (Gao et al., 2023), DSVRBGD (Zhang et al., 2023), and DSGDA-GT (Wang et al., 2024). Note that all of these methods rely on second-order information for updates, except DSGDA-GT, which is fully first-order. In our experiments, we set the learning rate of these baseline methods according to their theoretical results in the original paper.

From Figure 6, we can clearly observe that our algorithm, which relies only on first-order variance-reduced gradient updates, requires substantially less time to converge compared with methods that depend on second-order Jacobians or Hessians. Although DSGDA-GT also uses fully first-order information, its high complexity of $O(\epsilon^{-7})$ leads to significantly slower convergence, offering limited practical advantage.

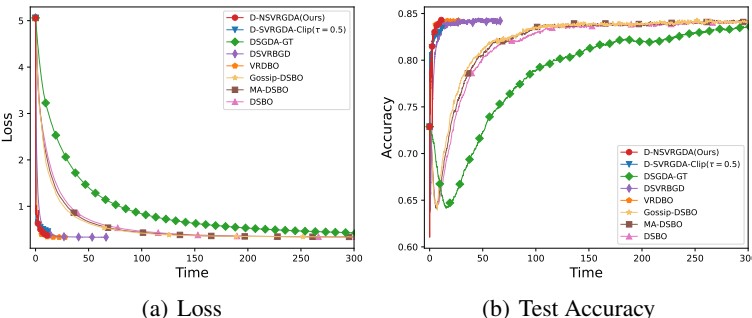

(a) Loss             (b) Test Accuracy

Figure 6: The upper-level loss function value and test accuracy for the nonconvex-strongly-convex bilevel problem with respect to the time consumed (seconds) on a9a dataset.

## A.3 MODEL PRUNING

In this experiment, we verify the performance of our algorithm on the model pruning task. Following Zhang et al. (2022), model pruning can be formulated as a bilevel optimization problem. Formally, in the decentralized setting, its loss function is defined as follows:

$$\min_{x} \frac{1}{K} \sum_{k=1}^{K} \mathcal{L}^{(k)}(x \odot y^*(x))$$

$$s.t. \quad y^*(x) = \arg\min_{y} \frac{1}{K} \sum_{k=1}^{K} \mathcal{L}^{(k)}(x \odot y) , \tag{8}$$

where $y \in \mathbb{R}^d$ denotes the parameter of a deep neural network, and $x \in \{0,1\}^d$ is a binary mask, where 0 indicates pruning the corresponding neuron. Since Liu et al. (2022a) shows that optimizing

an overparameterized deep neural network satisfies the PL condition, the model pruning problem satisfies the nonconvex-PL assumption when pruning a deep neural network. In this experiment, we use the same neural network architecture as in the first three experiments and keep the other experimental settings unchanged. For the pruning rate, we prune $80\%$ of the neurons.

Figure 7 shows the upper-level loss value and test accuracy of D-NSVRGDA and other baselines on the model pruning task defined in Eq. (8). We also evaluate D-SVRGDA-Clip under different clipping threshold values. From the figure, we observe that our algorithm, D-NSVRGDA, consistently outperforms the baseline methods in terms of both the loss value and test accuracy. This further confirms the effectiveness of our algorithm in handling heavy-tailed noise in new applications.

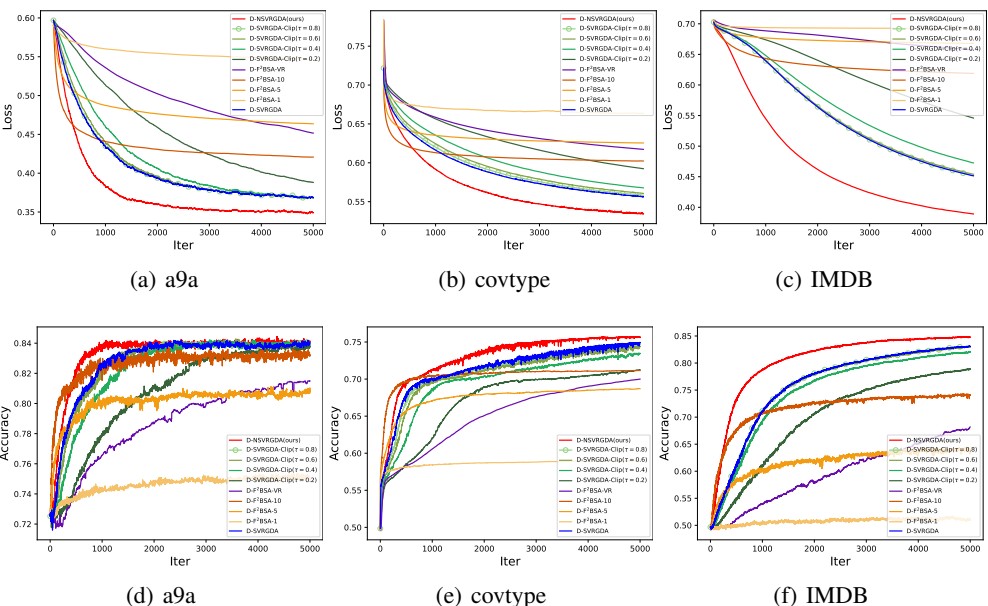

Figure 7: The upper-level loss function value and test accuracy on real-world datasets for model pruning task. (Add new baselines: different variants of D-F$^2$BSA)

### A.4   MODEL PRUNING FOR RNN

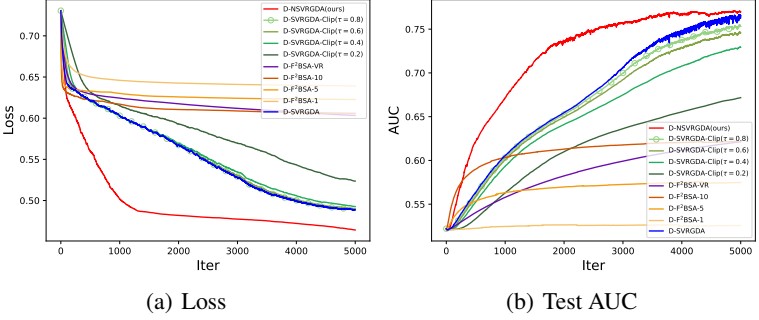

Figure 8: The upper-level loss function value and test AUC score in the RNN pruning task.

In this section, we add an additional experiment to further verify the performance of our algorithm on more complicated applications. Specifically, we consider the model pruning task in Eq. (8) for the text classification application using a recurrent neural network, as the language data typically incurs the heavy-tailed noise. In detail, we use Sentiment140 dataset (Go et al., 2009) and use a two-layer recurrent neural network as the classifier where the embedding size is 300 and the number of hidden neurons is 128. Then, in the lower-level optimization problem, we learn the weight of the recurrent neural network in the lower-level optimization problem, while learning the pruning mask in the upper-level optimization problem. In this experiment, we use the same experimental settings for all methods as the last experiment regarding MLP pruning.

Figure 8 shows the upper-level loss function value and test AUC (area-under-the-curve) score of D-NSVRGDA and all baselines on the RNN pruning task. Similar to the MLP pruning task, our algorithm, D-NSVRGDA, consistently outperforms all baseline methods in terms of both the loss value and test accuracy for the RNN pruning task. This further confirms the effectiveness of our algorithm in handling heavy-tailed noise in large-scale real-world applications.

## B    PROOF SKETCH

Establishing the convergence rate for Algorithm 1 is significantly more challenging than for existing methods (Liu & Zhou, 2025; Hübler et al., 2024) that address single-level problems in a single-machine setting. The main difficulties arise from: 1) **the interaction between gradients with respect to three variables due to the bilevel structure**, and 2) **the consensus error introduced by the decentralized setting**. On the other hand, both challenges are compounded by heavy-tailed noise, which makes the analysis more difficult than that in existing decentralized bilevel optimization methods that rely on the finite variance assumption.

### B.1    NOVELTY OF OUR CONVERGENCE ANALYSIS

Here, we highlight the novelty of our convergence analysis in handling the unique challenges caused by the heavy-tailed noise for the decentralized bilevel optimization. Specifically, bounding $\mathbb{E}[||\nabla\Phi(\bar{x}_t)||]$ is quite challenging in the presence of heavy-tailed noise for nonconvex bilevel optimization. **The reason is that its upper bound relies on the optimization errors regarding $y$ and $z$, and the update of $y$ and $z$ relies on normalized gradient estimators to handle heavy-tailed noise.**

#### B.1.1    NOVELTY OVER METHODS WITH BOUNDED VARIANCE

When there does not exist heavy-tailed noise, the commonly used approach (Kwon et al., 2023a) for handling the optimization errors regarding $y$ and $z$ is to bound $||\bar{y}_t - y_\delta^*(\bar{x}_t)||^2$ and $||\bar{z}_t - y^*(\bar{x}_t)||^2$. However, **this approach does NOT work for the normalized gradient estimator**. The reason is that **it requires the standard stochastic gradient estimator without normalization and requires strong convexity**. Specifically, the second to last step in Lemma C.1 of Kwon et al. (2023a) holds only for the original gradient and strong convexity. As a result, Lemma C.1 of Kwon et al. (2023a) cannot handle the normalized gradient estimator in our algorithm. For example, if using Lemma C.1 of Kwon et al. (2023a) to bound $||\bar{z}_{t+1} - y^*(\bar{x}_{t+1})||^2$, it is incapable of handling $||\bar{z}_t - \eta_z \frac{1}{K}\sum_{k=1}^{K}\frac{r_t^{(k)}}{||r_t^{(k)}||} - y^*(\bar{x}_{t+1})||^2$ **in the presence of the normalized gradient and the absence of strong convexity**.

In our proof, we proposed a novel approach to handle the normalized gradient estimator when bounding the optimization errors regarding $y$ and $z$. Generally speaking, **we bound optimization errors regarding $y$ and $z$ from the perspective of function values, instead of variables**. Specifically, as shown in Lemma B.1, we proposed bounding $\mathbb{E}[||\nabla_2 h_\delta(\bar{x}_t, \bar{y}_t)||]$ and $\mathbb{E}[||\nabla_2 g(\bar{x}_t, \bar{z}_t)||]$, instead of $||\bar{y}_t - y_\delta^*(\bar{x}_t)||$ and $||\bar{z}_t - y^*(\bar{x}_t)||$. For example, to bound $\mathbb{E}[||\nabla_2 h_\delta(\bar{x}_t, \bar{y}_t)||]$, we study the evolvement of the function values: $h_\delta(\bar{x}_{t+1}, \bar{y}_{t+1})$ and $h_\delta^*(\bar{x}_{t+1})$. In particular, by upper bounding $h_\delta(\bar{x}_{t+1}, \bar{y}_{t+1}) - h_\delta(\bar{x}_t, \bar{y}_t)$ and $h_\delta^*(\bar{x}_t) - h_\delta^*(\bar{x}_{t+1})$, where the normalized gradient estimator is much easier to handle and the strong convexity is NOT required, we can obtain the upper bound of $\mathbb{E}[||\nabla_2 h_\delta(\bar{x}_t, \bar{y}_t)||]$ in Lemma C.8. Similarly, we bound $\mathbb{E}[||\nabla_2 g(\bar{x}_t, \bar{z}_t)||]$ in Lemma C.7. As such, we can successfully address the challenge about the optimization error with respect to $y$ and $z$.

#### B.1.2    NOVELTY OVER METHODS FOR SINGLE-LEVEL OPTIMIZATION

The single-level optimization method for heavy-tailed noise, such as Liu & Zhou (2025), **cannot handle the interaction among three variables in our bilevel optimization problems**. For example, when bounding $\mathbb{E}[||\nabla_2 h_\delta(\bar{x}_t, \bar{y}_t)||]$ in Lemma C.8, we need to handle the interaction between $x$ and $y$. The existing single-level approaches (Liu & Zhou, 2025) are NOT capable of handling this interaction.

In our proof, we develop a novel approach to handle the interaction between two variables when bounding $\mathbb{E}[||\nabla_2 h_\delta(\bar{x}_t, \bar{y}_t)||]$ and $\mathbb{E}[||\nabla_2 g(\bar{x}_t, \bar{z}_t)||]$. For example, we use three steps to address this interaction when bounding $\mathbb{E}[||\nabla_2 h_\delta(\bar{x}_t, \bar{y}_t)||]$ in Lemma C.8, which is shown below:

- First, we figure out how the update of $y$ affects the evolvement of the function value $h_\delta(\bar{x}_{t+1}, \bar{y}_{t+1})$, i.e., studying $h_\delta(\bar{x}_{t+1}, \bar{y}_{t+1}) - h_\delta(\bar{x}_{t+1}, \bar{y}_t)$.
- Second, we study how the update of $x$ affects the evolvement of the function value $h_\delta(\bar{x}_{t+1}, \bar{y}_t)$, i.e., bounding $h_\delta(\bar{x}_{t+1}, \bar{y}_t) - h_\delta(\bar{x}_t, \bar{y}_t)$.

- Third, we investigate how the update of $x$ affects the evolvement of $h_\delta^*(\bar{x}_{t+1})$, i.e., bounding $h_\delta^*(\bar{x}_t) - h_\delta^*(\bar{x}_{t+1})$.

Finally, by combining these three upper bounds to obtain the upper bound of $h_\delta(\bar{x}_{t+1}, \bar{y}_{t+1}) - h_\delta^*(\bar{x}_{t+1})$, this can provide the upper bound of the optimization error regarding $y$, i.e., bounding $\mathbb{E}[\|\nabla_2 h_\delta(\bar{x}_t, \bar{y}_t)\|]$. With such a novel approach, we can successfully address the challenge caused by the interaction between three variables.

In summary, our proof is novel and has addressed unique challenges caused by the heavy-tailed noise for nonconvex decentralized bilevel optimization. To the best of our knowledge, this is the first paper proposing this technique to handle the heavy-tailed noise for nonconvex bilevel optimization.

## B.2 SOLUTION FOR THE FIRST CHALLENGE

**First Step:** Given that the gradients with respect to three variables interact with each other, we first disclose how they interact with each other in Lemma B.1.

**Lemma B.1.** *Given Assumption 3.1, we can obtain*

$$\frac{1}{T}\sum_{t=0}^{T-1}\mathbb{E}[\|\nabla\Phi(\bar{x}_t)\|] \leq \frac{\mathbb{E}[\Phi(\bar{x}_0)] - \mathbb{E}[\Phi(\bar{x}_T)]}{T} + 2\frac{1}{T}\sum_{t=0}^{T-1}\underbrace{\mathbb{E}[\|\nabla\Phi(\bar{x}_t) - \nabla\Phi_\delta(\bar{x}_t)\|]}_{\text{Approximation Error caused by the minimax reformulation}}$$

$$+ \frac{2(\delta L_f + L_g)}{\mu}\frac{1}{\delta}\frac{1}{T}\sum_{t=0}^{T-1}\underbrace{\mathbb{E}[\|\nabla_2 h_\delta(\bar{x}_t, \bar{y}_t)\|]}_{\text{Gradient regarding } y} + \frac{2L_g}{\mu}\frac{1}{\delta}\frac{1}{T}\sum_{t=0}^{T-1}\underbrace{\mathbb{E}[\|\nabla_2 g(\bar{x}_t, \bar{z}_t)\|]}_{\text{Gradient regarding } z} + \frac{\eta_x L_\Phi}{2}$$

$$+ \textbf{\textit{Gradient Errors}} + \textbf{\textit{Consensus Errors}} . \tag{9}$$

Here, **Gradient Errors** include $2\frac{1}{T}\sum_{t=0}^{T-1}\mathbb{E}[\|\frac{1}{K}\sum_{k=1}^{K}\nabla_1 f^{(k)}(x_t^{(k)}, y_t^{(k)}) - \frac{1}{K}\sum_{k=1}^{K}u_{1,t}^{(k)}\|]$, $2\frac{1}{\delta}\frac{1}{T}\sum_{t=0}^{T-1}\mathbb{E}[\|\frac{1}{K}\sum_{k=1}^{K}\nabla_1 g^{(k)}(x_t^{(k)}, y_t^{(k)}) - \frac{1}{K}\sum_{k=1}^{K}u_{2,t}^{(k)}\|]$, and $2\frac{1}{\delta}\frac{1}{T}\sum_{t=0}^{T-1}\mathbb{E}[\|\frac{1}{K}\sum_{k=1}^{K}\nabla_1 g^{(k)}(x_t^{(k)}, z_t^{(k)}) - \frac{1}{K}\sum_{k=1}^{K}u_{3,t}^{(k)}\|]$. **Consensus Errors** include: $2(L_f + \frac{2L_g}{\delta})\frac{1}{T}\sum_{t=0}^{T-1}\frac{1}{K}\sum_{k=1}^{K}\mathbb{E}[\|x_t^{(k)} - \bar{x}_t\|]$, $2(L_f + \frac{L_g}{\delta})\frac{1}{T}\sum_{t=0}^{T-1}\frac{1}{K}\sum_{k=1}^{K}\mathbb{E}[\|y_t^{(k)} - \bar{y}_t\|]$, $2\frac{L_g}{\delta}\frac{1}{T}\sum_{t=0}^{T-1}\frac{1}{K}\sum_{k=1}^{K}\mathbb{E}[\|z_t^{(k)} - \bar{z}_t\|]$, and $\frac{1}{T}\sum_{t=0}^{T-1}\frac{1}{K}\sum_{k=1}^{K}\mathbb{E}[\|\bar{p}_t - p_t^{(k)}\|]$, where the first three terms are the consensus error with respect to variables, while the last is about the gradient.

Lemma B.1 discloses that the gradient $\mathbb{E}[\|\nabla\Phi(\bar{x}_t)\|]$ regarding $x$ is influenced by $\mathbb{E}[\|\nabla_2 h_\delta(\bar{x}_t, \bar{y}_t)\|]$ regarding $y$ and $\mathbb{E}[\|\nabla_2 g(\bar{x}_t, \bar{z}_t)\|]$ regarding $z$. Meanwhile, Lemma B.1 reveals that the gradient $\mathbb{E}[\|\nabla\Phi(\bar{x}_t)\|]$ is also affected by the consensus errors regarding both variables and gradients. After revealing this explicit interaction, the remainder of the proof boils down to bounding each factor.

**Second Step:** After revealing the explicit interaction between three gradients, our next step is to bound $\mathbb{E}[\|\nabla_2 h_\delta(\bar{x}_t, \bar{y}_t)\|]$ with respect to $y$ and $\mathbb{E}[\|\nabla_2 g(\bar{x}_t, \bar{z}_t)\|]$ regarding $z$, so that $\mathbb{E}[\|\nabla\Phi(\bar{x}_t)\|]$ can be bounded. However, **this is challenging because $\mathbb{E}[\|\nabla_2 h_\delta(\bar{x}_t, \bar{y}_t)\|]$ is affected by the update of two variables simultaneously (the same applies to $\mathbb{E}[\|\nabla_2 g(\bar{x}_t, \bar{z}_t)\|]$), and is thus affected by two normalized variance-reduced gradients**. In our proof, we innovatively handle those normalized variance-reduced gradients and establish the following lemma.

**Lemma B.2.** *Given Assumption 3.1 and $\eta_x \leq \eta_y \frac{\mu}{2(\delta L_f + L_g)}$, we can obtain that*

$$\frac{1}{\delta}\frac{1}{T}\sum_{t=0}^{T-1}\mathbb{E}[\|\nabla_2 h_\delta(\bar{x}_t, \bar{y}_t)\|] \leq \frac{2(\frac{1}{\delta}\mathbb{E}[h_\delta(\bar{x}_0, \bar{y}_0) - h_\delta^*(\bar{x}_0)] - \frac{1}{\delta}\mathbb{E}[h_\delta(\bar{x}_T, \bar{y}_T) - h_\delta^*(\bar{x}_T)])}{\eta_y T}$$

$$+ \frac{1}{\delta}2\eta_x(\delta L_f + L_g) + \frac{1}{\delta}\eta_y(\delta L_f + L_g) + \frac{1}{\delta}\frac{\eta_x^2(\delta L_f + L_g)}{\eta_y} + \frac{1}{\delta}\frac{\eta_x^2 L_{h_\delta^*}}{\eta_y}$$

$$+ \textbf{\textit{Gradient Errors}} + \textbf{\textit{Consensus Errors}} . \tag{10}$$

Here, $h_\delta^*(x) = h_\delta(x, y^*(x))$ where $y^*(x) = \arg\min_y h_\delta(x, y)$. **Gradient Errors** include: $4\frac{1}{T}\sum_{t=0}^{T-1}\mathbb{E}[\|\frac{1}{K}\sum_{k=1}^{K}\nabla_2 f^{(k)}(x_t^{(k)}, y_t^{(k)}) - \frac{1}{K}\sum_{k=1}^{K}v_{1,t}^{(k)}\|]$ and $4\frac{1}{\delta}\frac{1}{T}\sum_{t=0}^{T-1}\mathbb{E}[\|\frac{1}{K}\sum_{k=1}^{K}\nabla_2 g^{(k)}(x_t^{(k)}, y_t^{(k)}) - \frac{1}{K}\sum_{k=1}^{K}v_{2,t}^{(k)}\|]$. **Consensus Errors** include:

$4 \left( L_f + \frac{L_g}{\delta} \right) \frac{1}{T} \sum_{t=0}^{T-1} \frac{1}{K} \sum_{k=1}^{K} \mathbb{E}[\|\bar{x}_t - x_t^{(k)}\|], 4 \left( L_f + \frac{L_g}{\delta} \right) \frac{1}{T} \sum_{t=0}^{T-1} \frac{1}{K} \sum_{k=1}^{K} \mathbb{E}[\|\bar{y}_t - y_t^{(k)}\|],$
and $2 \frac{1}{T} \sum_{t=0}^{T-1} \frac{1}{K} \sum_{k=1}^{K} \mathbb{E}[\|\bar{q}_t - q_t^{(k)}\|]$.

Lemma B.2 shows that $\mathbb{E}[\|\nabla_2 h_\delta(\bar{x}_t, \bar{y}_t)\|]$ is only affected by **Gradient Errors**, **Consensus Errors**, and some other terms that are not explicitly related to $\mathbb{E}[\|\nabla_2 g(\bar{x}_t, \bar{z}_t)\|]$ and $\mathbb{E}[\|\nabla\Phi(\bar{x}_t)\|]$. Therefore, we only need to provide the upper bound of **Gradient Errors** and **Consensus Errors** in order to bound $\mathbb{E}[\|\nabla_2 h_\delta(\bar{x}_t, \bar{y}_t)\|]$. Similarly, we can bound $\mathbb{E}[\|\nabla_2 g(\bar{x}_t, \bar{z}_t)\|]$ as Lemma B.2, which is deferred to Lemma C.7 in Appendix C.2 due to space limitation.

**Summarization.** First, it is worth noting that our proof is fundamentally different from existing decentralized bilevel optimization (Yang et al., 2022b; Gao et al., 2023; Chen et al., 2022a;b; Zhang et al., 2023; Kong et al., 2024; Zhu et al., 2024; Lu et al., 2022; Liu et al., 2022b; 2023a; Wang et al., 2024) or decentralized minimax optimization (Xian et al., 2021; Zhang et al., 2024; Huang & Chen, 2023) that rely on the finite-variance assumption. For example, the upper bound for $\mathbb{E}[\|\nabla\Phi(\bar{x}_t)\|]$ in those methods has a term with regard to $\sigma^2$, which could be infinity under heavy-tailed noise. On the contrary, our upper bound does not have this kind of terms. In fact, this is the first work showing how to handle the normalized variance-reduced gradient and heavy-tailed noise for decentralized bilevel optimization. Second, from Lemmas B.1, B.2, C.7, we can observe that they all are affected by **Gradient Errors** and **Consensus Errors**. Then, we need to bound them under heavy-tailed noise.

### B.3 SOLUTION FOR THE SECOND CHALLENGE

**First Step.** Since the consensus error regarding gradients involves the gradient estimator, e.g., $u_{1,t}^{(k)}$, it can be influenced by both **stochastic noises** and **gradient errors**. For example, Eq. (93) in Appendix C.5 shows that the consensus error regarding the gradient, $\mathbb{E}[\|p_t^{(k)} - \bar{p}_t\|]$, is influenced by stochastic noises, e.g., $\mathbb{E}[\|\nabla_1 f^{(k)}(x_{j-1}^{(k)}, y_{j-1}^{(k)}) - \nabla_1 f^{(k)}(x_{j-1}^{(k)}, y_{j-1}^{(k)}; \xi_j^{(k)})\|]$, and gradient errors, e.g., $\mathbb{E}[\|u_{1,j-1}^{(k)} - \nabla_1 f^{(k)}(x_{j-1}^{(k)}, y_{j-1}^{(k)})\|]$. Then, our fist step for this challenge is to establish the upper bound for **Gradient Errors**. For example, in Lemma B.3, we establish the upper bound for the Gradient Error, $\mathbb{E}[\|u_{1,t}^{(k)} - \nabla_1 f^{(k)}(x_t^{(k)}, y_t^{(k)})\|]$.

**Lemma B.3.** *Given Assumptions 3.1-3.3, we can obtain*

$$\sum_{k=1}^{K} \mathbb{E}[\|u_{1,t}^{(k)} - \nabla_1 f^{(k)}(x_t^{(k)}, y_t^{(k)})\|] \leq (1-\gamma_x)^t \frac{2\sqrt{2}}{B_0^{1-1/s}} \sigma K + \frac{4\eta_x L_f}{(1-\lambda)\sqrt{\gamma_x}} \sqrt{K}$$
$$+ \frac{4\eta_y L_f}{(1-\lambda)\sqrt{\gamma_x}} \sqrt{K} + 2\sqrt{2}\gamma_x^{1-1/s} \sigma K . \tag{11}$$

Note that bounding gradient errors requires addressing the communication step. Lemma B.3 demonstrates the influence of the spectral gap $1 - \lambda$ on this bound, which differs from the single-machine setting. Similarly, we established other Gradient Errors in Lemmas C.17- C.21 in Appendix C.4.

**Second Step.** The second step is to bound the consensus error regarding gradients in terms of **Gradient Errors**. For example, in Lemma B.4, we provide the upper bound for $\frac{1}{T} \sum_{t=0}^{T-1} \frac{1}{K} \sum_{k=1}^{K} \mathbb{E}[\|p_t^{(k)} - \bar{p}_t\|]$, demonstrating how the heavy-tailed noise $(\sigma)$, hyperparameters $(\eta_x, \eta_y, \eta_z, \gamma_x)$, penalty parameter $(\delta)$, and spectral gap $(1 - \lambda)$ affect this upper bound.

**Lemma B.4.** *Given Assumptions 3.1-3.3, we can obtain*

$$\frac{1}{T} \sum_{t=0}^{T-1} \frac{1}{K} \sum_{k=1}^{K} \mathbb{E}[\|p_t^{(k)} - \bar{p}_t\|] \leq \frac{2\lambda}{(1-\lambda)T} \frac{1}{\sqrt{K}} \sum_{k=1}^{K} \mathbb{E}[\|\nabla_1 f^{(k)}(x_0^{(k)}, y_0^{(k)})\|]$$

$$+ \frac{2\lambda}{(1-\lambda)T} \frac{1}{\delta} \frac{1}{\sqrt{K}} \sum_{k=1}^{K} \mathbb{E}[\|\nabla_1 g^{(k)}(x_0^{(k)}, y_0^{(k)})\|] + \frac{2\lambda}{(1-\lambda)T} \frac{1}{\delta} \frac{1}{\sqrt{K}} \sum_{k=1}^{K} \mathbb{E}[\|\nabla_1 g^{(k)}(x_0^{(k)}, z_0^{(k)})\|]$$

$$+ \frac{\lambda}{(1-\lambda)T} \frac{4\sqrt{2}\sqrt{K}}{B_0^{1-1/s}} \left( 1 + \frac{2}{\delta} \right) \sigma + \frac{\gamma_x \lambda \sqrt{K} \sigma}{(1-\lambda)^{3/2}} \left( 1 + \frac{2}{\delta} \right) + \frac{4\eta_x \lambda \sqrt{K}}{(1-\lambda)^{5/2}} \left( L_f + \frac{2L_g}{\delta} \right)$$

$$+ \frac{4\eta_y \lambda \sqrt{K}}{(1-\lambda)^{5/2}} \left( L_f + \frac{L_g}{\delta} \right) + \frac{4\eta_z \lambda \sqrt{K}}{(1-\lambda)^{5/2}} \frac{L_g}{\delta} + \frac{\lambda \sqrt{K}}{T(1-\lambda)^{3/2}} \frac{2\sqrt{2}\sigma}{B_0^{1-1/s}} \left( 1 + \frac{2}{\delta} \right)$$

$$+ \frac{2\sqrt{2}\gamma_x^{2-1/s} \lambda \sqrt{K}}{(1-\lambda)^{3/2}} \sigma \left( 1 + \frac{2}{\delta} \right) + \frac{4\eta_x \sqrt{\gamma_x} \lambda}{(1-\lambda)^{5/2}} \left( L_f + \frac{2L_g}{\delta} \right) + \frac{4\eta_y \sqrt{\gamma_x} \lambda}{(1-\lambda)^{5/2}} \left( L_f + \frac{2L_g}{\delta} \right) .$$

$$\tag{12}$$

Similarly, we established the upper bounds for other consensus errors regarding gradients in Lemmas C.30, C.31 in Appendix C.5.

After obtaining the upper bounds for the gradients, $\mathbb{E}[\|\nabla_2 h_\delta(\bar{x}_t, \bar{y}_t)\|]$ and $\mathbb{E}[\|\nabla_2 g(\bar{x}_t, \bar{z}_t)\|]$, the upper bounds for **Consensus Errors**, and the upper bounds for **Gradient Errors**, we plug them into Lemma B.1, we can finally obtain the convergence rate of Algorithm 1 in Theorem 4.1.

## C   MAIN PROOF

This section is organized as follows:

1. **Appendix C.1: Supporting Terminologies and Lemmas**
2. **Appendix C.2: Characterizing Interdependence between Gradients**
3. **Appendix C.3: Bounding Consecutive Updates**
4. **Appendix C.4: Bounding Gradient Errors**
5. **Appendix C.5: Bounding Consensus Errors**
6. **Appendix C.6: Proof of Theorem 4.1**

The proof of Theorem 4.1 follows the structure presented in Section B. Specifically, we first characterize the interdependence between different gradients in Appendix C.2 and then bound Gradient Errors in Appendix C.4 and Consensus Errors in Appendix C.5. Based on them, we prove Theorem 4.1 in Appendix C.6.

### C.1   SUPPORTING TERMINOLOGIES AND LEMMAS

We define the following terminologies for convergence analysis:

$$X_t = [x_t^{(1)}, x_t^{(2)}, \cdots, x_t^{(K)}], \quad Y_t = [y_t^{(1)}, y_t^{(2)}, \cdots, y_t^{(K)}], \quad Z_t = [z_t^{(1)}, z_t^{(2)}, \cdots, z_t^{(K)}],$$

$$P_t = [p_t^{(1)}, p_t^{(2)}, \cdots, p_t^{(K)}], \quad Q_t = [q_t^{(1)}, q_t^{(2)}, \cdots, q_t^{(K)}], \quad R_t = [r_t^{(1)}, r_t^{(2)}, \cdots, r_t^{(K)}],$$

$$U_t = [u_t^{(1)}, u_t^{(2)}, \cdots, u_t^{(K)}], \quad V_t = [v_t^{(1)}, v_t^{(2)}, \cdots, v_t^{(K)}], \quad W_t = [w_t^{(1)}, w_t^{(2)}, \cdots, w_t^{(K)}],$$

$$\hat{P}_t = [\frac{p_t^{(1)}}{\|p_t^{(1)}\|}, \frac{p_t^{(2)}}{\|p_t^{(2)}\|}, \cdots, \frac{p_t^{(K)}}{\|p_t^{(K)}\|}], \quad \hat{Q}_t = [\frac{q_t^{(1)}}{\|q_t^{(1)}\|}, \frac{q_t^{(2)}}{\|q_t^{(2)}\|}, \cdots, \frac{q_t^{(K)}}{\|q_t^{(K)}\|}],$$

$$\hat{R}_t = [\frac{r_t^{(1)}}{\|r_t^{(1)}\|}, \frac{r_t^{(2)}}{\|r_t^{(2)}\|}, \cdots, \frac{r_t^{(K)}}{\|r_t^{(K)}\|}],$$

$$\bar{X}_t = X_t \frac{\mathbf{1}\mathbf{1}^T}{K}, \quad \bar{Y}_t = Y_t \frac{\mathbf{1}\mathbf{1}^T}{K}, \quad \bar{Z}_t = Z_t \frac{\mathbf{1}\mathbf{1}^T}{K},$$

$$\bar{P}_t = P_t \frac{\mathbf{1}\mathbf{1}^T}{K}, \quad \bar{Q}_t = Q_t \frac{\mathbf{1}\mathbf{1}^T}{K}, \quad \bar{R}_t = R_t \frac{\mathbf{1}\mathbf{1}^T}{K},$$

$$\bar{\hat{P}}_t = \hat{P}_t \frac{\mathbf{1}\mathbf{1}^T}{K}, \quad \bar{\hat{Q}}_t = \hat{Q}_t \frac{\mathbf{1}\mathbf{1}^T}{K}, \quad \bar{\hat{R}}_t = \hat{R}_t \frac{\mathbf{1}\mathbf{1}^T}{K},$$

$$\bar{U}_t = U_t \frac{\mathbf{1}\mathbf{1}^T}{K}, \quad \bar{V}_t = V_t \frac{\mathbf{1}\mathbf{1}^T}{K}, \quad \bar{W}_t = W_t \frac{\mathbf{1}\mathbf{1}^T}{K}. \tag{13}$$

**Lemma C.1.** *Chen et al. (2024) Given Assumptions 3.1, then $\Phi(x)$ is $L_\Phi$-smooth, where the constant $L_\Phi = O(\ell\kappa^3)$.*

**Lemma C.2.** *Chen et al. (2024) Given Assumptions 3.1, then $Y^*(x)$ is continuous, i.e., for any $x_1, x_2 \in \mathbb{R}^{d_1}$, the following inequality holds:*

$$Dist(Y^*(x_1), Y^*(x_2)) \le C_{y^*}\|x_1 - x_2\|, \tag{14}$$

*where $C_{y^*} = \frac{L_g}{\mu} = O(\kappa)$, $Dist(\cdot, \cdot)$ denotes the distance between two sets.*

**Lemma C.3.** *(Appendix A of Karimi et al. (2016)) Given Assumptions 3.1, the following inequality holds:*

$$\|y^*(x) - z\|^2 \le \frac{1}{\mu^2}\|\nabla_2 g(x, z)\|^2, \qquad \|y_\delta^*(x) - y\|^2 \le \frac{1}{\mu^2}\|\nabla_2 h_\delta(x, y)\|^2. \tag{15}$$

**Lemma C.4.** *Given Assumptions 3.1, then $\nabla g^*(x)$ is continuous and $\nabla h_\delta^*(x)$ is also continuous, i.e., for any $x_1, x_2 \in \mathbb{R}^{d_1}$, the following inequalities hold:*

$$\|\nabla g^*(x_1) - \nabla g^*(x_2)\| \le L_{g^*}\|x_1 - x_2\|, \quad \|\nabla h_\delta^*(x_1) - \nabla h_\delta^*(x_2)\| \le L_{h_\delta^*}\|x_1 - x_2\|, \tag{16}$$

*where $L_{g^*} = L_g(1 + \frac{L_g}{\mu}) = O(\ell\kappa)$ and $L_{h_\delta^*} = (\delta L_f + L_g)(1 + \frac{\delta L_f + L_g}{\mu}) = O(\ell\kappa)$.*

This lemma be easily proved by following Lemma A.5 in Nouiehed et al. (2019).

**Lemma C.5.** *Liu & Zhou (2025) Given random vectors $v_t$ that satisfies $\mathbb{E}[v_t|\mathcal{F}_{t-1}] = 0$, where $\mathcal{F}_{t-1}$ is a natural filtration and $t \in \mathbb{N}$, then the following inequality holds:*

$$\mathbb{E}[\|\sum_{t=1}^{T} v_t\|] \leq 2\sqrt{2}\mathbb{E}[(\sum_{t=1}^{T} \|v_t\|^s)^{\frac{1}{s}}], \tag{17}$$

*where $T \in \mathbb{N}$ and $s \in [1, 2]$.*

## C.2 CHARACTERIZING INTERDEPENDENCE BETWEEN GRADIENTS

**Lemma C.6.** *Given Assumption 3.1, we obtain*

$$\frac{1}{T}\sum_{t=0}^{T-1}\mathbb{E}[\|\nabla\Phi(\bar{x}_t)\|] \leq \frac{\mathbb{E}[\Phi(\bar{x}_0)] - \mathbb{E}[\Phi(\bar{x}_T)]}{T} + 2\frac{1}{T}\sum_{t=0}^{T-1}\mathbb{E}[\|\nabla\Phi(\bar{x}_t) - \nabla\Phi_\delta(\bar{x}_t)\|]$$

$$+ \frac{2(\delta L_f + L_g)}{\mu}\frac{1}{\delta}\frac{1}{T}\sum_{t=0}^{T-1}\mathbb{E}[\|\nabla_2 h_\delta(\bar{x}_t, \bar{y}_t)\|] + \frac{2L_g}{\mu}\frac{1}{\delta}\frac{1}{T}\sum_{t=0}^{T-1}\mathbb{E}[\|\nabla_2 g(\bar{x}_t, \bar{z}_t)\|]$$

$$+ 2\frac{1}{T}\sum_{t=0}^{T-1}\mathbb{E}[\|\frac{1}{K}\sum_{k=1}^{K}\nabla_1 f^{(k)}(x_t^{(k)}, y_t^{(k)}) - \frac{1}{K}\sum_{k=1}^{K}u_{1,t}^{(k)}\|]$$

$$+ 2\frac{1}{\delta}\frac{1}{T}\sum_{t=0}^{T-1}\mathbb{E}[\|\frac{1}{K}\sum_{k=1}^{K}\nabla_1 g^{(k)}(x_t^{(k)}, y_t^{(k)}) - \frac{1}{K}\sum_{k=1}^{K}u_{2,t}^{(k)}\|]$$

$$+ 2\frac{1}{\delta}\frac{1}{T}\sum_{t=0}^{T-1}\mathbb{E}[\|\frac{1}{K}\sum_{k=1}^{K}\nabla_1 g^{(k)}(x_t^{(k)}, z_t^{(k)}) - \frac{1}{K}\sum_{k=1}^{K}u_{3,t}^{(k)}\|]$$

$$+ 2(L_f + \frac{2L_g}{\delta})\frac{1}{T}\sum_{t=0}^{T-1}\frac{1}{K}\sum_{k=1}^{K}\mathbb{E}[\|x_t^{(k)} - \bar{x}_t\|] + 2(L_f + \frac{L_g}{\delta})\frac{1}{T}\sum_{t=0}^{T-1}\frac{1}{K}\sum_{k=1}^{K}\mathbb{E}[\|y_t^{(k)} - \bar{y}_t\|]$$

$$+ 2\frac{L_g}{\delta}\frac{1}{T}\sum_{t=0}^{T-1}\frac{1}{K}\sum_{k=1}^{K}\mathbb{E}[\|z_t^{(k)} - \bar{z}_t\|] + \frac{1}{T}\sum_{t=0}^{T-1}\frac{1}{K}\sum_{k=1}^{K}\mathbb{E}[\|\bar{p}_t - p_t^{(k)}\|] + \frac{\eta_x L_\Phi}{2}. \tag{18}$$

*Proof.* Due to the smoothness of $\Phi(x)$, we obtain

$$\mathbb{E}[\Phi(\bar{x}_{t+1})] \leq \mathbb{E}[\Phi(\bar{x}_t)] + \mathbb{E}[\langle\nabla\Phi(\bar{x}_t), \bar{x}_{t+1} - \bar{x}_t\rangle] + \frac{L_\Phi^2}{2}\mathbb{E}[\|\bar{x}_{t+1} - \bar{x}_t\|^2]$$

$$= \mathbb{E}[\Phi(\bar{x}_t)] - \eta_x\mathbb{E}[\langle\nabla\Phi(\bar{x}_t), \frac{1}{K}\sum_{k=1}^{K}\frac{p_t^{(k)}}{\|p_t^{(k)}\|}\rangle] + \frac{\eta_x^2 L_\Phi^2}{2}\mathbb{E}[\|\frac{1}{K}\sum_{k=1}^{K}\frac{p_t^{(k)}}{\|p_t^{(k)}\|}\|^2]$$

$$\overset{(a)}{=} \mathbb{E}[\Phi(\bar{x}_t)] \underbrace{- \eta_x\mathbb{E}[\langle\nabla\Phi(\bar{x}_t) - \bar{p}_t, \frac{1}{K}\sum_{k=1}^{K}\frac{p_t^{(k)}}{\|p_t^{(k)}\|}\rangle]}_{T_1} \underbrace{- \eta_x\mathbb{E}[\langle\bar{p}_t, \frac{1}{K}\sum_{k=1}^{K}\frac{p_t^{(k)}}{\|p_t^{(k)}\|}\rangle]}_{T_2} + \frac{\eta_x^2 L_\Phi}{2}, \tag{19}$$

where $(a)$ holds due to $\|\frac{p_t^{(k)}}{\|p_t^{(k)}\|}\| = 1$.

For $T_1$, we bound it as follows:

$$T_1 \leq \eta_x\mathbb{E}[\|\nabla\Phi(\bar{x}_t) - \bar{p}_t\|\|\frac{1}{K}\sum_{k=1}^{K}\frac{p_t^{(k)}}{\|p_t^{(k)}\|}\|] \leq \eta_x\mathbb{E}[\|\nabla\Phi(\bar{x}_t) - \bar{p}_t\|]. \tag{20}$$

For $T_2$, we bound it as follows:

$$T_2 = -\eta_x\mathbb{E}[\langle\bar{p}_t, \frac{1}{K}\sum_{k=1}^{K}\frac{p_t^{(k)}}{\|p_t^{(k)}\|} - \frac{\bar{p}_t}{\|\bar{p}_t\|}\rangle] - \eta_x\mathbb{E}[\langle\bar{p}_t, \frac{\bar{p}_t}{\|\bar{p}_t\|}\rangle]$$

$$\leq \eta_x \mathbb{E}[\|\bar{p}_t\|] \| \frac{1}{K} \sum_{k=1}^{K} \frac{p_t^{(k)}}{\|p_t^{(k)}\|} - \frac{\bar{p}_t}{\|\bar{p}_t\|} \|] - \eta_x \mathbb{E}[\|\bar{p}_t\|]$$

$$= \eta_x \mathbb{E}[\|\bar{p}_t\|] \| \frac{1}{K} \sum_{k=1}^{K} \frac{p_t^{(k)}}{\|p_t^{(k)}\|} - \frac{1}{K} \sum_{k=1}^{K} \frac{p_t^{(k)}}{\|\bar{p}_t\|} \|] - \eta_x \mathbb{E}[\|\bar{p}_t\|]$$

$$\leq \eta_x \frac{1}{K} \sum_{k=1}^{K} \mathbb{E}[\|\bar{p}_t\| \|p_t^{(k)}\| \| \frac{1}{\|p_t^{(k)}\|} - \frac{1}{\|\bar{p}_t\|} \|] - \eta_x \mathbb{E}[\|\bar{p}_t\|]$$

$$= \eta_x \frac{1}{K} \sum_{k=1}^{K} \mathbb{E}[\|\bar{p}_t - p_t^{(k)}\|] - \eta_x \mathbb{E}[\|\bar{p}_t\|]$$

$$\overset{(a)}{\leq} \eta_x \frac{1}{K} \sum_{k=1}^{K} \mathbb{E}[\|\bar{p}_t - p_t^{(k)}\|] - \eta_x \mathbb{E}[\|\nabla\Phi(\bar{x}_t)\|] + \eta_x \mathbb{E}[\|\nabla\Phi(\bar{x}_t) - \bar{p}_t\|] , \tag{21}$$

where $(a)$ holds due to the following inequality:

$$\mathbb{E}[\|\nabla\Phi(\bar{x}_t)\|] \leq \mathbb{E}[\|\nabla\Phi(\bar{x}_t) - \bar{p}_t\|] + \mathbb{E}[\|\bar{p}_t\|] . \tag{22}$$

Therefore, we obtain

$$\mathbb{E}[\Phi(\bar{x}_{t+1})] \leq \mathbb{E}[\Phi(\bar{x}_t)] - \eta_x \mathbb{E}[\|\nabla\Phi(\bar{x}_t)\|] + \eta_x \frac{1}{K} \sum_{k=1}^{K} \mathbb{E}[\|\bar{p}_t - p_t^{(k)}\|] + \frac{\eta_x^2 L_\Phi}{2}$$

$$+ 2\eta_x \mathbb{E}[\|\nabla\Phi(\bar{x}_t) - \bar{p}_t\|] . \tag{23}$$

For $\mathbb{E}[\|\nabla\Phi(\bar{x}_t) - \bar{p}_t\|]$, we bound it as follows:

$$\mathbb{E}[\|\nabla\Phi(\bar{x}_t) - \bar{p}_t\|]$$
$$\leq \mathbb{E}[\|\nabla\Phi(\bar{x}_t) - \nabla\Phi_\delta(\bar{x}_t)\|] + \mathbb{E}[\|\nabla\Phi_\delta(\bar{x}_t) - \nabla_x\Phi_\delta(\bar{x}_t, \bar{y}_t, \bar{z}_t)\|]$$
$$+ \mathbb{E}[\|\nabla_x\Phi_\delta(\bar{x}_t, \bar{y}_t, \bar{z}_t) - \bar{p}_t\|]$$
$$\overset{(a)}{\leq} \mathbb{E}[\|\nabla\Phi(\bar{x}_t) - \nabla\Phi_\delta(\bar{x}_t)\|] + (L_f + \frac{L_g}{\delta})\mathbb{E}[\|y_\delta^*(\bar{x}_t) - \bar{y}_t\|] + \frac{L_g}{\delta}\mathbb{E}[\|y^*(\bar{x}_t) - \bar{z}_t\|]$$
$$+ \mathbb{E}[\|\nabla_x\Phi_\delta(\bar{x}_t, \bar{y}_t, \bar{z}_t) - \bar{p}_t\|]$$
$$\overset{(b)}{\leq} \mathbb{E}[\|\nabla\Phi(\bar{x}_t) - \nabla\Phi_\delta(\bar{x}_t)\|] + \frac{1}{\mu}(L_f + \frac{L_g}{\delta})\mathbb{E}[\|\nabla_2 h_\delta(\bar{x}_t, \bar{y}_t)\|] + \frac{1}{\mu}\frac{L_g}{\delta}\mathbb{E}[\|\nabla_2 g(\bar{x}_t, \bar{z}_t)\|]$$
$$+ \mathbb{E}[\|\nabla_1 f(\bar{x}_t, \bar{y}_t) - \bar{u}_{1,t}\|] + \frac{1}{\delta}\mathbb{E}[\|\nabla_1 g(\bar{x}_t, \bar{y}_t) - \bar{u}_{2,t}\|] + \frac{1}{\delta}\mathbb{E}[\|\nabla_1 g(\bar{x}_t, \bar{z}_t) - \bar{u}_{3,t}\|]$$
$$\leq \mathbb{E}[\|\nabla\Phi(\bar{x}_t) - \nabla\Phi_\delta(\bar{x}_t)\|] + \frac{1}{\mu}(L_f + \frac{L_g}{\delta})\mathbb{E}[\|\nabla_2 h_\delta(\bar{x}_t, \bar{y}_t)\|] + \frac{1}{\mu}\frac{L_g}{\delta}\mathbb{E}[\|\nabla_2 g(\bar{x}_t, \bar{z}_t)\|]$$
$$+ \mathbb{E}[\|\nabla_1 f(\bar{x}_t, \bar{y}_t) - \frac{1}{K}\sum_{k=1}^{K}\nabla_1 f^{(k)}(x_t^{(k)}, y_t^{(k)})\|] + \mathbb{E}[\|\frac{1}{K}\sum_{k=1}^{K}\nabla_1 f^{(k)}(x_t^{(k)}, y_t^{(k)}) - \frac{1}{K}\sum_{k=1}^{K}u_{1,t}^{(k)}\|]$$
$$+ \frac{1}{\delta}\mathbb{E}[\|\nabla_1 g(\bar{x}_t, \bar{y}_t) - \frac{1}{K}\sum_{k=1}^{K}\nabla_1 g^{(k)}(x_t^{(k)}, y_t^{(k)})\|] + \frac{1}{\delta}\mathbb{E}[\|\frac{1}{K}\sum_{k=1}^{K}\nabla_1 g^{(k)}(x_t^{(k)}, y_t^{(k)}) - \frac{1}{K}\sum_{k=1}^{K}u_{2,t}^{(k)}\|]$$
$$+ \frac{1}{\delta}\mathbb{E}[\|\nabla_1 g(\bar{x}_t, \bar{z}_t) - \frac{1}{K}\sum_{k=1}^{K}\nabla_1 g^{(k)}(x_t^{(k)}, z_t^{(k)})\|] + \frac{1}{\delta}\mathbb{E}[\|\frac{1}{K}\sum_{k=1}^{K}\nabla_1 g^{(k)}(x_t^{(k)}, z_t^{(k)}) - \frac{1}{K}\sum_{k=1}^{K}u_{3,t}^{(k)}\|]$$
$$\overset{(c)}{\leq} \mathbb{E}[\|\nabla\Phi(\bar{x}_t) - \nabla\Phi_\delta(\bar{x}_t)\|] + \frac{1}{\mu}(L_f + \frac{L_g}{\delta})\mathbb{E}[\|\nabla_2 h_\delta(\bar{x}_t, \bar{y}_t)\|] + \frac{1}{\mu}\frac{L_g}{\delta}\mathbb{E}[\|\nabla_2 g(\bar{x}_t, \bar{z}_t)\|]$$
$$+ \mathbb{E}[\|\frac{1}{K}\sum_{k=1}^{K}\nabla_1 f^{(k)}(x_t^{(k)}, y_t^{(k)}) - \frac{1}{K}\sum_{k=1}^{K}u_{1,t}^{(k)}\|]$$

$$+ \frac{1}{\delta}\mathbb{E}[\|\frac{1}{K}\sum_{k=1}^{K}\nabla_1 g^{(k)}(x_t^{(k)}, y_t^{(k)}) - \frac{1}{K}\sum_{k=1}^{K}u_{2,t}^{(k)}\|]$$

$$+ \frac{1}{\delta}\mathbb{E}[\|\frac{1}{K}\sum_{k=1}^{K}\nabla_1 g^{(k)}(x_t^{(k)}, z_t^{(k)}) - \frac{1}{K}\sum_{k=1}^{K}u_{3,t}^{(k)}\|] + (L_f + \frac{2L_g}{\delta})\frac{1}{K}\sum_{k=1}^{K}\mathbb{E}[\|x_t^{(k)} - \bar{x}_t\|]$$

$$+ (L_f + \frac{L_g}{\delta})\frac{1}{K}\sum_{k=1}^{K}\mathbb{E}[\|y_t^{(k)} - \bar{y}_t\|] + \frac{L_g}{\delta}\frac{1}{K}\sum_{k=1}^{K}\mathbb{E}[\|z_t^{(k)} - \bar{z}_t\|] , \tag{24}$$

where $(a)$ holds due to Assumption 3.1, $(b)$ holds due to Lemma C.3, and $(c)$ holds due to Assumption 3.1.

By combining the above two inequalities, we complete the proof.

$\square$

**Lemma C.7.** *Given Assumption 3.1 and $\eta_x \leq \frac{\mu}{2L_g}\eta_z$, we obtain*

$$\frac{1}{\delta}\frac{1}{T}\sum_{t=0}^{T-1}\mathbb{E}[\|\nabla_2 g(\bar{x}_t, \bar{z}_t)\|] \leq \frac{2(\frac{1}{\delta}\mathbb{E}[g(\bar{x}_0, \bar{z}_0) - g^*(\bar{x}_0)] - \frac{1}{\delta}\mathbb{E}[g(\bar{x}_T, \bar{z}_T) - g^*(\bar{x}_T)])}{\eta_z T}$$

$$+ 4\frac{1}{\delta}\frac{1}{T}\sum_{t=0}^{T-1}\mathbb{E}[\|\frac{1}{K}\sum_{k=1}^{K}\nabla_2 g^{(k)}(x_t^{(k)}, z_t^{(k)}) - \frac{1}{K}\sum_{k=1}^{K}w_{1,t}^{(k)}\|]$$

$$+ 4\frac{L_g}{\delta}\frac{1}{T}\sum_{t=0}^{T-1}\frac{1}{K}\sum_{k=1}^{K}\mathbb{E}[\|x_t^{(k)} - \bar{x}_t\|] + 4\frac{L_g}{\delta}\frac{1}{T}\sum_{t=0}^{T-1}\frac{1}{K}\sum_{k=1}^{K}\mathbb{E}[\|z_t^{(k)} - \bar{z}_t\|]$$

$$+ 2\frac{1}{T}\sum_{t=0}^{T-1}\frac{1}{K}\sum_{k=1}^{K}\mathbb{E}[\|\bar{r}_t - r_t^{(k)}\|] + \frac{1}{\delta}2\eta_x L_g + \frac{1}{\delta}\eta_z L_g + \frac{1}{\delta}\frac{\eta_x^2 L_g}{\eta_z} + \frac{1}{\delta}\frac{\eta_x^2 L_{g^*}}{\eta_z} . \tag{25}$$

*Proof.* Due to the smoothness of $g$, we obtain

$$\frac{1}{\delta}\mathbb{E}[g(\bar{x}_{t+1}, \bar{z}_{t+1})] \leq \frac{1}{\delta}\mathbb{E}[g(\bar{x}_{t+1}, \bar{z}_t)] + \frac{1}{\delta}\mathbb{E}[\langle\nabla_2 g(\bar{x}_{t+1}, \bar{z}_t), \bar{z}_{t+1} - \bar{z}_t\rangle] + \frac{1}{\delta}\frac{L_g}{2}\mathbb{E}[\|\bar{z}_{t+1} - \bar{z}_t\|^2]$$

$$= \frac{1}{\delta}\mathbb{E}[g(\bar{x}_{t+1}, \bar{z}_t)] - \eta_z\mathbb{E}[\langle\frac{1}{\delta}\nabla_2 g(\bar{x}_{t+1}, \bar{z}_t), \frac{1}{K}\sum_{k=1}^{K}\frac{r_t^{(k)}}{\|r_t^{(k)}\|}\rangle] + \frac{1}{\delta}\frac{\eta_z^2 L_g}{2}\mathbb{E}[\|\frac{1}{K}\sum_{k=1}^{K}\frac{r_t^{(k)}}{\|r_t^{(k)}\|}\|^2]$$

$$\stackrel{(a)}{=} \frac{1}{\delta}\mathbb{E}[g(\bar{x}_{t+1}, \bar{z}_t)] + \frac{1}{\delta}\frac{\eta_z^2 L_g}{2}$$

$$\underbrace{-\eta_z\mathbb{E}[\langle\frac{1}{\delta}\nabla_2 g(\bar{x}_{t+1}, \bar{z}_t) - \bar{r}_t, \frac{1}{K}\sum_{k=1}^{K}\frac{r_t^{(k)}}{\|r_t^{(k)}\|}\rangle]}_{T_1} \underbrace{-\eta_z\mathbb{E}[\langle\bar{r}_t, \frac{1}{K}\sum_{k=1}^{K}\frac{r_t^{(k)}}{\|r_t^{(k)}\|}\rangle]}_{T_2} , \tag{26}$$

where $(a)$ holds due to $\|\frac{r_t^{(k)}}{\|r_t^{(k)}\|}\| = 1$.

Similar to the proof of Lemma C.6, for $T_1$, we obtain

$$T_1 \leq \eta_z\mathbb{E}[\|\frac{1}{\delta}\nabla_2 g(\bar{x}_{t+1}, \bar{z}_t) - \bar{r}_t\|]$$

$$\leq \eta_z\mathbb{E}[\|\frac{1}{\delta}\nabla_2 g(\bar{x}_{t+1}, \bar{z}_t) - \frac{1}{\delta}\nabla_2 g(\bar{x}_t, \bar{z}_t)\|] + \eta_z\mathbb{E}[\|\frac{1}{\delta}\nabla_2 g(\bar{x}_t, \bar{z}_t) - \bar{r}_t\|]$$

$$\leq \eta_z\frac{L_g}{\delta}\mathbb{E}[\|\bar{x}_{t+1} - \bar{x}_t\|] + \eta_z\mathbb{E}[\|\frac{1}{\delta}\nabla_2 g(\bar{x}_t, \bar{z}_t) - \bar{r}_t\|]$$

$$= \eta_x\eta_z\frac{L_g}{\delta}\mathbb{E}[\|\frac{1}{K}\sum_{k=1}^{K}\frac{p_t^{(k)}}{\|p_t^{(k)}\|}\|] + \eta_z\mathbb{E}[\|\frac{1}{\delta}\nabla_2 g(\bar{x}_t, \bar{z}_t) - \bar{r}_t\|]$$

$$= \eta_x \eta_z \frac{L_g}{\delta} + \eta_z \mathbb{E}[\|\frac{1}{\delta}\nabla_2 g(\bar{x}_t, \bar{z}_t) - \bar{r}_t\|] . \tag{27}$$

In addition, similar to the proof of Lemma C.6, for $T_2$, we obtain

$$T_2 \leq \eta_z \frac{1}{K} \sum_{k=1}^{K} \mathbb{E}[\|\bar{r}_t - r_t^{(k)}\|] - \eta_z \mathbb{E}[\|\bar{r}_t\|]$$

$$\leq \eta_z \frac{1}{K} \sum_{k=1}^{K} \mathbb{E}[\|\bar{r}_t - r_t^{(k)}\|] - \eta_z \mathbb{E}[\|\frac{1}{\delta}\nabla_2 g(\bar{x}_t, \bar{z}_t)\|] + \eta_z \mathbb{E}[\|\frac{1}{\delta}\nabla_2 g(\bar{x}_t, \bar{z}_t) - \bar{r}_t\|] . \tag{28}$$

Then, we obtain

$$\frac{1}{\delta}\mathbb{E}[g(\bar{x}_{t+1}, \bar{z}_{t+1})] \leq \frac{1}{\delta}\mathbb{E}[g(\bar{x}_{t+1}, \bar{z}_t)] - \eta_z \mathbb{E}[\|\frac{1}{\delta}\nabla_2 g(\bar{x}_t, \bar{z}_t)\|] + \eta_z \frac{1}{K} \sum_{k=1}^{K} \mathbb{E}[\|\bar{r}_t - r_t^{(k)}\|]$$

$$+ 2\eta_z \mathbb{E}[\|\frac{1}{\delta}\nabla_2 g(\bar{x}_t, \bar{z}_t) - \bar{r}_t\|] + \eta_x \eta_z \frac{L_g}{\delta} + \frac{1}{\delta}\frac{\eta_z^2 L_g}{2} . \tag{29}$$

For $\mathbb{E}[\|\frac{1}{\delta}\nabla_2 g(\bar{x}_t, \bar{z}_t) - \bar{r}_t\|]$, we bound it as follows:

$$\mathbb{E}[\|\frac{1}{\delta}\nabla_2 g(\bar{x}_t, \bar{z}_t) - \bar{r}_t\|]$$

$$\leq \mathbb{E}[\|\frac{1}{\delta}\nabla_2 g(\bar{x}_t, \bar{z}_t) - \frac{1}{\delta}\frac{1}{K}\sum_{k=1}^{K}\nabla_2 g^{(k)}(x_t^{(k)}, z_t^{(k)})\|]$$

$$+ \mathbb{E}[\|\frac{1}{\delta}\frac{1}{K}\sum_{k=1}^{K}\nabla_2 g^{(k)}(x_t^{(k)}, z_t^{(k)}) - \frac{1}{\delta}\frac{1}{K}\sum_{k=1}^{K} w_{1,t}^{(k)}\|]$$

$$\overset{(a)}{\leq} \frac{L_g}{\delta}\frac{1}{K}\sum_{k=1}^{K}\mathbb{E}[\|x_t^{(k)} - \bar{x}_t\|] + \frac{L_g}{\delta}\frac{1}{K}\sum_{k=1}^{K}\mathbb{E}[\|z_t^{(k)} - \bar{z}_t\|]$$

$$+ \frac{1}{\delta}\mathbb{E}[\|\frac{1}{K}\sum_{k=1}^{K}\nabla_2 g^{(k)}(x_t^{(k)}, z_t^{(k)}) - \frac{1}{K}\sum_{k=1}^{K} w_{1,t}^{(k)}\|] , \tag{30}$$

where $(a)$ holds due to Assumption 3.1.

By combining the above two inequalities, we obtain

$$\frac{1}{\delta}\mathbb{E}[g(\bar{x}_{t+1}, \bar{z}_{t+1})] \leq \frac{1}{\delta}\mathbb{E}[g(\bar{x}_{t+1}, \bar{z}_t)] - \eta_z \mathbb{E}[\|\frac{1}{\delta}\nabla_2 g(\bar{x}_t, \bar{z}_t)\|] + \eta_z \frac{1}{K} \sum_{k=1}^{K} \mathbb{E}[\|\bar{r}_t - r_t^{(k)}\|]$$

$$+ 2\eta_z \frac{L_g}{\delta}\frac{1}{K}\sum_{k=1}^{K}\mathbb{E}[\|x_t^{(k)} - \bar{x}_t\|] + 2\eta_z \frac{L_g}{\delta}\frac{1}{K}\sum_{k=1}^{K}\mathbb{E}[\|z_t^{(k)} - \bar{z}_t\|]$$

$$+ 2\eta_z \frac{1}{\delta}\mathbb{E}[\|\frac{1}{K}\sum_{k=1}^{K}\nabla_2 g^{(k)}(x_t^{(k)}, z_t^{(k)}) - \frac{1}{K}\sum_{k=1}^{K} w_{1,t}^{(k)}\|] + \eta_x \eta_z \frac{L_g}{\delta} + \frac{1}{\delta}\frac{\eta_z^2 L_g}{2} . \tag{31}$$

Moreover, due to the smoothness of $g$, we obtain

$$\frac{1}{\delta}\mathbb{E}[g(\bar{x}_{t+1}, \bar{z}_t)] \leq \frac{1}{\delta}\mathbb{E}[g(\bar{x}_t, \bar{z}_t)] + \frac{1}{\delta}\mathbb{E}[\langle\nabla_1 g(\bar{x}_t, \bar{z}_t), \bar{x}_{t+1} - \bar{x}_t\rangle] + \frac{1}{\delta}\frac{L_g}{2}\mathbb{E}[\|\bar{x}_{t+1} - \bar{x}_t\|^2]$$

$$= \frac{1}{\delta}\mathbb{E}[g(\bar{x}_t, \bar{z}_t)] + \frac{1}{\delta}\mathbb{E}[\langle\nabla_1 g(\bar{x}_t, \bar{z}_t) - \nabla_x g(\bar{x}_t, y^*(\bar{x}_t)), \bar{x}_{t+1} - \bar{x}_t\rangle]$$

$$+ \frac{1}{\delta}\mathbb{E}[\langle\nabla_x g(\bar{x}_t, y^*(\bar{x}_t)), \bar{x}_{t+1} - \bar{x}_t\rangle] + \frac{1}{\delta}\frac{\eta_x^2 L_g}{2}\mathbb{E}[\|\bar{x}_{t+1} - \bar{x}_t\|^2]$$

$$= \frac{1}{\delta}\mathbb{E}[g(\bar{x}_t, \bar{z}_t)] - \eta_x \mathbb{E}[\langle\frac{1}{\delta}(\nabla_1 g(\bar{x}_t, \bar{z}_t) - \nabla_x g(\bar{x}_t, y^*(\bar{x}_t))), \frac{1}{K}\sum_{k=1}^{K}\frac{p_t^{(k)}}{\|p_t^{(k)}\|}\rangle]$$

$$+ \frac{1}{\delta}\mathbb{E}[\langle\nabla_x g(\bar{x}_t, y^*(\bar{x}_t)), \bar{x}_{t+1} - \bar{x}_t\rangle] + \frac{1}{\delta}\frac{\eta_x^2 L_g}{2}\mathbb{E}[\|\frac{1}{K}\sum_{k=1}^{K}\frac{p_t^{(k)}}{\|p_t^{(k)}\|}\|^2]$$

$$\overset{(a)}{\leq} \frac{1}{\delta}\mathbb{E}[g(\bar{x}_t, \bar{z}_t)] + \eta_x\frac{1}{\delta}\mathbb{E}[\|\nabla_1 g(\bar{x}_t, \bar{z}_t) - \nabla_1 g(\bar{x}_t, y^*(\bar{x}_t))\|]$$

$$+ \frac{1}{\delta}\mathbb{E}[\langle\nabla_x g(\bar{x}_t, y^*(\bar{x}_t)), \bar{x}_{t+1} - \bar{x}_t\rangle] + \frac{1}{\delta}\frac{\eta_x^2 L_g}{2}$$

$$\overset{(b)}{\leq} \frac{1}{\delta}\mathbb{E}[g(\bar{x}_t, \bar{z}_t)] + \eta_x\frac{L_g}{\delta}\mathbb{E}[\|\bar{z}_t - y^*(\bar{x}_t)\|] + \frac{1}{\delta}\mathbb{E}[\langle\nabla_x g(\bar{x}_t, y^*(\bar{x}_t)), \bar{x}_{t+1} - \bar{x}_t\rangle] + \frac{1}{\delta}\frac{\eta_x^2 L_g}{2}$$

$$\overset{(c)}{\leq} \frac{1}{\delta}\mathbb{E}[g(\bar{x}_t, \bar{z}_t)] + \eta_x\frac{L_g}{\mu\delta}\mathbb{E}[\|\nabla_2 g(\bar{x}_t, \bar{z}_t)\|] + \frac{1}{\delta}\mathbb{E}[\langle\nabla_x g(\bar{x}_t, y^*(\bar{x}_t)), \bar{x}_{t+1} - \bar{x}_t\rangle] + \frac{1}{\delta}\frac{\eta_x^2 L_g}{2} ,$$

$$\tag{32}$$

where $(a)$ holds due to $\nabla_x g(\bar{x}_t, y^*(\bar{x}_t)) = \nabla_1 g(\bar{x}_t, y^*(\bar{x}_t)) + \nabla y^*(\bar{x}_t)\nabla_2 g(\bar{x}_t, y^*(\bar{x}_t)) = \nabla_1 g(\bar{x}_t, y^*(\bar{x}_t))$ and $\|\frac{p_t^{(k)}}{\|p_t^{(k)}\|}\| = 1$, $(b)$ holds due to Assumption 3.1, and $(c)$ holds due to Lemma C.3.

Furthermore, due to the smoothness of $g^*(x)$ as shown in Lemma C.4, we obtain

$$\frac{1}{\delta}g^*(\bar{x}_{t+1}) \geq \frac{1}{\delta}g^*(\bar{x}_t) + \frac{1}{\delta}\langle\nabla g^*(\bar{x}_t), \bar{x}_{t+1} - \bar{x}_t\rangle - \frac{1}{\delta}\frac{L_{g^*}}{2}\|\bar{x}_{t+1} - \bar{x}_t\|^2$$

$$= \frac{1}{\delta}g^*(\bar{x}_t) + \frac{1}{\delta}\langle\nabla_x g(\bar{x}_t, y^*(\bar{x}_t)), \bar{x}_{t+1} - \bar{x}_t\rangle - \frac{1}{\delta}\frac{\eta_x^2 L_{g^*}}{2}\|\frac{1}{K}\sum_{k=1}^{K}\frac{p_t^{(k)}}{\|p_t^{(k)}\|}\|^2$$

$$= \frac{1}{\delta}g^*(\bar{x}_t) + \frac{1}{\delta}\langle\nabla_x g(\bar{x}_t, y^*(\bar{x}_t)), \bar{x}_{t+1} - \bar{x}_t\rangle - \frac{1}{\delta}\frac{\eta_x^2 L_{g^*}}{2} . \tag{33}$$

Then, we obtain

$$\frac{1}{\delta}g^*(\bar{x}_t) - \frac{1}{\delta}g^*(\bar{x}_{t+1}) \leq -\frac{1}{\delta}\langle\nabla_x g(\bar{x}_t, y^*(\bar{x}_t)), \bar{x}_{t+1} - \bar{x}_t\rangle + \frac{1}{\delta}\frac{\eta_x^2 L_{g^*}}{2} . \tag{34}$$

Finally, we obtain

$$\frac{1}{\delta}\mathbb{E}[g(\bar{x}_{t+1}, \bar{z}_{t+1})] - \frac{1}{\delta}\mathbb{E}[g^*(\bar{x}_{t+1})]$$

$$= \frac{1}{\delta}\mathbb{E}[g(\bar{x}_{t+1}, \bar{z}_{t+1})] - \frac{1}{\delta}\mathbb{E}[g(\bar{x}_{t+1}, \bar{z}_t)] + \frac{1}{\delta}\mathbb{E}[g(\bar{x}_{t+1}, \bar{z}_t)] - \frac{1}{\delta}\mathbb{E}[g(\bar{x}_t, \bar{z}_t)]$$

$$+ \frac{1}{\delta}\mathbb{E}[g(\bar{x}_t, \bar{z}_t)] - \frac{1}{\delta}\mathbb{E}[g^*(\bar{x}_t)] + \frac{1}{\delta}\mathbb{E}[g^*(\bar{x}_t)] - \frac{1}{\delta}\mathbb{E}[g^*(\bar{x}_{t+1})]$$

$$\leq \frac{1}{\delta}\mathbb{E}[g(\bar{x}_t, \bar{z}_t)] - \frac{1}{\delta}\mathbb{E}[g^*(\bar{x}_t)] - \eta_z\mathbb{E}[\|\frac{1}{\delta}\nabla_2 g(\bar{x}_t, \bar{z}_t)\|] + \eta_z\frac{1}{K}\sum_{k=1}^{K}\mathbb{E}[\|\bar{r}_t - r_t^{(k)}\|]$$

$$+ 2\eta_z\frac{L_g}{\delta}\frac{1}{K}\sum_{k=1}^{K}\mathbb{E}[\|x_t^{(k)} - \bar{x}_t\|] + 2\eta_z\frac{L_g}{\delta}\frac{1}{K}\sum_{k=1}^{K}\mathbb{E}[\|z_t^{(k)} - \bar{z}_t\|]$$

$$+ 2\eta_z\frac{1}{\delta}\mathbb{E}[\|\frac{1}{K}\sum_{k=1}^{K}\nabla_2 g^{(k)}(x_t^{(k)}, z_t^{(k)}) - \frac{1}{K}\sum_{k=1}^{K}w_{1,t}^{(k)}\|] + \eta_x\eta_z\frac{L_g}{\delta} + \frac{1}{\delta}\frac{\eta_z^2 L_g}{2}$$

$$+ \eta_x\frac{L_g}{\mu\delta}\mathbb{E}[\|\nabla_2 g(\bar{x}_t, \bar{z}_t)\|] + \frac{1}{\delta}\mathbb{E}[\langle\nabla_x g(\bar{x}_t, y^*(\bar{x}_t)), \bar{x}_{t+1} - \bar{x}_t\rangle] + \frac{1}{\delta}\frac{\eta_x^2 L_g}{2}$$

$$- \frac{1}{\delta}\mathbb{E}[\langle\nabla_x g(\bar{x}_t, y^*(\bar{x}_t)), \bar{x}_{t+1} - \bar{x}_t\rangle] + \frac{1}{\delta}\frac{\eta_x^2 L_{g^*}}{2}$$

$$= \frac{1}{\delta}\mathbb{E}[g(\bar{x}_t, \bar{z}_t)] - \frac{1}{\delta}\mathbb{E}[g^*(\bar{x}_t)] + \left(\eta_x\frac{L_g}{\mu} - \eta_z\right)\frac{1}{\delta}\mathbb{E}[\|\nabla_2 g(\bar{x}_t, \bar{z}_t)\|] + \eta_z\frac{1}{K}\sum_{k=1}^{K}\mathbb{E}[\|\bar{r}_t - r_t^{(k)}\|]$$

$$+ 2\eta_z \frac{L_g}{\delta} \frac{1}{K} \sum_{k=1}^{K} \mathbb{E}[\|x_t^{(k)} - \bar{x}_t\|] + 2\eta_z \frac{L_g}{\delta} \frac{1}{K} \sum_{k=1}^{K} \mathbb{E}[\|z_t^{(k)} - \bar{z}_t\|] + \frac{1}{\delta} \eta_x \eta_z L_g + \frac{1}{\delta} \frac{\eta_z^2 L_g}{2}$$

$$+ 2\eta_z \frac{1}{\delta} \mathbb{E}[\| \frac{1}{K} \sum_{k=1}^{K} \nabla_2 g^{(k)}(x_t^{(k)}, z_t^{(k)}) - \frac{1}{K} \sum_{k=1}^{K} w_{1,t}^{(k)} \|] + \frac{1}{\delta} \frac{\eta_x^2 L_g}{2} + \frac{1}{\delta} \frac{\eta_x^2 L_{g^*}}{2} . \tag{35}$$

By setting $\eta_x \leq \frac{\mu}{2L_g} \eta_z$, we complete the proof.

$\square$

**Lemma C.8.** *Given Assumption 3.1 and $\eta_x \leq \eta_y \frac{\mu}{2L_{h_\delta}}$, where $L_{h_\delta} = \delta L_f + L_g$, we obtain that*

$$\frac{1}{\delta} \frac{1}{T} \sum_{t=0}^{T-1} \mathbb{E}[\|\nabla_2 h_\delta(\bar{x}_t, \bar{y}_t)\|] \leq \frac{2(\frac{1}{\delta} \mathbb{E}[h_\delta(\bar{x}_0, \bar{y}_0) - h_\delta^*(\bar{x}_0)] - \frac{1}{\delta} \mathbb{E}[h_\delta(\bar{x}_T, \bar{y}_T) - h_\delta^*(\bar{x}_T)])}{\eta_y T}$$

$$+ 4\frac{1}{T} \sum_{t=0}^{T-1} \mathbb{E}[\| \frac{1}{K} \sum_{k=1}^{K} \nabla_2 f^{(k)}(x_t^{(k)}, y_t^{(k)}) - \frac{1}{K} \sum_{k=1}^{K} v_{1,t}^{(k)} \|]$$

$$+ 4\frac{1}{\delta} \frac{1}{T} \sum_{t=0}^{T-1} \mathbb{E}[\| \frac{1}{K} \sum_{k=1}^{K} \nabla_2 g^{(k)}(x_t^{(k)}, y_t^{(k)}) - \frac{1}{K} \sum_{k=1}^{K} v_{2,t}^{(k)} \|]$$

$$+ 4 \left( L_f + \frac{L_g}{\delta} \right) \frac{1}{T} \sum_{t=0}^{T-1} \frac{1}{K} \sum_{k=1}^{K} \mathbb{E}[\|\bar{x}_t - x_t^{(k)}\|] + 4 \left( L_f + \frac{L_g}{\delta} \right) \frac{1}{T} \sum_{t=0}^{T-1} \frac{1}{K} \sum_{k=1}^{K} \mathbb{E}[\|\bar{y}_t - y_t^{(k)}\|]$$

$$+ 2\frac{1}{T} \sum_{t=0}^{T-1} \frac{1}{K} \sum_{k=1}^{K} \mathbb{E}[\|\bar{q}_t - q_t^{(k)}\|] + \frac{1}{\delta} 2\eta_x L_{h_\delta} + \frac{1}{\delta} \eta_y L_{h_\delta} + \frac{1}{\delta} \frac{\eta_x^2 L_{h_\delta}}{\eta_y} + \frac{1}{\delta} \frac{\eta_x^2 L_{h_\delta^*}}{\eta_y} . \tag{36}$$

*Proof.* Given Assumptions 3.1, it is easy to know that $h_\delta(x, y) = \delta f(x, y) + g(x, y)$ is $L_{h_\delta}$-smooth with $L_{h_\delta} = \delta L_f + L_g$.

Then, based on its smoothness, we obtain

$$\frac{1}{\delta} \mathbb{E}[h_\delta(\bar{x}_{t+1}, \bar{y}_{t+1})] \leq \frac{1}{\delta} \mathbb{E}[h_\delta(\bar{x}_{t+1}, \bar{y}_t)] + \frac{1}{\delta} \mathbb{E}[\langle \nabla_2 h_\delta(\bar{x}_{t+1}, \bar{y}_t), \bar{y}_{t+1} - \bar{y}_t \rangle] + \frac{1}{\delta} \frac{L_{h_\delta}}{2} \mathbb{E}[\|\bar{y}_{t+1} - \bar{y}_t\|^2]$$

$$= \frac{1}{\delta} \mathbb{E}[h_\delta(\bar{x}_{t+1}, \bar{y}_t)] - \eta_y \mathbb{E}[\langle \frac{1}{\delta} \nabla_2 h_\delta(\bar{x}_{t+1}, \bar{y}_t), \frac{1}{K} \sum_{k=1}^{K} \frac{q_t^{(k)}}{\|q_t^{(k)}\|} \rangle] + \frac{1}{\delta} \frac{\eta_y^2 L_{h_\delta}}{2} \mathbb{E}[\| \frac{1}{K} \sum_{k=1}^{K} \frac{q_t^{(k)}}{\|q_t^{(k)}\|} \|^2]$$

$$\overset{(a)}{=} \frac{1}{\delta} \mathbb{E}[h_\delta(\bar{x}_{t+1}, \bar{y}_t)] + \frac{1}{\delta} \frac{\eta_y^2 L_{h_\delta}}{2}$$

$$\underbrace{- \eta_y \mathbb{E}[\langle \frac{1}{\delta} \nabla_2 h_\delta(\bar{x}_{t+1}, \bar{y}_t) - \bar{q}_t, \frac{1}{K} \sum_{k=1}^{K} \frac{q_t^{(k)}}{\|q_t^{(k)}\|} \rangle]}_{T_1} \underbrace{- \eta_y \mathbb{E}[\langle \bar{q}_t, \frac{1}{K} \sum_{k=1}^{K} \frac{q_t^{(k)}}{\|q_t^{(k)}\|} \rangle]}_{T_2} , \tag{37}$$

where $(a)$ holds due to $\| \frac{q_t^{(k)}}{\|q_t^{(k)}\|} \| = 1$.

For $T_1$, we bound it as follows:

$$T_1 \leq \eta_y \mathbb{E}[\| \frac{1}{\delta} \nabla_2 h_\delta(\bar{x}_{t+1}, \bar{y}_t) - \bar{q}_t \| \| \frac{1}{K} \sum_{k=1}^{K} \frac{q_t^{(k)}}{\|q_t^{(k)}\|} \|]$$

$$\leq \eta_y \mathbb{E}[\| \frac{1}{\delta} \nabla_2 h_\delta(\bar{x}_{t+1}, \bar{y}_t) - \bar{q}_t \|]$$

$$\leq \eta_y \mathbb{E}[\| \frac{1}{\delta} \nabla_2 h_\delta(\bar{x}_{t+1}, \bar{y}_t) - \frac{1}{\delta} \nabla_2 h_\delta(\bar{x}_t, \bar{y}_t) \|] + \eta_y \mathbb{E}[\| \frac{1}{\delta} \nabla_2 h_\delta(\bar{x}_t, \bar{y}_t) - \bar{q}_t \|]$$

$$\overset{(a)}{\leq} \eta_y L_{h_\delta} \frac{1}{\delta} \mathbb{E}[\|\bar{x}_{t+1} - \bar{x}_t\|] + \eta_y \mathbb{E}[\| \frac{1}{\delta} \nabla_2 h_\delta(\bar{x}_t, \bar{y}_t) - \bar{q}_t \|]$$

$$\overset{(b)}{=} \frac{1}{\delta}\eta_x\eta_y L_{h_\delta} + \eta_y \mathbb{E}[\|\frac{1}{\delta}\nabla_2 h_\delta(\bar{x}_t, \bar{y}_t) - \bar{q}_t\|] \ , \tag{38}$$

where $(a)$ holds due to Assumption 3.1, and $(b)$ holds due to $\|\frac{p_t^{(k)}}{\|p_t^{(k)}\|}\| = 1$.

Similar to the proof of Lemma C.6, for $T_2$, we bound it as follows:

$$T_2 \le \eta_y \frac{1}{K}\sum_{k=1}^{K}\mathbb{E}[\|\bar{q}_t - q_t^{(k)}\|] - \eta_y\mathbb{E}[\|\bar{q}_t\|]$$

$$\le \eta_y\frac{1}{K}\sum_{k=1}^{K}\mathbb{E}[\|\bar{q}_t - q_t^{(k)}\|] - \eta_y\mathbb{E}[\|\frac{1}{\delta}\nabla_2 h_\delta(\bar{x}_t, \bar{y}_t)\|] + \eta_y\mathbb{E}[\|\frac{1}{\delta}\nabla_2 h_\delta(\bar{x}_t, \bar{y}_t) - \bar{q}_t\|] \ . \tag{39}$$

Then, we obtain

$$\frac{1}{\delta}\mathbb{E}[h_\delta(\bar{x}_{t+1}, \bar{y}_{t+1})] \le \frac{1}{\delta}\mathbb{E}[h_\delta(\bar{x}_{t+1}, \bar{y}_t)] - \eta_y\mathbb{E}[\|\frac{1}{\delta}\nabla_2 h_\delta(\bar{x}_t, \bar{y}_t)\|] + \frac{1}{\delta}\frac{\eta_y^2 L_{h_\delta}}{2} + \frac{1}{\delta}\eta_x\eta_y L_{h_\delta}$$

$$+ \eta_y\frac{1}{K}\sum_{k=1}^{K}\mathbb{E}[\|\bar{q}_t - q_t^{(k)}\|] + 2\eta_y\mathbb{E}[\|\frac{1}{\delta}\nabla_2 h_\delta(\bar{x}_t, \bar{y}_t) - \bar{q}_t\|] \ . \tag{40}$$

For $\mathbb{E}[\|\frac{1}{\delta}\nabla_2 h_\delta(\bar{x}_t, \bar{y}_t) - \bar{q}_t\|]$, we bound it as follows:

$$\mathbb{E}[\|\frac{1}{\delta}\nabla_2 h_\delta(\bar{x}_t, \bar{y}_t) - \bar{q}_t\|]$$

$$= \mathbb{E}[\|\frac{1}{\delta}\nabla_2 h_\delta(\bar{x}_t, \bar{y}_t) - \bar{v}_t\|]$$

$$\le \mathbb{E}[\|\nabla_2 f(\bar{x}_t, \bar{y}_t) - \bar{v}_{1,t}\|] + \mathbb{E}[\|\frac{1}{\delta}\nabla_2 g(\bar{x}_t, \bar{y}_t) - \frac{1}{\delta}\bar{v}_{2,t}\|]$$

$$\le \mathbb{E}[\|\nabla_2 f(\bar{x}_t, \bar{y}_t) - \frac{1}{K}\sum_{k=1}^{K}\nabla_2 f^{(k)}(x_t^{(k)}, y_t^{(k)})\|] + \mathbb{E}[\|\frac{1}{K}\sum_{k=1}^{K}\nabla_2 f^{(k)}(x_t^{(k)}, y_t^{(k)}) - \frac{1}{K}\sum_{k=1}^{K}v_{1,t}^{(k)}\|]$$

$$+ \frac{1}{\delta}\mathbb{E}[\|\nabla_2 g(\bar{x}_t, \bar{y}_t) - \frac{1}{K}\sum_{k=1}^{K}\nabla_2 g^{(k)}(x_t^{(k)}, y_t^{(k)})\|] + \frac{1}{\delta}\mathbb{E}[\|\frac{1}{K}\sum_{k=1}^{K}\nabla_2 g^{(k)}(x_t^{(k)}, y_t^{(k)}) - \frac{1}{K}\sum_{k=1}^{K}v_{2,t}^{(k)}\|]$$

$$\overset{(a)}{\le} \left(L_f + \frac{L_g}{\delta}\right)\frac{1}{K}\sum_{k=1}^{K}\mathbb{E}[\|\bar{x}_t - x_t^{(k)}\|] + \left(L_f + \frac{L_g}{\delta}\right)\frac{1}{K}\sum_{k=1}^{K}\mathbb{E}[\|\bar{y}_t - y_t^{(k)}\|]$$

$$+ \mathbb{E}[\|\frac{1}{K}\sum_{k=1}^{K}\nabla_2 f^{(k)}(x_t^{(k)}, y_t^{(k)}) - \frac{1}{K}\sum_{k=1}^{K}v_{1,t}^{(k)}\|]$$

$$+ \frac{1}{\delta}\mathbb{E}[\|\frac{1}{K}\sum_{k=1}^{K}\nabla_2 g^{(k)}(x_t^{(k)}, y_t^{(k)}) - \frac{1}{K}\sum_{k=1}^{K}v_{2,t}^{(k)}\|] \ , \tag{41}$$

where $(a)$ holds due to Assumption 3.1.

By combining the above two inequalities, we obtain

$$\frac{1}{\delta}\mathbb{E}[h_\delta(\bar{x}_{t+1}, \bar{y}_{t+1})] \le \frac{1}{\delta}\mathbb{E}[h_\delta(\bar{x}_{t+1}, \bar{y}_t)] - \eta_y\mathbb{E}[\|\frac{1}{\delta}\nabla_2 h_\delta(\bar{x}_t, \bar{y}_t)\|] + \frac{1}{\delta}\frac{\eta_y^2 L_{h_\delta}}{2} + \frac{1}{\delta}\eta_x\eta_y L_{h_\delta}$$

$$+ \eta_y\frac{1}{K}\sum_{k=1}^{K}\mathbb{E}[\|\bar{q}_t - q_t^{(k)}\|] \tag{42}$$

$$+ 2\eta_y\left(L_f + \frac{L_g}{\delta}\right)\frac{1}{K}\sum_{k=1}^{K}\mathbb{E}[\|\bar{x}_t - x_t^{(k)}\|] + 2\eta_y\left(L_f + \frac{L_g}{\delta}\right)\frac{1}{K}\sum_{k=1}^{K}\mathbb{E}[\|\bar{y}_t - y_t^{(k)}\|]$$

$$+ 2\eta_y\mathbb{E}[\|\frac{1}{K}\sum_{k=1}^{K}\nabla_2 f^{(k)}(x_t^{(k)}, y_t^{(k)}) - \frac{1}{K}\sum_{k=1}^{K}v_{1,t}^{(k)}\|]$$

$$+ 2\eta_y \frac{1}{\delta} \mathbb{E}[\| \frac{1}{K} \sum_{k=1}^{K} \nabla_2 g^{(k)}(x_t^{(k)}, y_t^{(k)}) - \frac{1}{K} \sum_{k=1}^{K} v_{2,t}^{(k)} \|] . \tag{43}$$

In addition, due to the smoothness of $h_\delta(x, y)$, we further obtain

$$\frac{1}{\delta}\mathbb{E}[h_\delta(\bar{x}_{t+1}, \bar{y}_t)] \leq \frac{1}{\delta}\mathbb{E}[h_\delta(\bar{x}_t, \bar{y}_t)] + \frac{1}{\delta}\mathbb{E}[\langle \nabla_1 h_\delta(\bar{x}_t, \bar{y}_t), \bar{x}_{t+1} - \bar{x}_t \rangle] + \frac{1}{\delta}\frac{L_{h_\delta}}{2}\mathbb{E}[\|\bar{x}_{t+1} - \bar{x}_t\|^2]$$

$$= \frac{1}{\delta}\mathbb{E}[h_\delta(\bar{x}_t, \bar{y}_t)] + \frac{1}{\delta}\frac{L_{h_\delta}}{2}\mathbb{E}[\|\bar{x}_{t+1} - \bar{x}_t\|^2]$$

$$+ \mathbb{E}[\langle \frac{1}{\delta}\nabla_1 h_\delta(\bar{x}_t, \bar{y}_t) - \frac{1}{\delta}\nabla h_\delta^*(\bar{x}_t), \bar{x}_{t+1} - \bar{x}_t \rangle] + \mathbb{E}[\langle \frac{1}{\delta}\nabla h_\delta^*(\bar{x}_t), \bar{x}_{t+1} - \bar{x}_t \rangle]$$

$$= \frac{1}{\delta}\mathbb{E}[h_\delta(\bar{x}_t, \bar{y}_t)] + \frac{1}{\delta}\frac{\eta_x^2 L_{h_\delta}}{2}\mathbb{E}[\| \frac{1}{K} \sum_{k=1}^{K} \frac{p_t^{(k)}}{\|p_t^{(k)}\|} \|^2] + \mathbb{E}[\langle \frac{1}{\delta}\nabla_x h_\delta(\bar{x}_t, y_\delta^*(\bar{x}_t)), \bar{x}_{t+1} - \bar{x}_t \rangle$$

$$- \eta_x \mathbb{E}[\langle \frac{1}{\delta}\nabla_1 h_\delta(\bar{x}_t, \bar{y}_t) - \frac{1}{\delta}\nabla_x h_\delta(\bar{x}_t, y_\delta^*(\bar{x}_t)), \frac{1}{K} \sum_{k=1}^{K} \frac{p_t^{(k)}}{\|p_t^{(k)}\|} \rangle]]$$

$$\overset{(a)}{\leq} \frac{1}{\delta}\mathbb{E}[h_\delta(\bar{x}_t, \bar{y}_t)] + \eta_x \frac{1}{\delta}\mathbb{E}[\|\nabla_1 h_\delta(\bar{x}_t, \bar{y}_t) - \nabla_1 h_\delta(\bar{x}_t, y_\delta^*(\bar{x}_t))\|]$$

$$+ \mathbb{E}[\langle \frac{1}{\delta}\nabla_x h_\delta(\bar{x}_t, y_\delta^*(\bar{x}_t)), \bar{x}_{t+1} - \bar{x}_t \rangle] + \frac{1}{\delta}\frac{\eta_x^2 L_{h_\delta}}{2}$$

$$\overset{(b)}{\leq} \frac{1}{\delta}\mathbb{E}[h_\delta(\bar{x}_t, \bar{y}_t)] + \eta_x \frac{L_{h_\delta}}{\delta}\mathbb{E}[\|\bar{y}_t - y_\delta^*(\bar{x}_t)\|] + \mathbb{E}[\langle \frac{1}{\delta}\nabla_x h_\delta(\bar{x}_t, y_\delta^*(\bar{x}_t)), \bar{x}_{t+1} - \bar{x}_t \rangle] + \frac{1}{\delta}\frac{\eta_x^2 L_{h_\delta}}{2}$$

$$\overset{(c)}{\leq} \frac{1}{\delta}\mathbb{E}[h_\delta(\bar{x}_t, \bar{y}_t)] + \eta_x \frac{1}{\delta}\frac{L_{h_\delta}}{\mu}\mathbb{E}[\|\nabla_2 h_\delta(\bar{x}_t, \bar{y}_t)\|] + \mathbb{E}[\langle \frac{1}{\delta}\nabla_x h_\delta(\bar{x}_t, y_\delta^*(\bar{x}_t)), \bar{x}_{t+1} - \bar{x}_t \rangle] + \frac{1}{\delta}\frac{\eta_x^2 L_{h_\delta}}{2} ,$$
$$\tag{44}$$

where $(a)$ holds due to $\|\frac{p_t^{(k)}}{\|p_t^{(k)}\|}\| = 1$ and $\nabla h_\delta^*(\bar{x}_t) = \nabla_x h_\delta(\bar{x}_t, y_\delta^*(\bar{x}_t)) = \nabla_1 h_\delta(\bar{x}_t, y_\delta^*(\bar{x}_t)) + \nabla y_\delta^*(\bar{x}_t)\nabla_2 h_\delta(\bar{x}_t, y_\delta^*(\bar{x}_t)) = \nabla_1 h_\delta(\bar{x}_t, y_\delta^*(\bar{x}_t))$, $(b)$ holds due to Assumption 3.1, and $(c)$ holds due to Lemma C.3.

Furthermore, due to the smoothness of $h_\delta^*(x_t)$ as shown in Lemma C.4, we obtain

$$\frac{1}{\delta}h_\delta^*(\bar{x}_{t+1}) \geq \frac{1}{\delta}h_\delta^*(\bar{x}_t) + \frac{1}{\delta}\langle \nabla h_\delta^*(\bar{x}_t), \bar{x}_{t+1} - \bar{x}_t \rangle - \frac{1}{\delta}\frac{L_{h_\delta^*}}{2}\|\bar{x}_{t+1} - \bar{x}_t\|^2$$

$$= \frac{1}{\delta}h_\delta^*(\bar{x}_t) + \frac{1}{\delta}\langle \nabla_x h_\delta(\bar{x}_t, y_\delta^*(\bar{x}_t)), \bar{x}_{t+1} - \bar{x}_t \rangle - \frac{1}{\delta}\frac{\eta_x^2 L_{h_\delta^*}}{2}\| \frac{1}{K} \sum_{k=1}^{K} \frac{p_t^{(k)}}{\|p_t^{(k)}\|} \|^2$$

$$= \frac{1}{\delta}h_\delta^*(\bar{x}_t) + \frac{1}{\delta}\langle \nabla_x h_\delta(\bar{x}_t, y_\delta^*(\bar{x}_t)), \bar{x}_{t+1} - \bar{x}_t \rangle - \frac{1}{\delta}\frac{\eta_x^2 L_{h_\delta^*}}{2} . \tag{45}$$

Then, we obtain

$$\frac{1}{\delta}h_\delta^*(\bar{x}_t) - \frac{1}{\delta}h_\delta^*(\bar{x}_{t+1}) \leq -\frac{1}{\delta}\langle \nabla_x h_\delta(\bar{x}_t, y_\delta^*(\bar{x}_t)), \bar{x}_{t+1} - \bar{x}_t \rangle + \frac{1}{\delta}\frac{\eta_x^2 L_{h_\delta^*}}{2} . \tag{46}$$

Finally, we obtain

$$\frac{1}{\delta}\mathbb{E}[h_\delta(\bar{x}_{t+1}, \bar{y}_{t+1})] - \frac{1}{\delta}\mathbb{E}[h_\delta^*(\bar{x}_{t+1})]$$

$$= \frac{1}{\delta}\mathbb{E}[h_\delta(\bar{x}_{t+1}, \bar{y}_{t+1})] - \frac{1}{\delta}\mathbb{E}[h_\delta(\bar{x}_{t+1}, \bar{y}_t)] + \frac{1}{\delta}\mathbb{E}[h_\delta(\bar{x}_{t+1}, \bar{y}_t)] - \frac{1}{\delta}\mathbb{E}[h_\delta(\bar{x}_t, \bar{y}_t)]$$

$$+ \frac{1}{\delta}\mathbb{E}[h_\delta(\bar{x}_t, \bar{y}_t)] - \frac{1}{\delta}\mathbb{E}[h_\delta^*(\bar{x}_t)] + \frac{1}{\delta}\mathbb{E}[h_\delta^*(\bar{x}_t)] - \frac{1}{\delta}\mathbb{E}[h_\delta^*(\bar{x}_{t+1})]$$

$$\leq \frac{1}{\delta}\mathbb{E}[h_\delta(\bar{x}_t, \bar{y}_t)] - \frac{1}{\delta}\mathbb{E}[h_\delta^*(\bar{x}_t)]$$

$$- \eta_y \mathbb{E}[\|\frac{1}{\delta}\nabla_2 h_\delta(\bar{x}_t, \bar{y}_t)\|] + \frac{1}{\delta}\frac{\eta_y^2 L_{h_\delta}}{2} + \frac{1}{\delta}\eta_x \eta_y L_{h_\delta} + \eta_y \frac{1}{K}\sum_{k=1}^{K}\mathbb{E}[\|\bar{q}_t - q_t^{(k)}\|]$$

$$+ 2\eta_y \left( L_f + \frac{L_g}{\delta} \right) \frac{1}{K}\sum_{k=1}^{K}\mathbb{E}[\|\bar{x}_t - x_t^{(k)}\|] + 2\eta_y \left( L_f + \frac{L_g}{\delta} \right) \frac{1}{K}\sum_{k=1}^{K}\mathbb{E}[\|\bar{y}_t - y_t^{(k)}\|]$$

$$+ 2\eta_y \mathbb{E}[\|\frac{1}{K}\sum_{k=1}^{K}\nabla_2 f^{(k)}(x_t^{(k)}, y_t^{(k)}) - \frac{1}{K}\sum_{k=1}^{K}v_{1,t}^{(k)}\|]$$

$$+ 2\eta_y \frac{1}{\delta}\mathbb{E}[\|\frac{1}{K}\sum_{k=1}^{K}\nabla_2 g^{(k)}(x_t^{(k)}, y_t^{(k)}) - \frac{1}{K}\sum_{k=1}^{K}v_{2,t}^{(k)}\|]$$

$$+ \eta_x \frac{1}{\delta}\frac{L_{h_\delta}}{\mu}\mathbb{E}[\|\nabla_2 h_\delta(\bar{x}_t, \bar{y}_t)\|] + \mathbb{E}[\langle \frac{1}{\delta}\nabla_x h_\delta(\bar{x}_t, y_\delta^*(\bar{x}_t)), \bar{x}_{t+1} - \bar{x}_t\rangle] + \frac{1}{\delta}\frac{\eta_x^2 L_{h_\delta}}{2}$$

$$- \frac{1}{\delta}\mathbb{E}[\langle \nabla_x h_\delta(\bar{x}_t, y_\delta^*(\bar{x}_t)), \bar{x}_{t+1} - \bar{x}_t\rangle] + \frac{1}{\delta}\frac{\eta_x^2 L_{h_\delta^*}}{2}$$

$$= \frac{1}{\delta}\mathbb{E}[h_\delta(\bar{x}_t, \bar{y}_t)] - \frac{1}{\delta}\mathbb{E}[h_\delta^*(\bar{x}_t)] + \left( \eta_x \frac{L_{h_\delta}}{\mu} - \eta_y \right)\frac{1}{\delta}\mathbb{E}[\|\nabla_2 h_\delta(\bar{x}_t, \bar{y}_t)\|] + \eta_y \frac{1}{K}\sum_{k=1}^{K}\mathbb{E}[\|\bar{q}_t - q_t^{(k)}\|]$$

$$+ 2\eta_y \left( L_f + \frac{L_g}{\delta} \right)\frac{1}{K}\sum_{k=1}^{K}\mathbb{E}[\|\bar{x}_t - x_t^{(k)}\|] + 2\eta_y \left( L_f + \frac{L_g}{\delta} \right)\frac{1}{K}\sum_{k=1}^{K}\mathbb{E}[\|\bar{y}_t - y_t^{(k)}\|]$$

$$+ 2\eta_y \mathbb{E}[\|\frac{1}{K}\sum_{k=1}^{K}\nabla_2 f^{(k)}(x_t^{(k)}, y_t^{(k)}) - \frac{1}{K}\sum_{k=1}^{K}v_{1,t}^{(k)}\|] + \frac{1}{\delta}\frac{\eta_x^2 L_{h_\delta}}{2} + \frac{1}{\delta}\frac{\eta_x^2 L_{h_\delta^*}}{2}$$

$$+ 2\eta_y \frac{1}{\delta}\mathbb{E}[\|\frac{1}{K}\sum_{k=1}^{K}\nabla_2 g^{(k)}(x_t^{(k)}, y_t^{(k)}) - \frac{1}{K}\sum_{k=1}^{K}v_{2,t}^{(k)}\|] + \frac{1}{\delta}\frac{\eta_y^2 L_{h_\delta}}{2} + \frac{1}{\delta}\eta_x \eta_y L_{h_\delta} \,. \tag{47}$$

By setting $\eta_x \le \eta_y \frac{\mu}{2 L_{h_\delta}}$, we complete the proof.

$\square$

## C.3 BOUNDING CONSECUTIVE UPDATES

**Lemma C.9.** *Given Assumptions 3.1-3.3, we obtain*

$$\sum_{k=1}^{K}\mathbb{E}[\|x_{t+1}^{(k)} - x_t^{(k)}\|] \le \frac{4\eta_x}{1-\lambda}K \,; \quad \sum_{k=1}^{K}\mathbb{E}[\|y_{t+1}^{(k)} - y_t^{(k)}\|] \le \frac{4\eta_y}{1-\lambda}K \,;$$

$$\sum_{k=1}^{K}\mathbb{E}[\|z_{t+1}^{(k)} - z_t^{(k)}\|] \le \frac{4\eta_z}{1-\lambda}K \,. \tag{48}$$

*Proof.*

$$\mathbb{E}[\|X_{t+1} - X_t\|_F^2] = \mathbb{E}[\|(X_t - \eta_x \hat{P}_t)E - X_t\|_F^2]$$

$$\le 2\mathbb{E}[\|X_t E - X_t\|_F^2] + 2\eta_x^2 \mathbb{E}[\|\hat{P}_t E\|_F^2]$$

$$= 2\mathbb{E}[\|(X_t - \bar{X}_t)(E - I)\|_F^2] + 2\eta_x^2 \mathbb{E}[\|\hat{P}_t E\|_F^2]$$

$$\le 2\mathbb{E}[\|X_t - \bar{X}_t\|_F^2 \|E - I\|_2^2] + 2\eta_x^2 \mathbb{E}[\|\hat{P}_t\|_F^2 \|E\|_2^2]$$

$$\overset{(a)}{\le} 8\mathbb{E}[\|X_t - \bar{X}_t\|_F^2] + 2\eta_x^2 \mathbb{E}[\|\hat{P}_t\|_F^2]$$

$$\le 8\mathbb{E}[\|X_t - \bar{X}_t\|_F^2] + 2\eta_x^2 \sum_{k=1}^{K}\mathbb{E}[\|\frac{p_t^{(k)}}{\|p_t^{(k)}\|}\|^2]$$

$$\le 8\mathbb{E}[\|X_t - \bar{X}_t\|_F^2] + 2\eta_x^2 K$$

$$\overset{(b)}{\le} \frac{8\eta_x^2 \lambda^2}{(1-\lambda)^2} K + 2\eta_x^2 K \overset{(c)}{\le} \frac{10\eta_x^2}{(1-\lambda)^2} K \, , \tag{49}$$

where $(a)$ holds due to $\|E - I\|_2 \le 2$ and $\|E\|_2 \le 1$, $(b)$ holds due to Lemma C.28, and $(c)$ holds due to $\lambda < 1$.

Then, we obtain

$$\sum_{k=1}^{K} \mathbb{E}[\|x_{t+1}^{(k)} - x_t^{(k)}\||] = \sqrt{\left( \sum_{k=1}^{K} \mathbb{E}[\|x_{t+1}^{(k)} - x_t^{(k)}\||] \right)^2}$$

$$\le \sqrt{K \sum_{k=1}^{K} \mathbb{E}[\|x_{t+1}^{(k)} - x_t^{(k)}\|^2]} = \sqrt{K}\sqrt{\mathbb{E}[\|X_{t+1} - X_t\|_F^2]} \le \frac{4\eta_x}{1-\lambda} K \, . \tag{50}$$

The other two inequalities can be proved in a same approach. $\qquad\square$

**Lemma C.10.** *Given Assumptions 3.1-3.3, for $t > 0$, we obtain*

$$\sum_{k=1}^{K} \mathbb{E}[\|u_{1,t}^{(k)} - u_{1,t-1}^{(k)}\||] \le \gamma_x \sum_{k=1}^{K} \mathbb{E}[\|\nabla_1 f^{(k)}(x_{t-1}^{(k)}, y_{t-1}^{(k)}) - \nabla_1 f^{(k)}(x_{t-1}^{(k)}, y_{t-1}^{(k)}; \xi_t^{(k)})\||]$$

$$+ \gamma_x \sum_{k=1}^{K} \mathbb{E}[\|u_{1,t-1}^{(k)} - \nabla_1 f^{(k)}(x_{t-1}^{(k)}, y_{t-1}^{(k)})\||] + \frac{4\eta_x L_f}{1-\lambda} K + \frac{4\eta_y L_f}{1-\lambda} K \, . \tag{51}$$

*Proof.*

$$\sum_{k=1}^{K} \mathbb{E}[\|u_{1,t}^{(k)} - u_{1,t-1}^{(k)}\||]$$

$$= \sum_{k=1}^{K} \mathbb{E}[\|(1 - \gamma_x)(u_{1,t-1}^{(k)} - \nabla_1 f^{(k)}(x_{t-1}^{(k)}, y_{t-1}^{(k)}; \xi_t^{(k)})) + \nabla_1 f^{(k)}(x_t^{(k)}, y_t^{(k)}; \xi_t^{(k)}) - u_{1,t-1}^{(k)}\||]$$

$$\le \sum_{k=1}^{K} \mathbb{E}[\|\nabla_1 f^{(k)}(x_t^{(k)}, y_t^{(k)}; \xi_t^{(k)}) - \nabla_1 f^{(k)}(x_{t-1}^{(k)}, y_{t-1}^{(k)}; \xi_t^{(k)})\||]$$

$$+ \gamma_x \sum_{k=1}^{K} \mathbb{E}[\|u_{1,t-1}^{(k)} - \nabla_1 f^{(k)}(x_{t-1}^{(k)}, y_{t-1}^{(k)})\||]$$

$$+ \gamma_x \sum_{k=1}^{K} \mathbb{E}[\|\nabla_1 f^{(k)}(x_{t-1}^{(k)}, y_{t-1}^{(k)}) - \nabla_1 f^{(k)}(x_{t-1}^{(k)}, y_{t-1}^{(k)}; \xi_t^{(k)})\||]$$

$$\le L_f \sum_{k=1}^{K} \mathbb{E}[\|x_t^{(k)} - x_{t-1}^{(k)}\||] + L_f \sum_{k=1}^{K} \mathbb{E}[\|y_t^{(k)} - y_{t-1}^{(k)}\||]$$

$$+ \gamma_x \sum_{k=1}^{K} \mathbb{E}[\|u_{1,t-1}^{(k)} - \nabla_1 f^{(k)}(x_{t-1}^{(k)}, y_{t-1}^{(k)})\||]$$

$$+ \gamma_x \sum_{k=1}^{K} \mathbb{E}[\|\nabla_1 f^{(k)}(x_{t-1}^{(k)}, y_{t-1}^{(k)}) - \nabla_1 f^{(k)}(x_{t-1}^{(k)}, y_{t-1}^{(k)}; \xi_t^{(k)})\||]$$

$$\overset{(a)}{\le} \frac{4\eta_x L_f}{1-\lambda} K + \frac{4\eta_y L_f}{1-\lambda} K + \gamma_x \sum_{k=1}^{K} \mathbb{E}[\|u_{1,t-1}^{(k)} - \nabla_1 f^{(k)}(x_{t-1}^{(k)}, y_{t-1}^{(k)})\||]$$

$$+ \gamma_x \sum_{k=1}^{K} \mathbb{E}[\|\nabla_1 f^{(k)}(x_{t-1}^{(k)}, y_{t-1}^{(k)}) - \nabla_1 f^{(k)}(x_{t-1}^{(k)}, y_{t-1}^{(k)}; \xi_t^{(k)})\||] \, , \tag{52}$$

where $(a)$ holds due to Lemma C.9.

$$\square$$

**Lemma C.11.** *Given Assumptions 3.1-3.3, for $t > 0$, we obtain*

$$\sum_{k=1}^{K} \mathbb{E}[\|u_{2,t}^{(k)} - u_{2,t-1}^{(k)}\|] \leq \gamma_x \sum_{k=1}^{K} \mathbb{E}[\|\nabla_1 g^{(k)}(x_{t-1}^{(k)}, y_{t-1}^{(k)}) - \nabla_1 g^{(k)}(x_{t-1}^{(k)}, y_{t-1}^{(k)}; \zeta_t^{(k)})\|]$$

$$+ \gamma_x \sum_{k=1}^{K} \mathbb{E}[\|u_{2,t-1}^{(k)} - \nabla_1 g^{(k)}(x_{t-1}^{(k)}, y_{t-1}^{(k)})\|] + \frac{4\eta_x L_g}{1-\lambda} K + \frac{4\eta_y L_g}{1-\lambda} K . \tag{53}$$

**Lemma C.12.** *Given Assumptions 3.1-3.3, for $t > 0$, we obtain*

$$\sum_{k=1}^{K} \mathbb{E}[\|u_{3,t}^{(k)} - u_{3,t-1}^{(k)}\|] \leq \gamma_x \sum_{k=1}^{K} \mathbb{E}[\|\nabla_1 g^{(k)}(x_{t-1}^{(k)}, z_{t-1}^{(k)}) - \nabla_1 g^{(k)}(x_{t-1}^{(k)}, z_{t-1}^{(k)}; \zeta_t^{(k)})\|]$$

$$+ \gamma_x \sum_{k=1}^{K} \mathbb{E}[\|u_{3,t-1}^{(k)} - \nabla_1 g^{(k)}(x_{t-1}^{(k)}, z_{t-1}^{(k)})\|] + \frac{4\eta_x L_g}{1-\lambda} K + \frac{4\eta_z L_g}{1-\lambda} K . \tag{54}$$

**Lemma C.13.** *Given Assumptions 3.1-3.3, for $t > 0$, we obtain*

$$\sum_{k=1}^{K} \mathbb{E}[\|v_{1,t}^{(k)} - v_{1,t-1}^{(k)}\|] \leq \gamma_y \sum_{k=1}^{K} \mathbb{E}[\|\nabla_2 f^{(k)}(x_{t-1}^{(k)}, y_{t-1}^{(k)}) - \nabla_2 f^{(k)}(x_{t-1}^{(k)}, y_{t-1}^{(k)}; \xi_t^{(k)})\|]$$

$$+ \gamma_y \sum_{k=1}^{K} \mathbb{E}[\|u_{1,t-1}^{(k)} - \nabla_1 f^{(k)}(x_{t-1}^{(k)}, y_{t-1}^{(k)})\|] + \frac{4\eta_x L_f}{1-\lambda} K + \frac{4\eta_y L_f}{1-\lambda} K . \tag{55}$$

**Lemma C.14.** *Given Assumptions 3.1-3.3, for $t > 0$, we obtain*

$$\sum_{k=1}^{K} \mathbb{E}[\|v_{2,t}^{(k)} - v_{2,t-1}^{(k)}\|] \leq \gamma_y \sum_{k=1}^{K} \mathbb{E}[\|\nabla_2 g^{(k)}(x_{t-1}^{(k)}, y_{t-1}^{(k)}) - \nabla_2 g^{(k)}(x_{t-1}^{(k)}, y_{t-1}^{(k)}; \zeta_t^{(k)})\|]$$

$$+ \gamma_y \sum_{k=1}^{K} \mathbb{E}[\|u_{2,t-1}^{(k)} - \nabla_1 g^{(k)}(x_{t-1}^{(k)}, y_{t-1}^{(k)})\|] + \frac{4\eta_x L_g}{1-\lambda} K + \frac{4\eta_y L_g}{1-\lambda} K . \tag{56}$$

**Lemma C.15.** *Given Assumptions 3.1-3.3, for $t > 0$, we obtain*

$$\sum_{k=1}^{K} \mathbb{E}[\|w_{1,t}^{(k)} - w_{1,t-1}^{(k)}\|] \leq \gamma_z \sum_{k=1}^{K} \mathbb{E}[\|\nabla_2 g^{(k)}(x_{t-1}^{(k)}, z_{t-1}^{(k)}) - \nabla_2 g^{(k)}(x_{t-1}^{(k)}, z_{t-1}^{(k)}; \zeta_t^{(k)})\|]$$

$$+ \gamma_z \sum_{k=1}^{K} \mathbb{E}[\|w_{1,t-1}^{(k)} - \nabla_2 g^{(k)}(x_{t-1}^{(k)}, z_{t-1}^{(k)})\|] + \frac{4\eta_x L_g}{1-\lambda} K + \frac{4\eta_z L_g}{1-\lambda} K . \tag{57}$$

Lemmas C.11 - C.15 can be easily proved by following Lemma C.10.

## C.4 BOUNDING GRADIENT ERRORS

**Lemma C.16.** *Given Assumptions 3.1-3.3, we obtain*

$$\sum_{k=1}^{K} \mathbb{E}[\|u_{1,t}^{(k)} - \nabla_1 f^{(k)}(x_t^{(k)}, y_t^{(k)})\|] \leq (1-\gamma_x)^t \frac{2\sqrt{2}\sigma K}{B_0^{1-1/s}} + \frac{4(\eta_x + \eta_y)L_f}{(1-\lambda)\sqrt{\gamma_x}}\sqrt{K} + 2\sqrt{2}\gamma_x^{1-1/s}\sigma K . \tag{58}$$

*Proof.* When $t > 0$, based on Algorithm 1, we obtain

$$u_{1,t}^{(k)} - \nabla_1 f^{(k)}(x_t^{(k)}, y_t^{(k)})$$

$$= (1 - \gamma_x)(u_{1,t-1}^{(k)} - \nabla_1 f^{(k)}(x_{t-1}^{(k)}, y_{t-1}^{(k)})) + (1 - \gamma_x)\Big(\nabla_1 f^{(k)}(x_t^{(k)}, y_t^{(k)}; \xi_t^{(k)})$$

$$- \nabla_1 f^{(k)}(x_{t-1}^{(k)}, y_{t-1}^{(k)}; \xi_t^{(k)}) - \nabla_1 f^{(k)}(x_t^{(k)}, y_t^{(k)}) + \nabla_1 f^{(k)}(x_{t-1}^{(k)}, y_{t-1}^{(k)})\Big)$$

$$+ \gamma_x(\nabla_1 f^{(k)}(x_t^{(k)}, y_t^{(k)}; \xi_t^{(k)}) - \nabla_1 f^{(k)}(x_t^{(k)}, y_t^{(k)}))$$

$$= (1 - \gamma_x)^t (u_{1,0}^{(k)} - \nabla_1 f^{(k)}(x_0^{(k)}, y_0^{(k)})) + \sum_{j=1}^{t}(1 - \gamma_x)^{t-j+1}\Big(\nabla_1 f^{(k)}(x_j^{(k)}, y_j^{(k)}; \xi_j^{(k)})$$

$$- \nabla_1 f^{(k)}(x_{j-1}^{(k)}, y_{j-1}^{(k)}; \xi_j^{(k)}) + \nabla_1 f^{(k)}(x_{j-1}^{(k)}, y_{j-1}^{(k)}) - \nabla_1 f^{(k)}(x_j^{(k)}, y_j^{(k)})\Big)$$

$$+ \sum_{j=1}^{t} \gamma_x (1 - \gamma_x)^{t-j}(\nabla_1 f^{(k)}(x_j^{(k)}, y_j^{(k)}; \xi_j^{(k)}) - \nabla_1 f^{(k)}(x_j^{(k)}, y_j^{(k)})) . \tag{59}$$

Then, we obtain

$$\sum_{k=1}^{K} \mathbb{E}[\|u_{1,t}^{(k)} - \nabla_1 f^{(k)}(x_t^{(k)}, y_t^{(k)})\|]$$

$$\le (1 - \gamma_x)^t \sum_{k=1}^{K} \mathbb{E}[\|u_{1,0}^{(k)} - \nabla_1 f^{(k)}(x_0^{(k)}, y_0^{(k)})\|]$$

$$+ \sum_{k=1}^{K} \mathbb{E}[\| \sum_{j=1}^{t}(1 - \gamma_x)^{t-j+1}\Big(\nabla_1 f^{(k)}(x_j^{(k)}, y_j^{(k)}; \xi_j^{(k)}) - \nabla_1 f^{(k)}(x_{j-1}^{(k)}, y_{j-1}^{(k)}; \xi_j^{(k)})$$

$$+ \nabla_1 f^{(k)}(x_{j-1}^{(k)}, y_{j-1}^{(k)}) - \nabla_1 f^{(k)}(x_j^{(k)}, y_j^{(k)})\Big)\|]$$

$$+ \sum_{k=1}^{K} \mathbb{E}[\|\gamma_x \sum_{j=1}^{t}(1 - \gamma_x)^{t-j}(\nabla_1 f^{(k)}(x_j^{(k)}, y_j^{(k)}; \xi_j^{(k)}) - \nabla_1 f^{(k)}(x_j^{(k)}, y_j^{(k)}))\|] . \tag{60}$$

For the first term on the right-hand side of Eq. (60), we bound it as follows:

$$\sum_{k=1}^{K} \mathbb{E}[\|u_{1,0}^{(k)} - \nabla_1 f^{(k)}(x_0^{(k)}, y_0^{(k)})\|]$$

$$= \sum_{k=1}^{K} \mathbb{E}[\| \frac{1}{B_0} \sum_{b=1}^{B_0} \nabla_1 f^{(k)}(x_0^{(k)}, y_0^{(k)}; \xi_{0,b}^{(k)}) - \nabla_1 f^{(k)}(x_0^{(k)}, y_0^{(k)})\|]$$

$$= \sum_{k=1}^{K} \frac{1}{B_0} \mathbb{E}[\| \sum_{b=1}^{B_0} (\nabla_1 f^{(k)}(x_0^{(k)}, y_0^{(k)}; \xi_{0,b}^{(k)}) - \nabla_1 f^{(k)}(x_0^{(k)}, y_0^{(k)}))\|]$$

$$\overset{(a)}{\le} \sum_{k=1}^{K} \frac{2\sqrt{2}}{B_0} \mathbb{E}[(\sum_{b=1}^{B_0} \|\nabla_1 f^{(k)}(x_0^{(k)}, y_0^{(k)}; \xi_0^{(k)}) - \nabla_1 f^{(k)}(x_0^{(k)}, y_0^{(k)})\|^s)^{\frac{1}{s}}]$$

$$\overset{(b)}{\le} \sum_{k=1}^{K} \frac{2\sqrt{2}}{B_0} (\sum_{b=1}^{B_0} \mathbb{E}[\|\nabla_1 f^{(k)}(x_0^{(k)}, y_0^{(k)}; \xi_0^{(k)}) - \nabla_1 f^{(k)}(x_0^{(k)}, y_0^{(k)})\|^s])^{\frac{1}{s}}$$

$$\overset{(c)}{\le} \frac{2\sqrt{2}K}{B_0^{1-1/s}} \sigma , \tag{61}$$

where $B_0$ represents the batch size in the initial iteration, $(a)$ holds due to Lemma C.5, $(b)$ holds due to Hölder's inequality, and $(c)$ holds due to Assumption 3.2.

To bound the second term on the right-hand side of Eq. (60), we first bound the following one:

$$\sum_{k=1}^{K} \mathbb{E}[\| \sum_{j=1}^{t}(1 - \gamma_x)^{t-j+1}\Big(\nabla_1 f^{(k)}(x_j^{(k)}, y_j^{(k)}; \xi_j^{(k)}) - \nabla_1 f^{(k)}(x_{j-1}^{(k)}, y_{j-1}^{(k)}; \xi_j^{(k)})$$

$$+ \nabla_1 f^{(k)}(x_{j-1}^{(k)}, y_{j-1}^{(k)}) - \nabla_1 f^{(k)}(x_j^{(k)}, y_j^{(k)}) \Big) \|^2]$$

$$= \sum_{k=1}^{K} \sum_{j=1}^{t} (1-\gamma_x)^{2(t-j+1)} \mathbb{E}[\| \nabla_1 f^{(k)}(x_j^{(k)}, y_j^{(k)}; \xi_j^{(k)}) - \nabla_1 f^{(k)}(x_{j-1}^{(k)}, y_{j-1}^{(k)}; \xi_j^{(k)})$$

$$+ \nabla_1 f^{(k)}(x_{j-1}^{(k)}, y_{j-1}^{(k)}) - \nabla_1 f^{(k)}(x_j^{(k)}, y_j^{(k)}) \|^2]$$

$$\leq \sum_{k=1}^{K} \sum_{j=1}^{t} (1-\gamma_x)^{2(t-j+1)} \mathbb{E}[\| \nabla_1 f^{(k)}(x_j^{(k)}, y_j^{(k)}; \xi_j^{(k)}) - \nabla_1 f^{(k)}(x_{j-1}^{(k)}, y_{j-1}^{(k)}; \xi_j^{(k)}) \|^2]$$

$$\leq \sum_{k=1}^{K} \sum_{j=1}^{t} (1-\gamma_x)^{2(t-j+1)} L_f^2 (\mathbb{E}[\|x_j^{(k)} - x_{j-1}^{(k)}\|^2] + \mathbb{E}[\|y_j^{(k)} - y_{j-1}^{(k)}\|^2])$$

$$= \sum_{j=1}^{t} (1-\gamma_x)^{2(t-j+1)} L_f^2 (\mathbb{E}[\|X_j - X_{j-1}\|_F^2] + \mathbb{E}[\|Y_j - Y_{j-1}\|_F^2])$$

$$\overset{(a)}{\leq} \sum_{j=1}^{t} (1-\gamma_x)^{2(t-j+1)} L_f^2 \left( \frac{10\eta_x^2}{(1-\lambda)^2} K + \frac{10\eta_y^2}{(1-\lambda)^2} K \right)$$

$$\leq \frac{1}{1-(1-\gamma_x)^2} L_f^2 \left( \frac{10\eta_x^2}{(1-\lambda)^2} K + \frac{10\eta_y^2}{(1-\lambda)^2} K \right)$$

$$\overset{(b)}{\leq} \frac{10\eta_x^2}{(1-\lambda)^2} \frac{L_f^2}{\gamma_x} K + \frac{10\eta_y^2}{(1-\lambda)^2} \frac{L_f^2}{\gamma_x} K , \tag{62}$$

where $(a)$ holds due to Eq. (49), and $(b)$ holds due to $\gamma_x < 1$.

Then, we bound the second term on the right-hand side of Eq. (60) as follows:

$$\sum_{k=1}^{K} \mathbb{E}[\| \sum_{j=1}^{t} (1-\gamma_x)^{t-j+1} \Big( \nabla_1 f^{(k)}(x_j^{(k)}, y_j^{(k)}; \xi_j^{(k)}) - \nabla_1 f^{(k)}(x_{j-1}^{(k)}, y_{j-1}^{(k)}; \xi_j^{(k)})$$

$$+ \nabla_1 f^{(k)}(x_{j-1}^{(k)}, y_{j-1}^{(k)}) - \nabla_1 f^{(k)}(x_j^{(k)}, y_j^{(k)}) \Big) \|]$$

$$\leq \frac{4\eta_x L_f}{(1-\lambda)\sqrt{\gamma_x}} \sqrt{K} + \frac{4\eta_y L_f}{(1-\lambda)\sqrt{\gamma_x}} \sqrt{K} . \tag{63}$$

For the third term on the right-hand side of Eq. (60), we bound it as follows:

$$\mathbb{E}[\| \sum_{j=1}^{t} \gamma_x (1-\gamma_x)^{t-j} (\nabla_1 f^{(k)}(x_j^{(k)}, y_j^{(k)}; \xi_j^{(k)}) - \nabla_1 f^{(k)}(x_j^{(k)}, y_j^{(k)})) \|]$$

$$\overset{(a)}{\leq} 2\sqrt{2} \mathbb{E}\left[ \left( \sum_{j=1}^{t} \| \gamma_x (1-\gamma_x)^{t-j} (\nabla_1 f^{(k)}(x_j^{(k)}, y_j^{(k)}; \xi_j^{(k)}) - \nabla_1 f^{(k)}(x_j^{(k)}, y_j^{(k)})) \|^s \right)^{1/s} \right]$$

$$= 2\sqrt{2} \mathbb{E}\left[ \left( \sum_{j=1}^{t} \gamma_x^s (1-\gamma_x)^{s(t-j)} \| \nabla_1 f^{(k)}(x_j^{(k)}, y_j^{(k)}; \xi_j^{(k)}) - \nabla_1 f^{(k)}(x_j^{(k)}, y_j^{(k)}) \|^s \right)^{1/s} \right]$$

$$\overset{(b)}{\leq} 2\sqrt{2} \left( \mathbb{E}\left[ \sum_{j=1}^{t} \gamma_x^s (1-\gamma_x)^{s(t-j)} \| \nabla_1 f^{(k)}(x_j^{(k)}, y_j^{(k)}; \xi_j^{(k)}) - \nabla_1 f^{(k)}(x_j^{(k)}, y_j^{(k)}) \|^s \right] \right)^{1/s}$$

$$\overset{(c)}{\leq} 2\sqrt{2} \left( \sum_{j=1}^{t} \gamma_x^s (1-\gamma_x)^{s(t-j)} \right)^{1/s} \sigma , \tag{64}$$

where $(a)$ holds due to Lemma C.5, $(b)$ holds due to Hölder's inequality, and $(c)$ holds due to Assumption 3.2.

Finally, when $t > 0$, from

$$\left(\sum_{j=1}^{t}(1-\gamma_x)^{s(t-j)}\right)^{1/s} \leq \left(\frac{1}{1-(1-\gamma_x)^s}\right)^{1/s} \leq \left(\frac{1}{1-(1-\gamma_x)}\right)^{1/s} \leq \gamma_x^{-1/s} , \qquad (65)$$

we obtain

$$\sum_{k=1}^{K}\mathbb{E}[\|u_{1,t}^{(k)} - \nabla_1 f^{(k)}(x_t^{(k)}, y_t^{(k)})\|]$$

$$\leq (1-\gamma_x)^t \frac{2\sqrt{2}K}{B_0^{1-1/s}}\sigma + \frac{4\eta_x L_f}{(1-\lambda)\sqrt{\gamma_x}}\sqrt{K} + \frac{4\eta_y L_f}{(1-\lambda)\sqrt{\gamma_x}}\sqrt{K} + 2\sqrt{2}\gamma_x^{1-1/s}\sigma K . \qquad (66)$$

Based on Eq. (61), it is easy to know that this upper bound also holds when $t = 0$. $\qquad\square$

**Lemma C.17.** *Given Assumptions 3.1-3.3, we obtain*

$$\sum_{k=1}^{K}\mathbb{E}[\|u_{2,t}^{(k)} - \nabla_1 g^{(k)}(x_t^{(k)}, y_t^{(k)})\|] \leq (1-\gamma_x)^t \frac{2\sqrt{2}\sigma K}{B_0^{1-1/s}} + \frac{4(\eta_x+\eta_y)L_g}{(1-\lambda)\sqrt{\gamma_x}}\sqrt{K} + 2\sqrt{2}\gamma_x^{1-1/s}\sigma K . \qquad (67)$$

**Lemma C.18.** *Given Assumptions 3.1-3.3, we obtain*

$$\sum_{k=1}^{K}\mathbb{E}[\|u_{3,t}^{(k)} - \nabla_1 g^{(k)}(x_t^{(k)}, z_t^{(k)})\|] \leq (1-\gamma_x)^t \frac{2\sqrt{2}\eta_y}{B_0^{1-1/s}}x + \frac{4(\eta_x+\eta_z)L_g}{(1-\lambda)\sqrt{\gamma_x}}\sqrt{K} + 2\sqrt{2}\gamma_x^{1-1/s}\sigma K . \qquad (68)$$

**Lemma C.19.** *Given Assumptions 3.1-3.3, we obtain*

$$\sum_{k=1}^{K}\mathbb{E}[\|v_{1,t}^{(k)} - \nabla_2 f^{(k)}(x_t^{(k)}, y_t^{(k)})\|] \leq (1-\gamma_y)^t \frac{2\sqrt{2}\sigma K}{B_0^{1-1/s}} + \frac{4(\eta_x+\eta_y)L_f}{(1-\lambda)\sqrt{\gamma_y}}\sqrt{K} + 2\sqrt{2}\gamma_y^{1-1/s}\sigma K . \qquad (69)$$

**Lemma C.20.** *Given Assumptions 3.1-3.3, we obtain*

$$\sum_{k=1}^{K}\mathbb{E}[\|v_{2,t}^{(k)} - \nabla_2 g^{(k)}(x_t^{(k)}, y_t^{(k)})\|] \leq (1-\gamma_y)^t \frac{2\sqrt{2}\sigma K}{B_0^{1-1/s}} + \frac{4(\eta_x+\eta_y)L_g}{(1-\lambda)\sqrt{\gamma_y}}\sqrt{K} + 2\sqrt{2}\gamma_y^{1-1/s}\sigma K . \qquad (70)$$

**Lemma C.21.** *Given Assumptions 3.1-3.3, we obtain*

$$\sum_{k=1}^{K}\mathbb{E}[\|w_{1,t}^{(k)} - \nabla_2 g^{(k)}(x_t^{(k)}, z_t^{(k)})\|] \leq (1-\gamma_z)^t \frac{2\sqrt{2}\sigma K}{B_0^{1-1/s}} + \frac{4(\eta_x+\eta_z)L_g}{(1-\lambda)\sqrt{\gamma_z}}\sqrt{K} + 2\sqrt{2}\gamma_z^{1-1/s}\sigma K . \qquad (71)$$

Lemmas C.17 - C.21 can be easily proved by following Lemma C.16.

**Lemma C.22.** *Given Assumptions 3.1-3.3, we obtain*

$$\frac{1}{T}\sum_{t=0}^{T-1}\mathbb{E}[\|\frac{1}{K}\sum_{k=1}^{K}u_{1,t}^{(k)} - \frac{1}{K}\sum_{k=1}^{K}\nabla_1 f^{(k)}(x_t^{(k)}, y_t^{(k)})\|]$$

$$\leq \frac{1}{\gamma_x T}\frac{2\sqrt{2}\sigma}{B_0^{1-1/s}} + \frac{4(\eta_x+\eta_y)L_f}{(1-\lambda)\sqrt{\gamma_x}}\frac{1}{\sqrt{K}} + \frac{2\sqrt{2}\gamma_x^{1-1/s}\sigma}{K^{1-1/s}} . \qquad (72)$$

*Proof.* When $t > 0$, same as the proof of Lemma C.16, we obtain

$$\mathbb{E}[\|\frac{1}{K}\sum_{k=1}^{K}(u_{1,t}^{(k)} - \nabla_1 f^{(k)}(x_t^{(k)}, y_t^{(k)}))\|]$$

$$\leq (1-\gamma_x)^t \mathbb{E}[\|\frac{1}{K}\sum_{k=1}^{K}(u_{1,0}^{(k)} - \nabla_1 f^{(k)}(x_0^{(k)}, y_0^{(k)}))\|]$$

$$+ \mathbb{E}[\|\sum_{j=1}^{t}(1-\gamma_x)^{t-j+1}\frac{1}{K}\sum_{k=1}^{K}\left(\nabla_1 f^{(k)}(x_j^{(k)}, y_j^{(k)}; \xi_j^{(k)}) - \nabla_1 f^{(k)}(x_{j-1}^{(k)}, y_{j-1}^{(k)}; \xi_j^{(k)})\right.$$

$$\left. + \nabla_1 f^{(k)}(x_{j-1}^{(k)}, y_{j-1}^{(k)}) - \nabla_1 f^{(k)}(x_j^{(k)}, y_j^{(k)})\right)\|]$$

$$+ \mathbb{E}[\|\gamma_x\sum_{j=1}^{t}(1-\gamma_x)^{t-j}\frac{1}{K}\sum_{k=1}^{K}(\nabla_1 f^{(k)}(x_j^{(k)}, y_j^{(k)}; \xi_j^{(k)}) - \nabla_1 f^{(k)}(x_j^{(k)}, y_j^{(k)}))\|] . \quad (73)$$

Then, for the first term on the right-hand side of Eq. (73), we obtain

$$\mathbb{E}[\|\frac{1}{K}\sum_{k=1}^{K}(u_{1,0}^{(k)} - \nabla_1 f^{(k)}(x_0^{(k)}, y_0^{(k)}))\|] \leq \frac{1}{K}\sum_{k=1}^{K}\mathbb{E}[\|u_{1,0}^{(k)} - \nabla_1 f^{(k)}(x_0^{(k)}, y_0^{(k)})\|] \overset{(a)}{\leq} \frac{2\sqrt{2}}{B_0^{1-1/s}}\sigma , \quad (74)$$

where $(a)$ holds due to Eq. (61).

To bound the second term on the right-hand side of Eq. (73), we first bound the following one:

$$\mathbb{E}[\|\sum_{j=1}^{t}(1-\gamma_x)^{t-j+1}\frac{1}{K}\sum_{k=1}^{K}(\nabla_1 f^{(k)}(x_j^{(k)}, y_j^{(k)}; \xi_j^{(k)}) - \nabla_1 f^{(k)}(x_{j-1}^{(k)}, y_{j-1}^{(k)}; \xi_j^{(k)})$$

$$+ \nabla_1 f^{(k)}(x_{j-1}^{(k)}, y_{j-1}^{(k)}) - \nabla_1 f^{(k)}(x_j^{(k)}, y_j^{(k)}))\|^2]$$

$$= \frac{1}{K^2}\sum_{k=1}^{K}\sum_{j=1}^{t}(1-\gamma_x)^{2(t-j+1)}\mathbb{E}[\|\nabla_1 f^{(k)}(x_j^{(k)}, y_j^{(k)}; \xi_j^{(k)}) - \nabla_1 f^{(k)}(x_{j-1}^{(k)}, y_{j-1}^{(k)}; \xi_j^{(k)})$$

$$+ \nabla_1 f^{(k)}(x_{j-1}^{(k)}, y_{j-1}^{(k)}) - \nabla_1 f^{(k)}(x_j^{(k)}, y_j^{(k)})\|^2]$$

$$\leq \frac{1}{K^2}\sum_{k=1}^{K}\sum_{j=1}^{t}(1-\gamma_x)^{2(t-j+1)}\mathbb{E}[\|\nabla_1 f^{(k)}(x_j^{(k)}, y_j^{(k)}; \xi_j^{(k)}) - \nabla_1 f^{(k)}(x_{j-1}^{(k)}, y_{j-1}^{(k)}; \xi_j^{(k)})\|^2]$$

$$\leq \frac{1}{K^2}\sum_{k=1}^{K}\sum_{j=1}^{t}(1-\gamma_x)^{2(t-j+1)}L_f^2(\mathbb{E}[\|x_j^{(k)} - x_{j-1}^{(k)}\|^2] + \mathbb{E}[\|y_j^{(k)} - y_{j-1}^{(k)}\|^2])$$

$$= \frac{1}{K^2}\sum_{j=1}^{t}(1-\gamma_x)^{2(t-j+1)}L_f^2(\mathbb{E}[\|X_j - X_{j-1}\|_F^2] + \mathbb{E}[\|Y_j - Y_{j-1}\|_F^2])$$

$$\overset{(a)}{\leq} \frac{1}{K^2}\sum_{j=1}^{t}(1-\gamma_x)^{2(t-j+1)}L_f^2\left(\frac{10\eta_x^2}{(1-\lambda)^2}K + \frac{10\eta_y^2}{(1-\lambda)^2}K\right)$$

$$\leq \frac{1}{1-(1-\gamma_x)^2}L_f^2\left(\frac{10\eta_x^2}{(1-\lambda)^2}\frac{1}{K} + \frac{10\eta_y^2}{(1-\lambda)^2}\frac{1}{K}\right)$$

$$\overset{(b)}{\leq} \frac{10\eta_x^2}{(1-\lambda)^2}\frac{L_f^2}{\gamma_x}\frac{1}{K} + \frac{10\eta_y^2}{(1-\lambda)^2}\frac{L_f^2}{\gamma_x}\frac{1}{K} , \quad (75)$$

where $(a)$ holds due to Eq. (49), and $(b)$ holds due to $\gamma_x < 1$.

Then, we bound the second term on the right-hand side of Eq. (60) as follows:

$$\mathbb{E}[\|\sum_{j=1}^{t}(1-\gamma_x)^{t-j+1}\frac{1}{K}\sum_{k=1}^{K}\left(\nabla_1 f^{(k)}(x_j^{(k)}, y_j^{(k)}; \xi_j^{(k)}) - \nabla_1 f^{(k)}(x_{j-1}^{(k)}, y_{j-1}^{(k)}; \xi_j^{(k)})\right.$$

$$+ \nabla_1 f^{(k)}(x_{j-1}^{(k)}, y_{j-1}^{(k)}) - \nabla_1 f^{(k)}(x_j^{(k)}, y_j^{(k)})\Big)\|]$$

$$\leq \frac{4\eta_x L_f}{(1-\lambda)\sqrt{\gamma_x}} \frac{1}{\sqrt{K}} + \frac{4\eta_y L_f}{(1-\lambda)\sqrt{\gamma_x}} \frac{1}{\sqrt{K}} \ . \tag{76}$$

For the third term on the right-hand side of Eq. (73), we bound it as follows:

$$\mathbb{E}[\| \sum_{j=1}^{t} \gamma_x(1-\gamma_x)^{t-j} \frac{1}{K} \sum_{k=1}^{K} (\nabla_1 f^{(k)}(x_j^{(k)}, y_j^{(k)}; \xi_j^{(k)}) - \nabla_1 f^{(k)}(x_j^{(k)}, y_j^{(k)}))\|]$$

$$= \frac{1}{K}\mathbb{E}[\| \sum_{j=1}^{t} \gamma_x(1-\gamma_x)^{t-j} \sum_{k=1}^{K} (\nabla_1 f^{(k)}(x_j^{(k)}, y_j^{(k)}; \xi_j^{(k)}) - \nabla_1 f^{(k)}(x_j^{(k)}, y_j^{(k)}))\|]$$

$$\overset{(a)}{\leq} \frac{2\sqrt{2}}{K}\mathbb{E}\left[\left(\sum_{k=1}^{K}\sum_{j=1}^{t}\|\gamma_x(1-\gamma_x)^{t-j}(\nabla_1 f^{(k)}(x_j^{(k)}, y_j^{(k)}; \xi_j^{(k)}) - \nabla_1 f^{(k)}(x_j^{(k)}, y_j^{(k)}))\|^s\right)^{1/s}\right]$$

$$= \frac{2\sqrt{2}}{K}\mathbb{E}\left[\left(\sum_{k=1}^{K}\sum_{j=1}^{t}\gamma_x^s(1-\gamma_x)^{s(t-j)}\|\nabla_1 f^{(k)}(x_j^{(k)}, y_j^{(k)}; \xi_j^{(k)}) - \nabla_1 f^{(k)}(x_j^{(k)}, y_j^{(k)})\|^s\right)^{1/s}\right]$$

$$\overset{(b)}{\leq} \frac{2\sqrt{2}}{K}\left(\mathbb{E}\left[\sum_{k=1}^{K}\sum_{j=1}^{t}\gamma_x^s(1-\gamma_x)^{s(t-j)}\|\nabla_1 f^{(k)}(x_j^{(k)}, y_j^{(k)}; \xi_j^{(k)}) - \nabla_1 f^{(k)}(x_j^{(k)}, y_j^{(k)})\|^s\right]\right)^{1/s}$$

$$\overset{(c)}{\leq} \frac{2\sqrt{2}}{K}\left(\sum_{k=1}^{K}\sum_{j=1}^{t}\gamma_x^s(1-\gamma_x)^{s(t-j)}\right)^{1/s}\sigma \ , \tag{77}$$

where $(a)$ holds due to Lemma C.5, $(b)$ holds due to Hölder's inequality, and $(c)$ holds due to Assumption 3.2.

Finally, when $t > 0$, from

$$\left(\sum_{k=1}^{K}\sum_{j=1}^{t}\gamma_x^s(1-\gamma_x)^{s(t-j)}\right)^{1/s} \leq \left(\frac{K}{1-(1-\gamma_x)^s}\right)^{1/s} \leq \left(\frac{K}{1-(1-\gamma_x)^s}\right)^{1/s} \leq \frac{\gamma_x^{-1/s}}{K^{-1/s}} \ , \tag{78}$$

we obtain

$$\mathbb{E}[\| \frac{1}{K}\sum_{k=1}^{K} u_{1,t}^{(k)} - \frac{1}{K}\sum_{k=1}^{K} \nabla_1 f^{(k)}(x_t^{(k)}, y_t^{(k)})\|]$$

$$\leq (1-\gamma_x)^t \frac{2\sqrt{2}}{B_0^{1-1/s}}\sigma + \frac{4\eta_x L_f}{(1-\lambda)\sqrt{\gamma_x}} \frac{1}{\sqrt{K}} + \frac{4\eta_y L_f}{(1-\lambda)\sqrt{\gamma_x}} \frac{1}{\sqrt{K}} + \frac{2\sqrt{2}}{K^{1-1/s}}\gamma_x^{1-1/s}\sigma \ . \tag{79}$$

Similarly, it is easy to know that this upper bound also holds when $t = 0$. Then, we obtain

$$\frac{1}{T}\sum_{t=0}^{T-1}\mathbb{E}[\| \frac{1}{K}\sum_{k=1}^{K} u_{1,t}^{(k)} - \frac{1}{K}\sum_{k=1}^{K} \nabla_1 f^{(k)}(x_t^{(k)}, y_t^{(k)})\|]$$

$$\leq \frac{1}{T}\sum_{t=0}^{T-1}(1-\gamma_x)^t \frac{2\sqrt{2}}{B_0^{1-1/s}}\sigma + \frac{4\eta_x L_f}{(1-\lambda)\sqrt{\gamma_x}} \frac{1}{\sqrt{K}} + \frac{4\eta_y L_f}{(1-\lambda)\sqrt{\gamma_x}} \frac{1}{\sqrt{K}} + \frac{2\sqrt{2}}{K^{1-1/s}}\gamma_x^{1-1/s}\sigma$$

$$\leq \frac{1}{\gamma_x T} \frac{2\sqrt{2}}{B_0^{1-1/s}}\sigma + \frac{4\eta_x L_f}{(1-\lambda)\sqrt{\gamma_x}} \frac{1}{\sqrt{K}} + \frac{4\eta_y L_f}{(1-\lambda)\sqrt{\gamma_x}} \frac{1}{\sqrt{K}} + \frac{2\sqrt{2}}{K^{1-1/s}}\gamma_x^{1-1/s}\sigma \ . \tag{80}$$

$$\square$$

**Lemma C.23.** *Given Assumptions 3.1-3.3, we obtain*

$$\frac{1}{T}\sum_{t=0}^{T-1}\mathbb{E}[\|\frac{1}{K}\sum_{k=1}^{K}u_{2,t}^{(k)} - \frac{1}{K}\sum_{k=1}^{K}\nabla_1 g^{(k)}(x_t^{(k)}, y_t^{(k)})\|]$$

$$\leq \frac{1}{\gamma_x T}\frac{2\sqrt{2}\sigma}{B_0^{1-1/s}} + \frac{4(\eta_x+\eta_y)L_g}{(1-\lambda)\sqrt{\gamma_x}}\frac{1}{\sqrt{K}} + \frac{2\sqrt{2}\gamma_x^{1-1/s}\sigma}{K^{1-1/s}} \ . \tag{81}$$

**Lemma C.24.** *Given Assumptions 3.1-3.3, we obtain*

$$\frac{1}{T}\sum_{t=0}^{T-1}\mathbb{E}[\|\frac{1}{K}\sum_{k=1}^{K}u_{3,t}^{(k)} - \frac{1}{K}\sum_{k=1}^{K}\nabla_1 g^{(k)}(x_t^{(k)}, z_t^{(k)})\|]$$

$$\leq \frac{1}{\gamma_x T}\frac{2\sqrt{2}\sigma}{B_0^{1-1/s}} + \frac{4(\eta_x+\eta_z)L_g}{(1-\lambda)\sqrt{\gamma_x}}\frac{1}{\sqrt{K}} + \frac{2\sqrt{2}\gamma_x^{1-1/s}\sigma}{K^{1-1/s}} \ . \tag{82}$$

**Lemma C.25.** *Given Assumptions 3.1-3.3, we obtain*

$$\frac{1}{T}\sum_{t=0}^{T-1}\mathbb{E}[\|\frac{1}{K}\sum_{k=1}^{K}v_{1,t}^{(k)} - \frac{1}{K}\sum_{k=1}^{K}\nabla_2 f^{(k)}(x_t^{(k)}, y_t^{(k)})\|]$$

$$\leq \frac{1}{\gamma_y T}\frac{2\sqrt{2}}{B_0^{1-1/s}}\sigma + \frac{4(\eta_x+\eta_y)L_f}{(1-\lambda)\sqrt{\gamma_y}}\frac{1}{\sqrt{K}} + \frac{2\sqrt{2}\gamma_y^{1-1/s}\sigma}{K^{1-1/s}} \ . \tag{83}$$

**Lemma C.26.** *Given Assumptions 3.1-3.3, we obtain*

$$\frac{1}{T}\sum_{t=0}^{T-1}\mathbb{E}[\|\frac{1}{K}\sum_{k=1}^{K}v_{2,t}^{(k)} - \frac{1}{K}\sum_{k=1}^{K}\nabla_2 g^{(k)}(x_t^{(k)}, y_t^{(k)})\|]$$

$$\leq \frac{1}{\gamma_y T}\frac{2\sqrt{2}}{B_0^{1-1/s}}\sigma + \frac{4(\eta_x+\eta_y)L_g}{(1-\lambda)\sqrt{\gamma_y}}\frac{1}{\sqrt{K}} + \frac{2\sqrt{2}\gamma_y^{1-1/s}\sigma}{K^{1-1/s}} \ . \tag{84}$$

**Lemma C.27.** *Given Assumptions 3.1-3.3, we obtain*

$$\frac{1}{T}\sum_{t=0}^{T-1}\mathbb{E}[\|\frac{1}{K}\sum_{k=1}^{K}w_{1,t}^{(k)} - \frac{1}{K}\sum_{k=1}^{K}\nabla_2 g^{(k)}(x_t^{(k)}, z_t^{(k)})\|]$$

$$\leq \frac{1}{\gamma_z T}\frac{2\sqrt{2}\sigma}{B_0^{1-1/s}} + \frac{4(\eta_x+\eta_z)L_g}{(1-\lambda)\sqrt{\gamma_z}}\frac{1}{\sqrt{K}} + \frac{2\sqrt{2}\gamma_z^{1-1/s}\sigma}{K^{1-1/s}} \ . \tag{85}$$

## C.5 BOUNDING CONSENSUS ERRORS

**Lemma C.28.** *Given Assumptions 3.1-3.3, we obtain*

$$\frac{1}{K}\sum_{k=1}^{K}\mathbb{E}[\|x_t^{(k)} - \bar{x}_t\|] \leq \frac{\eta_x\lambda}{1-\lambda} \ ; \quad \frac{1}{K}\sum_{k=1}^{K}\mathbb{E}[\|y_t^{(k)} - \bar{y}_t\|] \leq \frac{\eta_y\lambda}{1-\lambda} \ ;$$

$$\frac{1}{K}\sum_{k=1}^{K}\mathbb{E}[\|z_t^{(k)} - \bar{z}_t\|] \leq \frac{\eta_z\lambda}{1-\lambda} \ . \tag{86}$$

*Proof.*

$$\frac{1}{K}\mathbb{E}[\|X_t - \bar{X}_t\|_F^2]$$

$$= \frac{1}{K}\mathbb{E}[\|(X_{t-1} - \eta_x\hat{P}_{t-1})E - (\bar{X}_{t-1} - \eta_x\bar{\hat{P}}_{t-1})\|_F^2]$$

$$\overset{(a)}{=} \frac{1}{K}\mathbb{E}[\|((X_{t-1} - \eta_x\hat{P}_{t-1}) - (\bar{X}_{t-1} - \eta_x\bar{\hat{P}}_{t-1}))(E - \frac{\mathbf{1}\mathbf{1}^T}{K})\|_F^2]$$

$$\leq \frac{1}{K}\mathbb{E}[\|(X_{t-1} - \eta_x\hat{P}_{t-1}) - (\bar{X}_{t-1} - \eta_x\bar{\hat{P}}_{t-1})\|_F^2\|E - \frac{\mathbf{1}\mathbf{1}^T}{K}\|_2^2]$$

$$\overset{(b)}{\leq} \lambda^2\frac{1}{K}\mathbb{E}[\|(X_{t-1} - \eta_x\hat{P}_{t-1}) - (\bar{X}_{t-1} - \eta_x\bar{\hat{P}}_{t-1})\|_F^2]$$

$$\leq \lambda^2(1 + 1/a)\frac{1}{K}\mathbb{E}[\|X_{t-1} - \bar{X}_{t-1}\|_F^2] + \eta_x^2\lambda^2(1 + a)\frac{1}{K}\mathbb{E}[\|\hat{P}_{t-1} - \bar{\hat{P}}_{t-1}\|_F^2]$$

$$\overset{(c)}{\leq} \lambda\frac{1}{K}\mathbb{E}[\|X_{t-1} - \bar{X}_{t-1}\|_F^2] + \eta_x^2\frac{\lambda^2}{1 - \lambda}\frac{1}{K}\mathbb{E}[\|\hat{P}_{t-1} - \bar{\hat{P}}_{t-1}\|_F^2]$$

$$\leq \lambda\frac{1}{K}\mathbb{E}[\|X_{t-1} - \bar{X}_{t-1}\|_F^2] + \eta_x^2\frac{\lambda^2}{1 - \lambda}\frac{1}{K}\mathbb{E}[\|\hat{P}_{t-1}\|_F^2]$$

$$= \lambda\frac{1}{K}\mathbb{E}[\|X_{t-1} - \bar{X}_{t-1}\|_F^2] + \eta_x^2\frac{\lambda^2}{1 - \lambda}\frac{1}{K}\sum_{k=1}^{K}\mathbb{E}[\|\frac{p_{t-1}^{(k)}}{\|p_{t-1}^{(k)}\|}\|^2]$$

$$= \lambda\frac{1}{K}\mathbb{E}[\|X_{t-1} - \bar{X}_{t-1}\|_F^2] + \eta_x^2\frac{\lambda^2}{1 - \lambda}$$

$$\leq \frac{\eta_x^2\lambda^2}{(1 - \lambda)^2}, \tag{87}$$

where $(a)$ holds because $E$ is a doubly stochastic matrix, $(b)$ holds due to Assumption 3.3, $(c)$ holds due to $a = \frac{\lambda}{1-\lambda}$.

Then, we obtain

$$\frac{1}{K}\sum_{k=1}^{K}\mathbb{E}[\|x_t^{(k)} - \bar{x}_t\|] = \sqrt{\frac{1}{K^2}\left(\sum_{k=1}^{K}\mathbb{E}[\|x_t^{(k)} - \bar{x}_t\|]\right)^2}$$

$$\leq \sqrt{\frac{1}{K^2}K\sum_{k=1}^{K}\mathbb{E}[\|x_t^{(k)} - \bar{x}_t\|^2]} \leq \sqrt{\frac{1}{K}\mathbb{E}[\|X_t - \bar{X}_t\|_F^2]} \leq \frac{\eta_x\lambda}{1 - \lambda}. \tag{88}$$

The other two inequalities can be proved in a same approach. $\square$

**Lemma C.29.** *Given Assumptions 3.1-3.3, we obtain*

$$\frac{1}{T}\sum_{t=0}^{T-1}\frac{1}{K}\sum_{k=1}^{K}\mathbb{E}[\|p_t^{(k)} - \bar{p}_t\|]$$

$$\leq \frac{2\lambda}{(1 - \lambda)T}\frac{1}{\sqrt{K}}\sum_{k=1}^{K}\mathbb{E}[\|\nabla_1 f^{(k)}(x_0^{(k)}, y_0^{(k)})\|] + \frac{2\lambda}{(1 - \lambda)T}\frac{1}{\delta}\frac{1}{\sqrt{K}}\sum_{k=1}^{K}\mathbb{E}[\|\nabla_1 g^{(k)}(x_0^{(k)}, y_0^{(k)})\|]$$

$$+ \frac{2\lambda}{(1 - \lambda)T}\frac{1}{\delta}\frac{1}{\sqrt{K}}\sum_{k=1}^{K}\mathbb{E}[\|\nabla_1 g^{(k)}(x_0^{(k)}, z_0^{(k)})\|] + \frac{\lambda}{(1 - \lambda)T}\frac{4\sqrt{2}\sqrt{K}}{B_0^{1-1/s}}\left(1 + \frac{2}{\delta}\right)\sigma$$

$$+ \frac{\gamma_x\lambda\sqrt{K}\sigma}{(1 - \lambda)^{3/2}}\left(1 + \frac{2}{\delta}\right) + \frac{4\eta_x\lambda\sqrt{K}}{(1 - \lambda)^{5/2}}\left(L_f + \frac{2L_g}{\delta}\right) + \frac{4\eta_y\lambda\sqrt{K}}{(1 - \lambda)^{5/2}}\left(L_f + \frac{L_g}{\delta}\right) + \frac{4\eta_z\lambda\sqrt{K}}{(1 - \lambda)^{5/2}}\frac{L_g}{\delta}$$

$$+ \frac{\lambda\sqrt{K}}{T(1 - \lambda)^{3/2}}\frac{2\sqrt{2}\sigma}{B_0^{1-1/s}}\left(1 + \frac{2}{\delta}\right) + \frac{2\sqrt{2}\gamma_x^{2-1/s}\lambda\sqrt{K}}{(1 - \lambda)^{3/2}}\sigma\left(1 + \frac{2}{\delta}\right)$$

$$+ \frac{4\eta_x\sqrt{\gamma_x}\lambda}{(1 - \lambda)^{5/2}}\left(L_f + \frac{2L_g}{\delta}\right) + \frac{4\eta_y\sqrt{\gamma_x}\lambda}{(1 - \lambda)^{5/2}}\left(L_f + \frac{2L_g}{\delta}\right). \tag{89}$$

*Proof.* When $t > 0$, similar to the proof of Lemma C.28, we obtain

$$\frac{1}{K}\mathbb{E}[\|P_t - \bar{P}_t\|_F^2]$$

$$= \frac{1}{K}\mathbb{E}[\|(P_{t-1} - U_{t-1} + U_t)E - (\bar{P}_{t-1} - \bar{U}_{t-1} + \bar{U}_t)\|_F^2]$$

$$\overset{(a)}{\leq} \lambda^2 \frac{1}{K} \mathbb{E}[\|(P_{t-1} - U_{t-1} + U_t) - (\bar{P}_{t-1} - \bar{U}_{t-1} + \bar{U}_t)\|_F^2]$$

$$\leq \lambda^2 (1 + 1/a) \frac{1}{K} \mathbb{E}[\|P_{t-1} - \bar{P}_{t-1}\|_F^2] + \lambda^2 (1 + a) \frac{1}{K} \mathbb{E}[\|U_t - U_{t-1} - (\bar{U}_t - \bar{U}_{t-1})\|_F^2]$$

$$\leq \lambda^2 (1 + 1/a) \frac{1}{K} \mathbb{E}[\|P_{t-1} - \bar{P}_{t-1}\|_F^2] + \lambda^2 (1 + a) \frac{1}{K} \mathbb{E}[\|U_t - U_{t-1}\|_F^2]$$

$$\overset{(b)}{\leq} \lambda \frac{1}{K} \mathbb{E}[\|P_{t-1} - \bar{P}_{t-1}\|_F^2] + \frac{\lambda^2}{1 - \lambda} \frac{1}{K} \mathbb{E}[\|U_t - U_{t-1}\|_F^2]$$

$$\leq \lambda^t \frac{1}{K} \mathbb{E}[\|P_0 - \bar{P}_0\|_F^2] + \sum_{j=1}^{t} \lambda^{t-j} \frac{\lambda^2}{1 - \lambda} \frac{1}{K} \mathbb{E}[\|U_j - U_{j-1}\|_F^2] , \tag{90}$$

where $(a)$ holds due to Assumption 3.3, $(b)$ holds due to $a = \frac{\lambda}{1-\lambda}$.

For $\frac{1}{K} \mathbb{E}[\|P_0 - \bar{P}_0\|_F^2]$, we bound it as follows:

$$\frac{1}{K} \mathbb{E}[\|P_0 - \bar{P}_0\|_F^2] = \frac{1}{K} \mathbb{E}[\|U_0 E - \bar{U}_0\|_F^2] = \frac{1}{K} \mathbb{E}[\|(U_0 - \bar{U}_0)(E - \frac{\mathbf{1}\mathbf{1}^T}{K})\|_F^2]$$

$$\leq \frac{1}{K} \mathbb{E}[\|U_0 - \bar{U}_0\|_F^2 \|E - \frac{\mathbf{1}\mathbf{1}^T}{K}\|_2^2] \leq \lambda^2 \frac{1}{K} \mathbb{E}[\|U_0 - \bar{U}_0\|_F^2] . \tag{91}$$

Then, we obtain

$$\frac{1}{K} \sum_{k=1}^{K} \mathbb{E}[\|p_t^{(k)} - \bar{p}_t\|] = \sqrt{\frac{1}{K^2} \left( \sum_{k=1}^{K} \mathbb{E}[\|p_t^{(k)} - \bar{p}_t\|] \right)^2}$$

$$\leq \sqrt{\frac{1}{K^2} K \sum_{k=1}^{K} \mathbb{E}[\|p_t^{(k)} - \bar{p}_t\|^2]} = \sqrt{\frac{1}{K} \mathbb{E}[\|P_t - \bar{P}_t\|_F^2]}$$

$$\leq \sqrt{\lambda^{t+2} \frac{1}{K} \mathbb{E}[\|U_0 - \bar{U}_0\|_F^2]} + \sqrt{\sum_{j=1}^{t} \lambda^{t-j} \frac{\lambda^2}{1 - \lambda} \frac{1}{K} \mathbb{E}[\|U_j - U_{j-1}\|_F^2]}$$

$$\overset{(a)}{\leq} \lambda^{1+t/2} \frac{1}{\sqrt{K}} \sum_{k=1}^{K} \mathbb{E}[\|u_0^{(k)} - \bar{u}_0\|] + \sum_{j=1}^{t} \lambda^{(t-j)/2} \frac{\lambda}{\sqrt{1-\lambda}} \frac{1}{\sqrt{K}} \sum_{k=1}^{K} \mathbb{E}[\|u_j^{(k)} - u_{j-1}^{(k)}\|]$$

$$\overset{(b)}{\leq} \lambda^{1+t/2} \frac{1}{\sqrt{K}} \sum_{k=1}^{K} \mathbb{E}[\|u_0^{(k)} - \bar{u}_0\|] + \sum_{j=1}^{t} \lambda^{(t-j)/2} \frac{\lambda}{\sqrt{1-\lambda}} \frac{1}{\sqrt{K}} \sum_{k=1}^{K} \mathbb{E}[\|u_{1,j}^{(k)} - u_{1,j-1}^{(k)}\|]$$

$$+ \sum_{j=1}^{t} \lambda^{(t-j)/2} \frac{\lambda}{\sqrt{1-\lambda}} \frac{1}{\delta} \frac{1}{\sqrt{K}} \sum_{k=1}^{K} \mathbb{E}[\|u_{2,j}^{(k)} - u_{2,j-1}^{(k)}\|]$$

$$+ \sum_{j=1}^{t} \lambda^{(t-j)/2} \frac{\lambda}{\sqrt{1-\lambda}} \frac{1}{\delta} \frac{1}{\sqrt{K}} \sum_{k=1}^{K} \mathbb{E}[\|u_{3,j}^{(k)} - u_{3,j-1}^{(k)}\|] , \tag{92}$$

where $(a)$ and $(b)$ hold due to $\sqrt{\sum_{i=1}^{n} a_i} \leq \sum_{i=1}^{n} \sqrt{a_i}$ for any $a_i \geq 0$ and $n > 1$.

Note that this upper bound also holds when $t = 0$ according to Eq. (91).

Then, due to Lemmas C.10 - C.12, we obtain

$$\frac{1}{K} \sum_{k=1}^{K} \mathbb{E}[\|p_t^{(k)} - \bar{p}_t\|]$$

$$\leq \lambda^{1+t/2} \frac{1}{\sqrt{K}} \sum_{k=1}^{K} \mathbb{E}[\|u_0^{(k)} - \bar{u}_0\|]$$

$$+ \gamma_x \sum_{j=1}^{t} \lambda^{(t-j)/2} \frac{\lambda}{\sqrt{1-\lambda}} \frac{1}{\sqrt{K}} \sum_{k=1}^{K} \mathbb{E}[\|\nabla_1 f^{(k)}(x_{j-1}^{(k)}, y_{j-1}^{(k)}) - \nabla_1 f^{(k)}(x_{j-1}^{(k)}, y_{j-1}^{(k)}; \xi_j^{(k)})\|]$$

$$+ \gamma_x \sum_{j=1}^{t} \lambda^{(t-j)/2} \frac{\lambda}{\sqrt{1-\lambda}} \frac{1}{\delta} \frac{1}{\sqrt{K}} \sum_{k=1}^{K} \mathbb{E}[\|\nabla_1 g^{(k)}(x_{j-1}^{(k)}, y_{j-1}^{(k)}) - \nabla_1 g^{(k)}(x_{j-1}^{(k)}, y_{j-1}^{(k)}; \zeta_j^{(k)})\|]$$

$$+ \gamma_x \sum_{j=1}^{t} \lambda^{(t-j)/2} \frac{\lambda}{\sqrt{1-\lambda}} \frac{1}{\delta} \frac{1}{\sqrt{K}} \sum_{k=1}^{K} \mathbb{E}[\|\nabla_1 g^{(k)}(x_{j-1}^{(k)}, z_{j-1}^{(k)}) - \nabla_1 g^{(k)}(x_{j-1}^{(k)}, z_{j-1}^{(k)}; \zeta_j^{(k)})\|]$$

$$+ \gamma_x \sum_{j=1}^{t} \lambda^{(t-j)/2} \frac{\lambda}{\sqrt{1-\lambda}} \frac{1}{\sqrt{K}} \sum_{k=1}^{K} \mathbb{E}[\|u_{1,j-1}^{(k)} - \nabla_1 f^{(k)}(x_{j-1}^{(k)}, y_{j-1}^{(k)})\|]$$

$$+ \gamma_x \sum_{j=1}^{t} \lambda^{(t-j)/2} \frac{\lambda}{\sqrt{1-\lambda}} \frac{1}{\delta} \frac{1}{\sqrt{K}} \sum_{k=1}^{K} \mathbb{E}[\|u_{2,j-1}^{(k)} - \nabla_1 g^{(k)}(x_{j-1}^{(k)}, y_{j-1}^{(k)})\|]$$

$$+ \gamma_x \sum_{j=1}^{t} \lambda^{(t-j)/2} \frac{\lambda}{\sqrt{1-\lambda}} \frac{1}{\delta} \frac{1}{\sqrt{K}} \sum_{k=1}^{K} \mathbb{E}[\|u_{3,j-1}^{(k)} - \nabla_1 g^{(k)}(x_{j-1}^{(k)}, z_{j-1}^{(k)})\|]$$

$$+ \frac{4\eta_x \sqrt{K}}{1-\lambda} \frac{\lambda}{\sqrt{1-\lambda}} \left( L_f + \frac{2L_g}{\delta} \right) \sum_{j=1}^{t} \lambda^{(t-j)/2}$$

$$+ \frac{4\eta_y \sqrt{K}}{1-\lambda} \frac{\lambda}{\sqrt{1-\lambda}} \left( L_f + \frac{L_g}{\delta} \right) \sum_{j=1}^{t} \lambda^{(t-j)/2} + \frac{4\eta_z \sqrt{K}}{1-\lambda} \frac{\lambda}{\sqrt{1-\lambda}} \frac{L_g}{\delta} \sum_{j=1}^{t} \lambda^{(t-j)/2} . \tag{93}$$

Therefore, we obtain

$$\frac{1}{T} \sum_{t=0}^{T-1} \frac{1}{K} \sum_{k=1}^{K} \mathbb{E}[\|p_t^{(k)} - \bar{p}_t\|]$$

$$= \frac{1}{T} \frac{1}{K} \sum_{k=1}^{K} \mathbb{E}[\|p_0^{(k)} - \bar{p}_0\|] + \frac{1}{T} \sum_{t=1}^{T-1} \frac{1}{K} \sum_{k=1}^{K} \mathbb{E}[\|p_t^{(k)} - \bar{p}_t\|]$$

$$\leq \lambda \frac{1}{T} \sum_{t=0}^{T-1} \lambda^{t/2} \frac{1}{\sqrt{K}} \sum_{k=1}^{K} \mathbb{E}[\|u_0^{(k)} - \bar{u}_0\|]$$

$$+ \gamma_x \frac{\lambda}{\sqrt{1-\lambda}} \frac{1}{T} \sum_{t=1}^{T-1} \sum_{j=1}^{t} \lambda^{(t-j)/2} \frac{1}{\sqrt{K}} \sum_{k=1}^{K} \mathbb{E}[\|\nabla_1 f^{(k)}(x_{j-1}^{(k)}, y_{j-1}^{(k)}) - \nabla_1 f^{(k)}(x_{j-1}^{(k)}, y_{j-1}^{(k)}; \xi_j^{(k)})\|]$$

$$+ \gamma_x \frac{\lambda}{\sqrt{1-\lambda}} \frac{1}{\delta} \frac{1}{T} \sum_{t=1}^{T-1} \sum_{j=1}^{t} \lambda^{(t-j)/2} \frac{1}{\sqrt{K}} \sum_{k=1}^{K} \mathbb{E}[\|\nabla_1 g^{(k)}(x_{j-1}^{(k)}, y_{j-1}^{(k)}) - \nabla_1 g^{(k)}(x_{j-1}^{(k)}, y_{j-1}^{(k)}; \zeta_j^{(k)})\|]$$

$$+ \gamma_x \frac{\lambda}{\sqrt{1-\lambda}} \frac{1}{\delta} \frac{1}{T} \sum_{t=1}^{T-1} \sum_{j=1}^{t} \lambda^{(t-j)/2} \frac{1}{\sqrt{K}} \sum_{k=1}^{K} \mathbb{E}[\|\nabla_1 g^{(k)}(x_{j-1}^{(k)}, z_{j-1}^{(k)}) - \nabla_1 g^{(k)}(x_{j-1}^{(k)}, z_{j-1}^{(k)}; \zeta_j^{(k)})\|]$$

$$+ \gamma_x \frac{\lambda}{\sqrt{1-\lambda}} \frac{1}{T} \sum_{t=1}^{T-1} \sum_{j=1}^{t} \lambda^{(t-j)/2} \frac{1}{\sqrt{K}} \sum_{k=1}^{K} \mathbb{E}[\|u_{1,j-1}^{(k)} - \nabla_1 f^{(k)}(x_{j-1}^{(k)}, y_{j-1}^{(k)})\|]$$

$$+ \gamma_x \frac{\lambda}{\sqrt{1-\lambda}} \frac{1}{\delta} \frac{1}{T} \sum_{t=1}^{T-1} \sum_{j=1}^{t} \lambda^{(t-j)/2} \frac{1}{\sqrt{K}} \sum_{k=1}^{K} \mathbb{E}[\|u_{2,j-1}^{(k)} - \nabla_1 g^{(k)}(x_{j-1}^{(k)}, y_{j-1}^{(k)})\|]$$

$$+ \gamma_x \frac{\lambda}{\sqrt{1-\lambda}} \frac{1}{\delta} \frac{1}{T} \sum_{t=1}^{T-1} \sum_{j=1}^{t} \lambda^{(t-j)/2} \frac{1}{\sqrt{K}} \sum_{k=1}^{K} \mathbb{E}[\|u_{3,j-1}^{(k)} - \nabla_1 g^{(k)}(x_{j-1}^{(k)}, z_{j-1}^{(k)})\|]$$

$$+ \frac{4\eta_x\sqrt{K}}{1-\lambda}\frac{\lambda}{\sqrt{1-\lambda}}\left(L_f + \frac{2L_g}{\delta}\right)\frac{1}{T}\sum_{t=1}^{T-1}\sum_{j=1}^{t}\lambda^{(t-j)/2}$$

$$+ \frac{4\eta_y\sqrt{K}}{1-\lambda}\frac{\lambda}{\sqrt{1-\lambda}}\left(L_f + \frac{L_g}{\delta}\right)\frac{1}{T}\sum_{t=1}^{T-1}\sum_{j=1}^{t}\lambda^{(t-j)/2} + \frac{4\eta_z\sqrt{K}}{1-\lambda}\frac{\lambda}{\sqrt{1-\lambda}}\frac{L_g}{\delta}\frac{1}{T}\sum_{t=1}^{T-1}\sum_{j=1}^{t}\lambda^{(t-j)/2}\,. \tag{94}$$

Note that $\nabla_1 f^{(k)}(x_{j-1}^{(k)}, y_{j-1}^{(k)}; \xi_j^{(k)})$ is computed with the samples in the $j \geq 1$-th iteration, where the batch size is 1, then for any $j \in \{1, \cdots, t\}$, we obtain

$$\mathbb{E}[\|\nabla_1 f^{(k)}(x_{j-1}^{(k)}, y_{j-1}^{(k)}) - \nabla_1 f^{(k)}(x_{j-1}^{(k)}, y_{j-1}^{(k)}; \xi_j^{(k)})\|]$$
$$= \mathbb{E}[(\|\nabla_1 f^{(k)}(x_{j-1}^{(k)}, y_{j-1}^{(k)}) - \nabla_1 f^{(k)}(x_{j-1}^{(k)}, y_{j-1}^{(k)}; \xi_j^{(k)})\|^s)^{1/s}]$$
$$\overset{(a)}{\leq} (\mathbb{E}[\|\nabla_1 f^{(k)}(x_{j-1}^{(k)}, y_{j-1}^{(k)}) - \nabla_1 f^{(k)}(x_{j-1}^{(k)}, y_{j-1}^{(k)}; \xi_j^{(k)})\|^s])^{1/s}$$
$$\overset{(b)}{\leq} \sigma\,, \tag{95}$$

where $(a)$ holds due to Hölder's inequality, and $(b)$ holds due to Assumption 3.2.

Similarly, we obtain

$$\mathbb{E}[\|\nabla_1 g^{(k)}(x_{j-1}^{(k)}, y_{j-1}^{(k)}) - \nabla_1 g^{(k)}(x_{j-1}^{(k)}, y_{j-1}^{(k)}; \zeta_j^{(k)})\|] \leq \sigma\,,$$
$$\mathbb{E}[\|\nabla_1 g^{(k)}(x_{j-1}^{(k)}, z_{j-1}^{(k)}) - \nabla_1 g^{(k)}(x_{j-1}^{(k)}, z_{j-1}^{(k)}; \zeta_j^{(k)})\|] \leq \sigma\,. \tag{96}$$

Then, we obtain

$$\frac{1}{T}\sum_{t=0}^{T-1}\frac{1}{K}\sum_{k=1}^{K}\mathbb{E}[\|p_t^{(k)} - \bar{p}_t\|]$$

$$\leq \lambda\frac{1}{T}\sum_{t=0}^{T-1}\lambda^{t/2}\frac{1}{\sqrt{K}}\sum_{k=1}^{K}\mathbb{E}[\|u_0^{(k)} - \bar{u}_0\|] + \gamma_x\frac{\lambda\sqrt{K}\sigma}{\sqrt{1-\lambda}}\left(1 + \frac{2}{\delta}\right)\frac{1}{T}\sum_{t=1}^{T-1}\sum_{j=1}^{t}\lambda^{(t-j)/2}$$

$$+ \frac{4\eta_x\sqrt{K}}{1-\lambda}\frac{\lambda}{\sqrt{1-\lambda}}\left(L_f + \frac{2L_g}{\delta}\right)\frac{1}{T}\sum_{t=1}^{T-1}\sum_{j=1}^{t}\lambda^{(t-j)/2}$$

$$+ \frac{4\eta_y\sqrt{K}}{1-\lambda}\frac{\lambda}{\sqrt{1-\lambda}}\left(L_f + \frac{L_g}{\delta}\right)\frac{1}{T}\sum_{t=1}^{T-1}\sum_{j=1}^{t}\lambda^{(t-j)/2} + \frac{4\eta_z\sqrt{K}}{1-\lambda}\frac{\lambda}{\sqrt{1-\lambda}}\frac{L_g}{\delta}\frac{1}{T}\sum_{t=1}^{T-1}\sum_{j=1}^{t}\lambda^{(t-j)/2}$$

$$+ \gamma_x\frac{\lambda}{\sqrt{1-\lambda}}\frac{1}{T}\sum_{t=1}^{T-1}\sum_{j=1}^{t}\lambda^{(t-j)/2}\frac{1}{\sqrt{K}}\sum_{k=1}^{K}\mathbb{E}[\|u_{1,j-1}^{(k)} - \nabla_1 f^{(k)}(x_{j-1}^{(k)}, y_{j-1}^{(k)})\|]$$

$$+ \gamma_x\frac{\lambda}{\sqrt{1-\lambda}}\frac{1}{\delta}\frac{1}{T}\sum_{t=1}^{T-1}\sum_{j=1}^{t}\lambda^{(t-j)/2}\frac{1}{\sqrt{K}}\sum_{k=1}^{K}\mathbb{E}[\|u_{2,j-1}^{(k)} - \nabla_1 g^{(k)}(x_{j-1}^{(k)}, y_{j-1}^{(k)})\|]$$

$$+ \gamma_x\frac{\lambda}{\sqrt{1-\lambda}}\frac{1}{\delta}\frac{1}{T}\sum_{t=1}^{T-1}\sum_{j=1}^{t}\lambda^{(t-j)/2}\frac{1}{\sqrt{K}}\sum_{k=1}^{K}\mathbb{E}[\|u_{3,j-1}^{(k)} - \nabla_1 g^{(k)}(x_{j-1}^{(k)}, z_{j-1}^{(k)})\|]\,. \tag{97}$$

Then, based on Lemmas C.16, C.17, C.18, we obtain

$$\frac{1}{T}\sum_{t=0}^{T-1}\frac{1}{K}\sum_{k=1}^{K}\mathbb{E}[\|p_t^{(k)} - \bar{p}_t\|]$$

$$\leq \lambda\frac{1}{T}\sum_{t=0}^{T-1}\lambda^{t/2}\frac{1}{\sqrt{K}}\sum_{k=1}^{K}\mathbb{E}[\|u_0^{(k)} - \bar{u}_0\|] + \gamma_x\frac{\lambda\sqrt{K}\sigma}{\sqrt{1-\lambda}}\left(1 + \frac{2}{\delta}\right)\frac{1}{T}\sum_{t=1}^{T-1}\sum_{j=1}^{t}\lambda^{(t-j)/2}$$

$$+ \frac{4\eta_x\sqrt{K}}{1-\lambda} \frac{\lambda}{\sqrt{1-\lambda}} \left(L_f + \frac{2L_g}{\delta}\right) \frac{1}{T} \sum_{t=1}^{T-1}\sum_{j=1}^{t} \lambda^{(t-j)/2}$$

$$+ \frac{4\eta_y\sqrt{K}}{1-\lambda} \frac{\lambda}{\sqrt{1-\lambda}} \left(L_f + \frac{L_g}{\delta}\right) \frac{1}{T} \sum_{t=1}^{T-1}\sum_{j=1}^{t} \lambda^{(t-j)/2} + \frac{4\eta_z\sqrt{K}}{1-\lambda} \frac{\lambda}{\sqrt{1-\lambda}} \frac{L_g}{\delta} \frac{1}{T} \sum_{t=1}^{T-1}\sum_{j=1}^{t} \lambda^{(t-j)/2}$$

$$+ \gamma_x \frac{\lambda\sqrt{K}}{\sqrt{1-\lambda}} \frac{2\sqrt{2}}{B_0^{1-1/s}} \sigma \left(1 + \frac{2}{\delta}\right) \frac{1}{T} \sum_{t=1}^{T-1} (1-\gamma_x)^t \sum_{j=1}^{t} \lambda^{(t-j)/2}$$

$$+ \gamma_x \frac{\lambda\sqrt{K}}{\sqrt{1-\lambda}} 2\sqrt{2}\gamma_x^{1-1/s}\sigma \left(1 + \frac{2}{\delta}\right) \frac{1}{T} \sum_{t=1}^{T-1}\sum_{j=1}^{t} \lambda^{(t-j)/2}$$

$$+ \gamma_x \frac{\lambda}{\sqrt{1-\lambda}} \frac{4\eta_x}{(1-\lambda)\sqrt{\gamma_x}} \left(L_f + \frac{2L_g}{\delta}\right) \frac{1}{T} \sum_{t=1}^{T-1}\sum_{j=1}^{t} \lambda^{(t-j)/2}$$

$$+ \gamma_x \frac{\lambda}{\sqrt{1-\lambda}} \frac{4\eta_y}{(1-\lambda)\sqrt{\gamma_x}} \left(L_f + \frac{2L_g}{\delta}\right) \frac{1}{T} \sum_{t=1}^{T-1}\sum_{j=1}^{t} \lambda^{(t-j)/2} . \tag{98}$$

Because

$$\frac{1}{T} \sum_{t=0}^{T-1} \lambda^{t/2} \le \frac{1}{(1-\sqrt{\lambda})T} \le \frac{1}{(1-\lambda)T} ,$$

$$\frac{1}{T} \sum_{t=1}^{T-1}\sum_{j=1}^{t} \lambda^{(t-j)/2} \le \frac{1}{T} \sum_{t=1}^{T-1}\sum_{j=1}^{T-1} \lambda^{(T-1-j)/2} \le \frac{1}{1-\sqrt{\lambda}} \le \frac{1}{1-\lambda} ,$$

$$\frac{1}{T} \sum_{t=1}^{T-1} (1-\gamma_x)^t \sum_{j=1}^{t} \lambda^{(t-j)/2} \le \frac{1}{T} \sum_{t=1}^{T-1} (1-\gamma_x)^t \sum_{j=1}^{T-1} \lambda^{(T-1-j)/2}$$

$$\le \frac{1}{1-\sqrt{\lambda}} \frac{1}{T} \sum_{t=1}^{T-1} (1-\gamma_x)^t \le \frac{1}{(1-\lambda)\gamma_x T} , \tag{99}$$

we obtain

$$\frac{1}{T} \sum_{t=0}^{T-1} \frac{1}{K} \sum_{k=1}^{K} \mathbb{E}[\|p_t^{(k)} - \bar{p}_t\|]$$

$$\le \frac{\lambda}{(1-\lambda)T} \frac{1}{\sqrt{K}} \sum_{k=1}^{K} \mathbb{E}[\|u_0^{(k)} - \bar{u}_0\|] + \frac{\gamma_x\lambda\sqrt{K}\sigma}{(1-\lambda)^{3/2}} \left(1 + \frac{2}{\delta}\right)$$

$$+ \frac{4\eta_x\lambda\sqrt{K}}{(1-\lambda)^{5/2}} \left(L_f + \frac{2L_g}{\delta}\right) + \frac{4\eta_y\lambda\sqrt{K}}{(1-\lambda)^{5/2}} \left(L_f + \frac{L_g}{\delta}\right) + \frac{4\eta_z\lambda\sqrt{K}}{(1-\lambda)^{5/2}} \frac{L_g}{\delta}$$

$$+ \frac{\lambda\sqrt{K}}{T(1-\lambda)^{3/2}} \frac{2\sqrt{2}\sigma}{B_0^{1-1/s}} \left(1 + \frac{2}{\delta}\right) + \frac{2\sqrt{2}\gamma_x^{2-1/s}\lambda\sqrt{K}}{(1-\lambda)^{3/2}} \sigma \left(1 + \frac{2}{\delta}\right)$$

$$+ \frac{4\eta_x\sqrt{\gamma_x}\lambda}{(1-\lambda)^{5/2}} \left(L_f + \frac{2L_g}{\delta}\right) + \frac{4\eta_y\sqrt{\gamma_x}\lambda}{(1-\lambda)^{5/2}} \left(L_f + \frac{2L_g}{\delta}\right) . \tag{100}$$

For $\sum_{k=1}^{K} \mathbb{E}[\|u_0^{(k)} - \bar{u}_0\|]$, we bound it as follows:

$$\sum_{k=1}^{K} \mathbb{E}[\|u_0^{(k)} - \bar{u}_0\|] = \sum_{k=1}^{K} \mathbb{E}[\|u_{1,0}^{(k)} + \frac{1}{\delta}u_{2,0}^{(k)} + \frac{1}{\delta}u_{3,0}^{(k)} - \frac{1}{K}\sum_{j=1}^{K}(u_{1,0}^{(j)} + \frac{1}{\delta}u_{2,0}^{(j)} + \frac{1}{\delta}u_{3,0}^{(j)})\|]$$

$$\le \sum_{k=1}^{K} \mathbb{E}[\|u_{1,0}^{(k)} - \frac{1}{K}\sum_{j=1}^{K} u_{1,0}^{(j)}\|] + \sum_{k=1}^{K} \mathbb{E}[\|\frac{1}{\delta}u_{2,0}^{(k)} - \frac{1}{K}\sum_{j=1}^{K} \frac{1}{\delta}u_{2,0}^{(j)}\|] + \sum_{k=1}^{K} \mathbb{E}[\|\frac{1}{\delta}u_{3,0}^{(k)} - \frac{1}{K}\sum_{j=1}^{K} \frac{1}{\delta}u_{3,0}^{(j)}\|]$$

$$\leq \sum_{k=1}^{K} \mathbb{E}[\|\nabla_1 f^{(k)}(x_0^{(k)}, y_0^{(k)}; \xi_0^{(k)}) - \frac{1}{K}\sum_{j=1}^{K} \nabla_1 f^{(j)}(x_0^{(j)}, y_0^{(j)}; \xi_0^{(j)})\|]$$

$$+ \frac{1}{\delta}\sum_{k=1}^{K} \mathbb{E}[\|\nabla_1 g^{(k)}(x_0^{(k)}, y_0^{(k)}; \zeta_0^{(k)}) - \frac{1}{K}\sum_{j=1}^{K} \nabla_1 g^{(j)}(x_0^{(j)}, y_0^{(j)}; \zeta_0^{(j)})\|]$$

$$+ \frac{1}{\delta}\sum_{k=1}^{K} \mathbb{E}[\|\nabla_1 g^{(k)}(x_0^{(k)}, z_0^{(k)}; \zeta_0^{(k)}) - \frac{1}{K}\sum_{j=1}^{K} \nabla_1 g^{(j)}(x_0^{(j)}, z_0^{(j)}; \zeta_0^{(j)})\|] . \tag{101}$$

For the first term on the last step of Eq. (101), we bound it as follows:

$$\sum_{k=1}^{K} \mathbb{E}[\|\nabla_1 f^{(k)}(x_0^{(k)}, y_0^{(k)}; \xi_0^{(k)}) - \frac{1}{K}\sum_{j=1}^{K} \nabla_1 f^{(j)}(x_0^{(j)}, y_0^{(j)}; \xi_0^{(j)})\|]$$

$$\leq \sum_{k=1}^{K} \mathbb{E}[\|\nabla_1 f^{(k)}(x_0^{(k)}, y_0^{(k)}; \xi_0^{(k)}) - \nabla_1 f^{(k)}(x_0^{(k)}, y_0^{(k)})\|]$$

$$+ \sum_{k=1}^{K} \mathbb{E}[\|\nabla_1 f^{(k)}(x_0^{(k)}, y_0^{(k)}) - \frac{1}{K}\sum_{j=1}^{K} \nabla_1 f^{(j)}(x_0^{(j)}, y_0^{(j)})\|]$$

$$+ \sum_{k=1}^{K} \mathbb{E}[\|\frac{1}{K}\sum_{j=1}^{K} \nabla_1 f^{(j)}(x_0^{(j)}, y_0^{(j)}) - \frac{1}{K}\sum_{j=1}^{K} \nabla_1 f^{(j)}(x_0^{(j)}, y_0^{(j)}; \xi_0^{(j)})\|]$$

$$\leq 2\sum_{k=1}^{K} \mathbb{E}[\|\nabla_1 f^{(k)}(x_0^{(k)}, y_0^{(k)}; \xi_0^{(k)}) - \nabla_1 f^{(k)}(x_0^{(k)}, y_0^{(k)})\|] + 2\sum_{k=1}^{K} \mathbb{E}[\|\nabla_1 f^{(k)}(x_0^{(k)}, y_0^{(k)})\|]$$

$$\overset{(a)}{\leq} 2\sum_{k=1}^{K} \mathbb{E}[\|\nabla_1 f^{(k)}(x_0^{(k)}, y_0^{(k)})\|] + \frac{4\sqrt{2}K}{B_0^{1-1/s}}\sigma , \tag{102}$$

where $(a)$ holds due to Eq. (61)

Similarly, we bound the second term on the last step of Eq. (101) as follows:

$$\frac{1}{\delta}\sum_{k=1}^{K} \mathbb{E}[\|\nabla_1 g^{(k)}(x_0^{(k)}, y_0^{(k)}; \zeta_0^{(k)}) - \frac{1}{K}\sum_{j=1}^{K} \nabla_1 g^{(j)}(x_0^{(j)}, y_0^{(j)}; \zeta_0^{(j)})\|]$$

$$\leq \frac{2}{\delta}\sum_{k=1}^{K} \mathbb{E}[\|\nabla_1 g^{(k)}(x_0^{(k)}, y_0^{(k)})\|] + \frac{4\sqrt{2}K}{B_0^{1-1/s}}\frac{1}{\delta}\sigma . \tag{103}$$

By combining them together, we obtain

$$\sum_{k=1}^{K} \mathbb{E}[\|u_0^{(k)} - \bar{u}_0\|] \leq 2\sum_{k=1}^{K} \mathbb{E}[\|\nabla_1 f^{(k)}(x_0^{(k)}, y_0^{(k)})\|] + \frac{2}{\delta}\sum_{k=1}^{K} \mathbb{E}[\|\nabla_1 g^{(k)}(x_0^{(k)}, y_0^{(k)})\|]$$

$$+ \frac{2}{\delta}\sum_{k=1}^{K} \mathbb{E}[\|\nabla_1 g^{(k)}(x_0^{(k)}, z_0^{(k)})\|] + \frac{4\sqrt{2}K}{B_0^{1-1/s}}\left(1 + \frac{2}{\delta}\right)\sigma . \tag{104}$$

Finally, we obtain

$$\frac{1}{T}\sum_{t=0}^{T-1}\frac{1}{K}\sum_{k=1}^{K} \mathbb{E}[\|p_t^{(k)} - \bar{p}_t\|]$$

$$\leq \frac{2\lambda}{(1-\lambda)T}\frac{1}{\sqrt{K}}\sum_{k=1}^{K} \mathbb{E}[\|\nabla_1 f^{(k)}(x_0^{(k)}, y_0^{(k)})\|] + \frac{2\lambda}{(1-\lambda)T}\frac{1}{\delta}\frac{1}{\sqrt{K}}\sum_{k=1}^{K} \mathbb{E}[\|\nabla_1 g^{(k)}(x_0^{(k)}, y_0^{(k)})\|]$$

$$
+ \frac{2\lambda}{(1-\lambda)T} \frac{1}{\delta} \frac{1}{\sqrt{K}} \sum_{k=1}^{K} \mathbb{E}[\|\nabla_1 g^{(k)}(x_0^{(k)}, z_0^{(k)})\|] + \frac{\lambda}{(1-\lambda)T} \frac{4\sqrt{2}\sqrt{K}}{B_0^{1-1/s}} \left(1 + \frac{2}{\delta}\right) \sigma
$$

$$
+ \frac{\gamma_x \lambda \sqrt{K} \sigma}{(1-\lambda)^{3/2}} \left(1 + \frac{2}{\delta}\right) + \frac{4\eta_x \lambda \sqrt{K}}{(1-\lambda)^{5/2}} \left(L_f + \frac{2L_g}{\delta}\right) + \frac{4\eta_y \lambda \sqrt{K}}{(1-\lambda)^{5/2}} \left(L_f + \frac{L_g}{\delta}\right) + \frac{4\eta_z \lambda \sqrt{K}}{(1-\lambda)^{5/2}} \frac{L_g}{\delta}
$$

$$
+ \frac{\lambda \sqrt{K}}{T(1-\lambda)^{3/2}} \frac{2\sqrt{2}\sigma}{B_0^{1-1/s}} \left(1 + \frac{2}{\delta}\right) + \frac{2\sqrt{2}\gamma_x^{2-1/s} \lambda \sqrt{K}}{(1-\lambda)^{3/2}} \sigma \left(1 + \frac{2}{\delta}\right)
$$

$$
+ \frac{4\eta_x \sqrt{\gamma_x} \lambda}{(1-\lambda)^{5/2}} \left(L_f + \frac{2L_g}{\delta}\right) + \frac{4\eta_y \sqrt{\gamma_x} \lambda}{(1-\lambda)^{5/2}} \left(L_f + \frac{2L_g}{\delta}\right) . \tag{105}
$$

$\square$

**Lemma C.30.** *Given Assumptions 3.1-3.3, we obtain*

$$
\frac{1}{T} \sum_{t=0}^{T-1} \frac{1}{K} \sum_{k=1}^{K} \mathbb{E}[\|q_t^{(k)} - \bar{q}_t\|]
$$

$$
\leq \frac{2\lambda}{(1-\lambda)T} \frac{1}{\sqrt{K}} \sum_{k=1}^{K} \mathbb{E}[\|\nabla_2 f^{(k)}(x_0^{(k)}, y_0^{(k)})\|] + \frac{2\lambda}{(1-\lambda)T} \frac{1}{\delta} \frac{1}{\sqrt{K}} \sum_{k=1}^{K} \mathbb{E}[\|\nabla_2 g^{(k)}(x_0^{(k)}, y_0^{(k)})\|]
$$

$$
+ \frac{\lambda}{(1-\lambda)T} \frac{4\sqrt{2}\sqrt{K}}{B_0^{1-1/s}} \left(1 + \frac{1}{\delta}\right) \sigma + \frac{\gamma_y \lambda \sqrt{K} \sigma}{(1-\lambda)^{3/2}} \left(1 + \frac{1}{\delta}\right)
$$

$$
+ \frac{4\eta_x \lambda \sqrt{K}}{(1-\lambda)^{5/2}} \left(L_f + \frac{L_g}{\delta}\right) + \frac{4\eta_y \lambda \sqrt{K}}{(1-\lambda)^{5/2}} \left(L_f + \frac{L_g}{\delta}\right)
$$

$$
+ \frac{\lambda \sqrt{K}}{T(1-\lambda)^{3/2}} \frac{2\sqrt{2}}{B_0^{1-1/s}} \left(1 + \frac{1}{\delta}\right) \sigma + \frac{2\sqrt{2}\gamma_y^{2-1/s} \lambda \sqrt{K}}{(1-\lambda)^{3/2}} \left(1 + \frac{1}{\delta}\right) \sigma
$$

$$
+ \frac{4\eta_x \sqrt{\gamma_y} \lambda}{(1-\lambda)^{5/2}} \left(L_f + \frac{L_g}{\delta}\right) + \frac{4\eta_y \sqrt{\gamma_y} \lambda}{(1-\lambda)^{5/2}} \left(L_f + \frac{L_g}{\delta}\right) . \tag{106}
$$

*Proof.* Following the proof of Lemma C.29, for any $t \geq 0$, we obtain

$$
\frac{1}{K} \sum_{k=1}^{K} \mathbb{E}[\|q_t^{(k)} - \bar{q}_t\|]
$$

$$
\leq \lambda^{1+t/2} \frac{1}{\sqrt{K}} \sum_{k=1}^{K} \mathbb{E}[\|v_0^{(k)} - \bar{v}_0\|] + \sum_{j=1}^{t} \lambda^{(t-j)/2} \frac{\lambda}{\sqrt{1-\lambda}} \frac{1}{\sqrt{K}} \sum_{k=1}^{K} \mathbb{E}[\|v_j^{(k)} - v_{j-1}^{(k)}\|]
$$

$$
\leq \lambda^{1+t/2} \frac{1}{\sqrt{K}} \sum_{k=1}^{K} \mathbb{E}[\|v_0^{(k)} - \bar{v}_0\|] + \sum_{j=1}^{t} \lambda^{(t-j)/2} \frac{\lambda}{\sqrt{1-\lambda}} \frac{1}{\sqrt{K}} \sum_{k=1}^{K} \mathbb{E}[\|v_{1,j}^{(k)} - v_{1,j-1}^{(k)}\|]
$$

$$
+ \sum_{j=1}^{t} \lambda^{(t-j)/2} \frac{\lambda}{\sqrt{1-\lambda}} \frac{1}{\delta} \frac{1}{\sqrt{K}} \sum_{k=1}^{K} \mathbb{E}[\|v_{2,j}^{(k)} - v_{2,j-1}^{(k)}\|] . \tag{107}
$$

Based on Lemma C.13 and Lemma C.14, we obtain

$$
\frac{1}{K} \sum_{k=1}^{K} \mathbb{E}[\|q_t^{(k)} - \bar{q}_t\|]
$$

$$
\leq \lambda^{1+t/2} \frac{1}{\sqrt{K}} \sum_{k=1}^{K} \mathbb{E}[\|v_0^{(k)} - \bar{v}_0\|]
$$

$$
+ \gamma_y \sum_{j=1}^{t} \lambda^{(t-j)/2} \frac{\lambda}{\sqrt{1-\lambda}} \frac{1}{\sqrt{K}} \sum_{k=1}^{K} \mathbb{E}[\|\nabla_2 f^{(k)}(x_{j-1}^{(k)}, y_{j-1}^{(k)}) - \nabla_2 f^{(k)}(x_{j-1}^{(k)}, y_{j-1}^{(k)}; \xi_j^{(k)})\|]
$$

$$+ \gamma_y \sum_{j=1}^{t} \lambda^{(t-j)/2} \frac{\lambda}{\sqrt{1-\lambda}} \frac{1}{\delta} \frac{1}{\sqrt{K}} \sum_{k=1}^{K} \mathbb{E}[\|\nabla_2 g^{(k)}(x_{j-1}^{(k)}, y_{j-1}^{(k)}) - \nabla_2 g^{(k)}(x_{j-1}^{(k)}, y_{j-1}^{(k)}; \zeta_j^{(k)})\|]$$

$$+ \gamma_y \sum_{j=1}^{t} \lambda^{(t-j)/2} \frac{\lambda}{\sqrt{1-\lambda}} \frac{1}{\sqrt{K}} \sum_{k=1}^{K} \mathbb{E}[\|v_{1,j-1}^{(k)} - \nabla_2 f^{(k)}(x_{j-1}^{(k)}, y_{j-1}^{(k)})\|]$$

$$+ \gamma_y \sum_{j=1}^{t} \lambda^{(t-j)/2} \frac{\lambda}{\sqrt{1-\lambda}} \frac{1}{\delta} \frac{1}{\sqrt{K}} \sum_{k=1}^{K} \mathbb{E}[\|v_{2,j-1}^{(k)} - \nabla_2 g^{(k)}(x_{j-1}^{(k)}, y_{j-1}^{(k)})\|]$$

$$+ \frac{4\eta_x \sqrt{K}}{1-\lambda} \frac{\lambda}{\sqrt{1-\lambda}} \left( L_f + \frac{L_g}{\delta} \right) \sum_{j=1}^{t} \lambda^{(t-j)/2}$$

$$+ \frac{4\eta_y \sqrt{K}}{1-\lambda} \frac{\lambda}{\sqrt{1-\lambda}} \left( L_f + \frac{L_g}{\delta} \right) \sum_{j=1}^{t} \lambda^{(t-j)/2} . \tag{108}$$

Then, we obtain

$$\frac{1}{T} \sum_{t=0}^{T-1} \frac{1}{K} \sum_{k=1}^{K} \mathbb{E}[\|q_t^{(k)} - \bar{q}_t\|]$$

$$\leq \frac{1}{T} \sum_{t=0}^{T-1} \lambda^{1+t/2} \frac{1}{\sqrt{K}} \sum_{k=1}^{K} \mathbb{E}[\|v_0^{(k)} - \bar{v}_0\|] + \frac{\gamma_y \lambda \sqrt{K} \sigma}{\sqrt{1-\lambda}} \left( 1 + \frac{1}{\delta} \right) \frac{1}{T} \sum_{t=0}^{T-1} \sum_{j=1}^{t} \lambda^{(t-j)/2}$$

$$+ \frac{4\eta_x \sqrt{K}}{1-\lambda} \frac{\lambda}{\sqrt{1-\lambda}} \left( L_f + \frac{L_g}{\delta} \right) \frac{1}{T} \sum_{t=0}^{T-1} \sum_{j=1}^{t} \lambda^{(t-j)/2}$$

$$+ \frac{4\eta_y \sqrt{K}}{1-\lambda} \frac{\lambda}{\sqrt{1-\lambda}} \left( L_f + \frac{L_g}{\delta} \right) \frac{1}{T} \sum_{t=0}^{T-1} \sum_{j=1}^{t} \lambda^{(t-j)/2}$$

$$+ \frac{\gamma_y \lambda \sqrt{K}}{\sqrt{1-\lambda}} \frac{2\sqrt{2}}{B_0^{1-1/s}} \sigma \left( 1 + \frac{1}{\delta} \right) \frac{1}{T} \sum_{t=0}^{T-1} (1-\gamma_y)^t \sum_{j=1}^{t} \lambda^{(t-j)/2}$$

$$+ \frac{4\eta_x \sqrt{\gamma_y}}{(1-\lambda)} \frac{\lambda}{\sqrt{1-\lambda}} \left( L_f + \frac{L_g}{\delta} \right) \frac{1}{T} \sum_{t=0}^{T-1} \sum_{j=1}^{t} \lambda^{(t-j)/2}$$

$$+ \frac{4\eta_y \sqrt{\gamma_y}}{(1-\lambda)} \frac{\lambda}{\sqrt{1-\lambda}} \left( L_f + \frac{L_g}{\delta} \right) \frac{1}{T} \sum_{t=0}^{T-1} \sum_{j=1}^{t} \lambda^{(t-j)/2}$$

$$+ \frac{\lambda \sqrt{K}}{\sqrt{1-\lambda}} 2\sqrt{2} \gamma_y^{2-1/s} \sigma \left( 1 + \frac{1}{\delta} \right) \frac{1}{T} \sum_{t=0}^{T-1} \sum_{j=1}^{t} \lambda^{(t-j)/2}$$

$$\leq \frac{\lambda}{(1-\lambda)T} \frac{1}{\sqrt{K}} \sum_{k=1}^{K} \mathbb{E}[\|v_0^{(k)} - \bar{v}_0\|] + \frac{\gamma_y \lambda \sqrt{K} \sigma}{(1-\lambda)^{3/2}} \left( 1 + \frac{1}{\delta} \right)$$

$$+ \frac{4\eta_x \lambda \sqrt{K}}{(1-\lambda)^{5/2}} \left( L_f + \frac{L_g}{\delta} \right) + \frac{4\eta_y \lambda \sqrt{K}}{(1-\lambda)^{5/2}} \left( L_f + \frac{L_g}{\delta} \right)$$

$$+ \frac{\lambda \sqrt{K}}{T(1-\lambda)^{3/2}} \frac{2\sqrt{2}}{B_0^{1-1/s}} \sigma \left( 1 + \frac{1}{\delta} \right) + \frac{2\sqrt{2} \gamma_y^{2-1/s} \lambda \sqrt{K}}{(1-\lambda)^{3/2}} \sigma \left( 1 + \frac{1}{\delta} \right)$$

$$+ \frac{4\eta_x \sqrt{\gamma_y} \lambda}{(1-\lambda)^{5/2}} \left( L_f + \frac{L_g}{\delta} \right) + \frac{4\eta_y \sqrt{\gamma_y} \lambda}{(1-\lambda)^{5/2}} \left( L_f + \frac{L_g}{\delta} \right) . \tag{109}$$

For $\sum_{k=1}^{K} \mathbb{E}[\|v_0^{(k)} - \bar{v}_0\|]$, we bound it as follows:

$$\sum_{k=1}^{K} \mathbb{E}[\|v_0^{(k)} - \bar{v}_0\|]$$

$$= \sum_{k=1}^{K} \mathbb{E}[\|v_{1,0}^{(k)} + \frac{1}{\delta} u_{2,0}^{(k)} - \frac{1}{K} \sum_{j=1}^{K} (v_{1,0}^{(j)} + \frac{1}{\delta} v_{2,0}^{(j)})\|]$$

$$\leq \sum_{k=1}^{K} \mathbb{E}[\|v_{1,0}^{(k)} - \frac{1}{K} \sum_{j=1}^{K} v_{1,0}^{(j)}\|] + \sum_{k=1}^{K} \mathbb{E}[\|\frac{1}{\delta} v_{2,0}^{(k)} - \frac{1}{K} \sum_{j=1}^{K} \frac{1}{\delta} v_{2,0}^{(j)}\|]$$

$$\leq \sum_{k=1}^{K} \mathbb{E}[\|\nabla_2 f^{(k)}(x_0^{(k)}, y_0^{(k)}; \xi_0^{(k)}) - \frac{1}{K} \sum_{j=1}^{K} \nabla_2 f^{(j)}(x_0^{(j)}, y_0^{(j)}; \xi_0^{(j)})\|]$$

$$+ \frac{1}{\delta} \sum_{k=1}^{K} \mathbb{E}[\|\nabla_2 g^{(k)}(x_0^{(k)}, y_0^{(k)}; \zeta_0^{(k)}) - \frac{1}{K} \sum_{j=1}^{K} \nabla_2 g^{(j)}(x_0^{(j)}, y_0^{(j)}; \zeta_0^{(j)})\|] \ . \qquad (110)$$

For the first term on the last step of Eq. (110), we bound it as follows:

$$\sum_{k=1}^{K} \mathbb{E}[\|\nabla_2 f^{(k)}(x_0^{(k)}, y_0^{(k)}; \xi_0^{(k)}) - \frac{1}{K} \sum_{j=1}^{K} \nabla_2 f^{(j)}(x_0^{(j)}, y_0^{(j)}; \xi_0^{(j)})\|]$$

$$\leq 2 \sum_{k=1}^{K} \mathbb{E}[\|\nabla_2 f^{(k)}(x_0^{(k)}, y_0^{(k)})\|] + \frac{4\sqrt{2}K}{B_0^{1-1/s}} \sigma \ . \qquad (111)$$

Similarly, we bound the second term on the last step of Eq. (110) as follows:

$$\frac{1}{\delta} \sum_{k=1}^{K} \mathbb{E}[\|\nabla_2 g^{(k)}(x_0^{(k)}, y_0^{(k)}; \zeta_0^{(k)}) - \frac{1}{K} \sum_{j=1}^{K} \nabla_2 g^{(j)}(x_0^{(j)}, y_0^{(j)}; \zeta_0^{(j)})\|]$$

$$\leq \frac{2}{\delta} \sum_{k=1}^{K} \mathbb{E}[\|\nabla_2 g^{(k)}(x_0^{(k)}, y_0^{(k)})\|] + \frac{4\sqrt{2}K}{B_0^{1-1/s}} \frac{1}{\delta} \sigma \ . \qquad (112)$$

By combining them together, we obtain

$$\sum_{k=1}^{K} \mathbb{E}[\|v_0^{(k)} - \bar{v}_0\|]$$

$$\leq 2 \sum_{k=1}^{K} \mathbb{E}[\|\nabla_2 f^{(k)}(x_0^{(k)}, y_0^{(k)})\|] + \frac{2}{\delta} \sum_{k=1}^{K} \mathbb{E}[\|\nabla_2 g^{(k)}(x_0^{(k)}, y_0^{(k)})\|] + \frac{4\sqrt{2}K}{B_0^{1-1/s}} \left(1 + \frac{1}{\delta}\right) \sigma \ .$$

$$(113)$$

Finally, we obtain

$$\frac{1}{T} \sum_{t=0}^{T-1} \frac{1}{K} \sum_{k=1}^{K} \mathbb{E}[\|q_t^{(k)} - \bar{q}_t\|]$$

$$\leq \frac{2\lambda}{(1-\lambda)T} \frac{1}{\sqrt{K}} \sum_{k=1}^{K} \mathbb{E}[\|\nabla_2 f^{(k)}(x_0^{(k)}, y_0^{(k)})\|] + \frac{2\lambda}{(1-\lambda)T} \frac{1}{\delta} \frac{1}{\sqrt{K}} \sum_{k=1}^{K} \mathbb{E}[\|\nabla_2 g^{(k)}(x_0^{(k)}, y_0^{(k)})\|]$$

$$+ \frac{\lambda}{(1-\lambda)T} \frac{4\sqrt{2}\sqrt{K}}{B_0^{1-1/s}} \left(1 + \frac{1}{\delta}\right) \sigma + \frac{\gamma_y \lambda \sqrt{K} \sigma}{(1-\lambda)^{3/2}} \left(1 + \frac{1}{\delta}\right)$$

$$+ \frac{4\eta_x \lambda \sqrt{K}}{(1-\lambda)^{5/2}} \left(L_f + \frac{L_g}{\delta}\right) + \frac{4\eta_y \lambda \sqrt{K}}{(1-\lambda)^{5/2}} \left(L_f + \frac{L_g}{\delta}\right)$$

$$+ \frac{\lambda\sqrt{K}}{T(1-\lambda)^{3/2}} \frac{2\sqrt{2}}{B_0^{1-1/s}} \sigma \left(1 + \frac{1}{\delta}\right) + \frac{2\sqrt{2}\gamma_y^{2-1/s}\lambda\sqrt{K}}{(1-\lambda)^{3/2}} \sigma \left(1 + \frac{1}{\delta}\right)$$

$$+ \frac{4\eta_x\sqrt{\gamma_y}\lambda}{(1-\lambda)^{5/2}} \left(L_f + \frac{L_g}{\delta}\right) + \frac{4\eta_y\sqrt{\gamma_y}\lambda}{(1-\lambda)^{5/2}} \left(L_f + \frac{L_g}{\delta}\right) . \tag{114}$$

$\square$

**Lemma C.31.** *Given Assumptions 3.1-3.3, we obtain*

$$\frac{1}{T}\sum_{t=0}^{T-1}\frac{1}{K}\sum_{k=1}^{K}\mathbb{E}[\|r_t^{(k)} - \bar{r}_t\|]$$

$$\leq \frac{2\lambda}{(1-\lambda)T}\frac{1}{\sqrt{K}}\frac{1}{\delta}\sum_{k=1}^{K}\mathbb{E}[\|\nabla_2 g^{(k)}(x_0^{(k)}, z_0^{(k)})\|] + \frac{\lambda}{(1-\lambda)T}\frac{4\sqrt{2}\sqrt{K}}{B_0^{1-1/s}}\frac{1}{\delta}\sigma + \frac{\gamma_z\lambda\sqrt{K}\sigma}{(1-\lambda)^{3/2}}\frac{1}{\delta}$$

$$+ \frac{4\eta_x\lambda\sqrt{K}}{(1-\lambda)^{5/2}}\frac{L_g}{\delta} + \frac{4\eta_y\lambda\sqrt{K}}{(1-\lambda)^{5/2}}\frac{L_g}{\delta} + \frac{\lambda\sqrt{K}}{T(1-\lambda)^{3/2}}\frac{2\sqrt{2}}{B_0^{1-1/s}}\frac{1}{\delta}\sigma + \frac{2\sqrt{2}\gamma_z^{2-1/s}\lambda\sqrt{K}}{(1-\lambda)^{3/2}}\frac{1}{\delta}\sigma$$

$$+ \frac{4\eta_x\sqrt{\gamma_z}\lambda}{(1-\lambda)^{5/2}}\frac{L_g}{\delta} + \frac{4\eta_z\sqrt{\gamma_z}\lambda}{(1-\lambda)^{5/2}}\frac{L_g}{\delta} . \tag{115}$$

*Proof.* Following the proof of Lemma C.29, we obtain

$$\frac{1}{K}\sum_{k=1}^{K}\mathbb{E}[\|r_t^{(k)} - \bar{r}_t\|]$$

$$\leq \lambda^{1+t/2}\frac{1}{\sqrt{K}}\sum_{k=1}^{K}\mathbb{E}[\|w_0^{(k)} - \bar{w}_0\|] + \sum_{j=1}^{t}\lambda^{(t-j)/2}\frac{\lambda}{\sqrt{1-\lambda}}\frac{1}{\sqrt{K}}\sum_{k=1}^{K}\mathbb{E}[\|w_j^{(k)} - w_{j-1}^{(k)}\|]$$

$$\leq \lambda^{1+t/2}\frac{1}{\sqrt{K}}\sum_{k=1}^{K}\mathbb{E}[\|w_0^{(k)} - \bar{w}_0\|] + \sum_{j=1}^{t}\lambda^{(t-j)/2}\frac{\lambda}{\sqrt{1-\lambda}}\frac{1}{\delta}\frac{1}{\sqrt{K}}\sum_{k=1}^{K}\mathbb{E}[\|w_{1,j}^{(k)} - w_{1,j-1}^{(k)}\|] . \tag{116}$$

Based on Lemma C.15, we obtain

$$\frac{1}{K}\sum_{k=1}^{K}\mathbb{E}[\|r_t^{(k)} - \bar{r}_t\|]$$

$$\leq \lambda^{1+t/2}\frac{1}{\sqrt{K}}\sum_{k=1}^{K}\mathbb{E}[\|w_0^{(k)} - \bar{w}_0\|]$$

$$+ \gamma_z\sum_{j=1}^{t}\lambda^{(t-j)/2}\frac{\lambda}{\sqrt{1-\lambda}}\frac{1}{\delta}\frac{1}{\sqrt{K}}\sum_{k=1}^{K}\mathbb{E}[\|\nabla_2 g^{(k)}(x_{j-1}^{(k)}, z_{j-1}^{(k)}) - \nabla_2 g^{(k)}(x_{j-1}^{(k)}, z_{j-1}^{(k)}; \zeta_j^{(k)})\|]$$

$$+ \gamma_z\sum_{j=1}^{t}\lambda^{(t-j)/2}\frac{\lambda}{\sqrt{1-\lambda}}\frac{1}{\delta}\frac{1}{\sqrt{K}}\sum_{k=1}^{K}\mathbb{E}[\|w_{1,j-1}^{(k)} - \nabla_2 g^{(k)}(x_{j-1}^{(k)}, z_{j-1}^{(k)})\|]$$

$$+ \frac{4\eta_x\sqrt{K}}{1-\lambda}\frac{\lambda}{\sqrt{1-\lambda}}\frac{L_g}{\delta}\sum_{j=1}^{t}\lambda^{(t-j)/2} + \frac{4\eta_y\sqrt{K}}{1-\lambda}\frac{\lambda}{\sqrt{1-\lambda}}\frac{L_g}{\delta}\sum_{j=1}^{t}\lambda^{(t-j)/2} . \tag{117}$$

Then, we obtain

$$\frac{1}{T}\sum_{t=0}^{T-1}\frac{1}{K}\sum_{k=1}^{K}\mathbb{E}[\|r_t^{(k)} - \bar{r}_t\|]$$

$$\leq \frac{1}{T} \sum_{t=0}^{T-1} \lambda^{1+t/2} \frac{1}{\sqrt{K}} \sum_{k=1}^{K} \mathbb{E}[\|w_0^{(k)} - \bar{w}_0\|]$$

$$+ \gamma_z \frac{1}{T} \sum_{t=0}^{T-1} \sum_{j=1}^{t} \lambda^{(t-j)/2} \frac{\lambda}{\sqrt{1-\lambda}} \frac{1}{\delta} \frac{1}{\sqrt{K}} \sum_{k=1}^{K} \mathbb{E}[\|\nabla_2 g^{(k)}(x_{j-1}^{(k)}, z_{j-1}^{(k)}) - \nabla_2 g^{(k)}(x_{j-1}^{(k)}, z_{j-1}^{(k)}; \zeta_j^{(k)})\|]$$

$$+ \gamma_z \frac{1}{T} \sum_{t=0}^{T-1} \sum_{j=1}^{t} \lambda^{(t-j)/2} \frac{\lambda}{\sqrt{1-\lambda}} \frac{1}{\delta} \frac{1}{\sqrt{K}} \sum_{k=1}^{K} \mathbb{E}[\|w_{1,j-1}^{(k)} - \nabla_2 g^{(k)}(x_{j-1}^{(k)}, z_{j-1}^{(k)})\|]$$

$$+ \frac{4\eta_x \sqrt{K}}{1-\lambda} \frac{\lambda}{\sqrt{1-\lambda}} \frac{L_g}{\delta} \frac{1}{T} \sum_{t=0}^{T-1} \sum_{j=1}^{t} \lambda^{(t-j)/2} + \frac{4\eta_y \sqrt{K}}{1-\lambda} \frac{\lambda}{\sqrt{1-\lambda}} \frac{L_g}{\delta} \frac{1}{T} \sum_{t=0}^{T-1} \sum_{j=1}^{t} \lambda^{(t-j)/2}$$

$$\leq \frac{\lambda}{(1-\lambda)T} \frac{1}{\sqrt{K}} \sum_{k=1}^{K} \mathbb{E}[\|w_0^{(k)} - \bar{w}_0\|] + \frac{\gamma_z \lambda \sqrt{K} \sigma}{(1-\lambda)^{3/2}} \frac{1}{\delta}$$

$$+ \frac{4\eta_x \lambda \sqrt{K}}{(1-\lambda)^{5/2}} \frac{L_g}{\delta} + \frac{4\eta_y \lambda \sqrt{K}}{(1-\lambda)^{5/2}} \frac{L_g}{\delta} + \frac{\lambda \sqrt{K}}{T(1-\lambda)^{3/2}} \frac{2\sqrt{2}}{B_0^{1-1/s}} \frac{1}{\delta} \sigma + \frac{2\sqrt{2}\gamma_z^{2-1/s} \lambda \sqrt{K}}{(1-\lambda)^{3/2}} \frac{1}{\delta} \sigma$$

$$+ \frac{4\eta_x \sqrt{\gamma_z} \lambda}{(1-\lambda)^{5/2}} \frac{L_g}{\delta} + \frac{4\eta_z \sqrt{\gamma_z} \lambda}{(1-\lambda)^{5/2}} \frac{L_g}{\delta} . \tag{118}$$

For $\sum_{k=1}^{K} \mathbb{E}[\|w_0^{(k)} - \bar{w}_0\|]$, we bound it as follows:

$$\sum_{k=1}^{K} \mathbb{E}[\|w_0^{(k)} - \bar{w}_0\|] = \sum_{k=1}^{K} \mathbb{E}[\|\frac{1}{\delta} w_{1,0}^{(k)} - \frac{1}{K} \sum_{j=1}^{K} \frac{1}{\delta} w_{1,0}^{(j)}\|]$$

$$= \frac{1}{\delta} \sum_{k=1}^{K} \mathbb{E}[\|\nabla_2 g^{(k)}(x_0^{(k)}, z_0^{(k)}; \zeta_0^{(k)}) - \frac{1}{K} \sum_{j=1}^{K} \nabla_2 g^{(j)}(x_0^{(j)}, z_0^{(j)}; \zeta_0^{(j)})\|]$$

$$\leq \frac{2}{\delta} \sum_{k=1}^{K} \mathbb{E}[\|\nabla_2 g^{(k)}(x_0^{(k)}, z_0^{(k)})\|] + \frac{4\sqrt{2}K}{B_0^{1-1/s}} \frac{1}{\delta} \sigma . \tag{119}$$

Finally, we obtain

$$\frac{1}{T} \sum_{t=0}^{T-1} \frac{1}{K} \sum_{k=1}^{K} \mathbb{E}[\|r_t^{(k)} - \bar{r}_t\|]$$

$$\leq \frac{\lambda}{(1-\lambda)T} \frac{1}{\sqrt{K}} \frac{2}{\delta} \sum_{k=1}^{K} \mathbb{E}[\|\nabla_2 g^{(k)}(x_0^{(k)}, z_0^{(k)})\|] + \frac{\lambda}{(1-\lambda)T} \frac{4\sqrt{2}\sqrt{K}}{B_0^{1-1/s}} \frac{1}{\delta} \sigma + \frac{\gamma_z \lambda \sqrt{K} \sigma}{(1-\lambda)^{3/2}} \frac{1}{\delta}$$

$$+ \frac{4\eta_x \lambda \sqrt{K}}{(1-\lambda)^{5/2}} \frac{L_g}{\delta} + \frac{4\eta_y \lambda \sqrt{K}}{(1-\lambda)^{5/2}} \frac{L_g}{\delta} + \frac{\lambda \sqrt{K}}{T(1-\lambda)^{3/2}} \frac{2\sqrt{2}}{B_0^{1-1/s}} \frac{1}{\delta} \sigma + \frac{2\sqrt{2}\gamma_z^{2-1/s} \lambda \sqrt{K}}{(1-\lambda)^{3/2}} \frac{1}{\delta} \sigma$$

$$+ \frac{4\eta_x \sqrt{\gamma_z} \lambda}{(1-\lambda)^{5/2}} \frac{L_g}{\delta} + \frac{4\eta_z \sqrt{\gamma_z} \lambda}{(1-\lambda)^{5/2}} \frac{L_g}{\delta} . \tag{120}$$

$\square$

## C.6 Proof of Theorem 4.1

*Proof.* By plugging the inequalities in Lemmas C.8, C.7 into Lemma C.6, we obtain

$$\frac{1}{T} \sum_{t=0}^{T-1} \mathbb{E}[\|\nabla\Phi(\bar{x}_t)\|]$$

$$\leq \frac{\mathbb{E}[\Phi(\bar{x}_0) - \Phi(\bar{x}_T)]}{\eta_x T} + 2\frac{1}{T} \sum_{t=0}^{T-1} \mathbb{E}[\|\nabla\Phi(\bar{x}_t) - \nabla\Phi_\delta(\bar{x}_t)\|]$$

$$+ \frac{4(\delta L_f + L_g)}{\mu} \frac{(\frac{1}{\delta}\mathbb{E}[h_\delta(\bar{x}_0, \bar{y}_0) - h_\delta^*(\bar{x}_0)] - \frac{1}{\delta}\mathbb{E}[h_\delta(\bar{x}_T, \bar{y}_T) - h_\delta^*(\bar{x}_T)])}{\eta_y T}$$

$$+ \frac{4L_g}{\mu} \frac{(\frac{1}{\delta}\mathbb{E}[g(\bar{x}_0, \bar{z}_0) - g^*(\bar{x}_0)] - \frac{1}{\delta}\mathbb{E}[g(\bar{x}_T, \bar{z}_T) - g^*(\bar{x}_T)])}{\eta_z T}$$

$$+ 2\frac{1}{T} \sum_{t=0}^{T-1} \mathbb{E}[\|\frac{1}{K}\sum_{k=1}^{K} \nabla_1 f^{(k)}(x_t^{(k)}, y_t^{(k)}) - \frac{1}{K}\sum_{k=1}^{K} u_{1,t}^{(k)}\|]$$

$$+ 2\frac{1}{\delta}\frac{1}{T} \sum_{t=0}^{T-1} \mathbb{E}[\|\frac{1}{K}\sum_{k=1}^{K} \nabla_1 g^{(k)}(x_t^{(k)}, y_t^{(k)}) - \frac{1}{K}\sum_{k=1}^{K} u_{2,t}^{(k)}\|]$$

$$+ 2\frac{1}{\delta}\frac{1}{T} \sum_{t=0}^{T-1} \mathbb{E}[\|\frac{1}{K}\sum_{k=1}^{K} \nabla_1 g^{(k)}(x_t^{(k)}, z_t^{(k)}) - \frac{1}{K}\sum_{k=1}^{K} u_{3,t}^{(k)}\|]$$

$$+ \frac{8(\delta L_f + L_g)}{\mu}\frac{1}{T} \sum_{t=0}^{T-1} \mathbb{E}[\|\frac{1}{K}\sum_{k=1}^{K} \nabla_2 f^{(k)}(x_t^{(k)}, y_t^{(k)}) - \frac{1}{K}\sum_{k=1}^{K} v_{1,t}^{(k)}\|]$$

$$+ \frac{8(\delta L_f + L_g)}{\mu}\frac{1}{\delta}\frac{1}{T} \sum_{t=0}^{T-1} \mathbb{E}[\|\frac{1}{K}\sum_{k=1}^{K} \nabla_2 g^{(k)}(x_t^{(k)}, y_t^{(k)}) - \frac{1}{K}\sum_{k=1}^{K} v_{2,t}^{(k)}\|]$$

$$+ \frac{8L_g}{\mu}\frac{1}{\delta}\frac{1}{T} \sum_{t=0}^{T-1} \mathbb{E}[\|\frac{1}{K}\sum_{k=1}^{K} \nabla_2 g^{(k)}(x_t^{(k)}, z_t^{(k)}) - \frac{1}{K}\sum_{k=1}^{K} w_{1,t}^{(k)}\|]$$

$$+ \left(2(L_f + \frac{2L_g}{\delta}) + \frac{8(\delta L_f + L_g)}{\mu}\left(L_f + \frac{L_g}{\delta}\right) + \frac{8L_g^2}{\mu}\frac{1}{\delta}\right)\frac{1}{T}\sum_{t=0}^{T-1}\frac{1}{K}\sum_{k=1}^{K}\mathbb{E}[\|\bar{x}_t - x_t^{(k)}\|]$$

$$+ \left(2(L_f + \frac{L_g}{\delta}) + \frac{8(\delta L_f + L_g)}{\mu}\left(L_f + \frac{L_g}{\delta}\right)\right)\frac{1}{T}\sum_{t=0}^{T-1}\frac{1}{K}\sum_{k=1}^{K}\mathbb{E}[\|\bar{y}_t - y_t^{(k)}\|]$$

$$+ \left(2\frac{L_g}{\delta} + \frac{8L_g^2}{\mu}\frac{1}{\delta}\right)\frac{1}{T}\sum_{t=0}^{T-1}\frac{1}{K}\sum_{k=1}^{K}\mathbb{E}[\|z_t^{(k)} - \bar{z}_t\|]$$

$$+ \frac{1}{T}\sum_{t=0}^{T-1}\frac{1}{K}\sum_{k=1}^{K}\mathbb{E}[\|\bar{p}_t - p_t^{(k)}\|] + \frac{4(\delta L_f + L_g)}{\mu}\frac{1}{T}\sum_{t=0}^{T-1}\frac{1}{K}\sum_{k=1}^{K}\mathbb{E}[\|\bar{q}_t - q_t^{(k)}\|] + \frac{4L_g}{\mu}\frac{1}{T}\sum_{t=0}^{T-1}\frac{1}{K}\sum_{k=1}^{K}\mathbb{E}[\|\bar{r}_t - r_t^{(k)}\|]$$

$$+ \frac{\eta_x L_\Phi}{2} + \frac{1}{\delta}\frac{4\eta_x(\delta L_f + L_g)^2}{\mu} + \frac{1}{\delta}\frac{2\eta_y(\delta L_f + L_g)^2}{\mu} + \frac{1}{\delta}\frac{\eta_x^2}{\eta_y}\frac{2(\delta L_f + L_g)^2}{\mu} + \frac{1}{\delta}\frac{\eta_x^2}{\eta_y}\frac{2L_{h_\delta^*}(\delta L_f + L_g)}{\mu}$$

$$+ \frac{1}{\delta}\frac{4\eta_x L_g^2}{\mu} + \frac{1}{\delta}\frac{2\eta_z L_g^2}{\mu} + \frac{1}{\delta}\frac{\eta_x^2}{\eta_z}\frac{2L_g^2}{\mu} + \frac{1}{\delta}\frac{\eta_x^2}{\eta_z}\frac{2L_{g^*}L_g}{\mu} . \tag{121}$$

By plugging Lemmas C.22, C.23, C.24, C.25, C.26, C.27, C.28, C.29, C.30, C.31 into the above inequality and setting $\eta_y = \eta_x \frac{4(\delta L_f + L_g)}{\mu}$ and $\eta_z = \eta_x \frac{4L_g}{\mu}$, we obtain

$$\frac{1}{T}\sum_{t=0}^{T-1}\mathbb{E}[\|\nabla\Phi(\bar{x}_t)\|] \le \frac{\mathbb{E}[\Phi(\bar{x}_0) - \Phi(\bar{x}_T)]}{\eta_x T} + 2\frac{1}{T}\sum_{t=0}^{T-1}\mathbb{E}[\|\nabla\Phi(\bar{x}_t) - \nabla\Phi_\delta(\bar{x}_t)\|]$$

$$+ \frac{1}{\delta}\frac{\mathbb{E}[(h_\delta(\bar{x}_0, \bar{y}_0) - h_\delta^*(\bar{x}_0)) - (h_\delta(\bar{x}_T, \bar{y}_T) - h_\delta^*(\bar{x}_T))]}{\eta_x T}$$

$$+ \frac{1}{\delta}\frac{\mathbb{E}[(g(\bar{x}_0, \bar{z}_0) - g^*(\bar{x}_0)) - (g(\bar{x}_T, \bar{z}_T) - g^*(\bar{x}_T))]}{\eta_x T}$$

$$+ \frac{\eta_x L_\Phi}{2} + \frac{1}{\delta}\frac{4\eta_x(\delta L_f + L_g)^2}{\mu} + \frac{1}{\delta}\frac{8\eta_x(\delta L_f + L_g)^3}{\mu^2} + \frac{1}{\delta}\frac{\eta_x(\delta L_f + L_g)}{2} + \frac{1}{\delta}\frac{\eta_x L_{h_\delta^*}}{2}$$

$$+ \frac{1}{\delta}\frac{4\eta_x L_g^2}{\mu} + \frac{1}{\delta}\frac{8\eta_x L_g^3}{\mu^2} + \frac{1}{\delta}\frac{\eta_x L_g}{2} + \frac{1}{\delta}\frac{\eta_x L_{g^*}}{2}$$

$$+ \frac{2\lambda}{(1-\lambda)T} \frac{1}{\sqrt{K}} \sum_{k=1}^{K} \mathbb{E}[\|\nabla_1 f^{(k)}(x_0^{(k)}, y_0^{(k)})\|] + \frac{2\lambda}{(1-\lambda)T} \frac{1}{\delta} \frac{1}{\sqrt{K}} \sum_{k=1}^{K} \mathbb{E}[\|\nabla_1 g^{(k)}(x_0^{(k)}, y_0^{(k)})\|]$$

$$+ \frac{2\lambda}{(1-\lambda)T} \frac{1}{\delta} \frac{1}{\sqrt{K}} \sum_{k=1}^{K} \mathbb{E}[\|\nabla_1 g^{(k)}(x_0^{(k)}, z_0^{(k)})\|] + \frac{4(\delta L_f + L_g)}{\mu} \frac{2\lambda}{(1-\lambda)T} \frac{1}{\sqrt{K}} \sum_{k=1}^{K} \mathbb{E}[\|\nabla_2 f^{(k)}(x_0^{(k)}, y_0^{(k)})\|]$$

$$+ \frac{4(\delta L_f + L_g)}{\mu} \frac{2\lambda}{(1-\lambda)T} \frac{1}{\delta} \frac{1}{\sqrt{K}} \sum_{k=1}^{K} \mathbb{E}[\|\nabla_2 g^{(k)}(x_0^{(k)}, y_0^{(k)})\|] + \frac{4L_g}{\mu} \frac{2\lambda}{(1-\lambda)T} \frac{1}{\sqrt{K}} \frac{1}{\delta} \sum_{k=1}^{K} \mathbb{E}[\|\nabla_2 g^{(k)}(x_0^{(k)}, z_0^{(k)})\|]$$

$$+ \left(1 + \frac{2}{\delta}\right) \frac{4\sqrt{2}}{K^{1-1/s}} \gamma_x^{1-1/s}\sigma + \left(1 + \frac{1}{\delta}\right) \frac{8(\delta L_f + L_g)}{\mu} \frac{2\sqrt{2}}{K^{1-1/s}} \gamma_y^{1-1/s}\sigma + \frac{1}{\delta} \frac{8L_g}{\mu} \frac{2\sqrt{2}}{K^{1-1/s}} \gamma_z^{1-1/s}\sigma$$

$$+ \left(1 + \frac{2}{\delta}\right) \frac{1}{\gamma_x T} \frac{4\sqrt{2}}{B_0^{1-1/s}}\sigma + \left(1 + \frac{1}{\delta}\right) \frac{8(\delta L_f + L_g)}{\mu} \frac{1}{\gamma_y T} \frac{2\sqrt{2}}{B_0^{1-1/s}}\sigma + \frac{1}{\delta} \frac{8L_g}{\mu} \frac{1}{\gamma_z T} \frac{2\sqrt{2}}{B_0^{1-1/s}}\sigma$$

$$+ \frac{\lambda}{(1-\lambda)T} \frac{4\sqrt{2}\sqrt{K}}{B_0^{1-1/s}} \left(\left(1 + \frac{2}{\delta}\right) + \frac{4(\delta L_f + L_g)}{\mu} \left(1 + \frac{1}{\delta}\right) + \frac{4L_g}{\mu} \frac{1}{\delta}\right)\sigma$$

$$+ \frac{\lambda\sqrt{K}}{T(1-\lambda)^{3/2}} \frac{2\sqrt{2}\sigma}{B_0^{1-1/s}} \left(\left(1 + \frac{2}{\delta}\right) + \frac{4(\delta L_f + L_g)}{\mu} \left(1 + \frac{1}{\delta}\right) + \frac{4L_g}{\mu} \frac{1}{\delta}\right)$$

$$+ \frac{\gamma_x \lambda\sqrt{K}\sigma}{(1-\lambda)^{3/2}} \left(1 + \frac{2}{\delta}\right) + \frac{\gamma_y \lambda\sqrt{K}\sigma}{(1-\lambda)^{3/2}} \frac{4(\delta L_f + L_g)}{\mu} \left(1 + \frac{1}{\delta}\right) + \frac{\gamma_z \lambda\sqrt{K}\sigma}{(1-\lambda)^{3/2}} \frac{4L_g}{\mu} \frac{1}{\delta}$$

$$+ \frac{2\sqrt{2}\gamma_x^{2-1/s}\lambda\sqrt{K}}{(1-\lambda)^{3/2}}\sigma \left(1 + \frac{2}{\delta}\right) + \frac{2\sqrt{2}\gamma_y^{2-1/s}\lambda\sqrt{K}}{(1-\lambda)^{3/2}} \frac{4(\delta L_f + L_g)}{\mu} \left(1 + \frac{1}{\delta}\right)\sigma + \frac{2\sqrt{2}\gamma_z^{2-1/s}\lambda\sqrt{K}}{(1-\lambda)^{3/2}} \frac{4L_g}{\mu} \frac{1}{\delta}\sigma$$

$$+ \frac{8\eta_x}{(1-\lambda)\sqrt{\gamma_x K}} \left(L_f + \frac{2L_g}{\delta} + \frac{4((\delta L_f + L_g)^2 + L_g^2)}{\mu} \frac{1}{\delta}\right)$$

$$+ \frac{4\eta_x}{(1-\lambda)\sqrt{\gamma_y K}} \left(1 + \frac{4(\delta L_f + L_g)}{\mu}\right) \frac{8(\delta L_f + L_g)^2}{\mu} \frac{1}{\delta} + \frac{4\eta_x}{(1-\lambda)\sqrt{\gamma_z K}} \left(1 + \frac{4L_g}{\mu}\right) \frac{8L_g^2}{\mu} \frac{1}{\delta}$$

$$+ \eta_x \left(2(L_f + \frac{2L_g}{\delta}) + \frac{8(\delta L_f + L_g)}{\mu} \left(L_f + \frac{L_g}{\delta}\right) + \frac{8L_g}{\mu} \frac{L_g}{\delta}\right) \frac{\lambda}{1-\lambda}$$

$$+ \eta_x \left(2(L_f + \frac{L_g}{\delta}) + \frac{8(\delta L_f + L_g)}{\mu} \left(L_f + \frac{L_g}{\delta}\right)\right) \frac{\lambda}{1-\lambda} \frac{4(\delta L_f + L_g)}{\mu}$$

$$+ \eta_x \left(\frac{2L_g}{\delta} + \frac{8L_g}{\mu} \frac{L_g}{\delta}\right) \frac{\lambda}{1-\lambda} \frac{4L_g}{\mu}$$

$$+ \frac{4\eta_x \lambda\sqrt{K}}{(1-\lambda)^{5/2}} \left(L_f + \frac{2L_g}{\delta} + \frac{1}{\delta} \frac{4(\delta L_f + L_g)^2}{\mu} + \frac{1}{\delta} \frac{4L_g^2}{\mu}\right)$$

$$+ \frac{4\eta_x \lambda\sqrt{K}}{(1-\lambda)^{5/2}} \frac{4(\delta L_f + L_g)}{\mu} \left(L_f + \frac{L_g}{\delta}\right) \left(1 + \frac{4(\delta L_f + L_g)}{\mu}\right)$$

$$+ \frac{4\eta_x \lambda\sqrt{K}}{(1-\lambda)^{5/2}} \frac{4L_g}{\mu} \frac{L_g}{\delta} \left(1 + \frac{4(\delta L_f + L_g)}{\mu}\right)$$

$$+ \frac{4\eta_x \sqrt{\gamma_x}\lambda}{(1-\lambda)^{5/2}} \left(L_f + \frac{2L_g}{\delta}\right) \left(1 + \frac{4(\delta L_f + L_g)}{\mu}\right)$$

$$+ \frac{4\eta_x \sqrt{\gamma_y}\lambda}{(1-\lambda)^{5/2}} \frac{4(\delta L_f + L_g)}{\mu} \left(L_f + \frac{L_g}{\delta}\right) \left(1 + \frac{4(\delta L_f + L_g)}{\mu}\right)$$

$$+ \frac{4\eta_x \sqrt{\gamma_z}\lambda}{(1-\lambda)^{5/2}} \frac{4L_g}{\mu} \frac{L_g}{\delta} . \tag{122}$$

Because $\kappa > 1$, $1 - \lambda < 1$, $\gamma_x < 1$, $\gamma_y < 1$, $\gamma_z < 1$, $s \in (1, 2]$, $L_\Phi = O(\ell \kappa^3)$, $L_{h_\delta^*} = O(\ell \kappa)$, and $L_{g^*} = O(\ell \kappa)$, it can be simplified to the following inequality:

$$
\frac{1}{T} \sum_{t=0}^{T-1} \mathbb{E}[\|\nabla \Phi(\bar{x}_t)\|] \leq \frac{\mathbb{E}[\Phi(\bar{x}_0) - \Phi(\bar{x}_T)]}{\eta_x T} + 2\frac{1}{T} \sum_{t=0}^{T-1} \mathbb{E}[\|\nabla \Phi(\bar{x}_t) - \nabla \Phi_\delta(\bar{x}_t)\|]
$$

$$
+ \frac{1}{\delta} \frac{\mathbb{E}[(h_\delta(\bar{x}_0, \bar{y}_0) - h_\delta^*(\bar{x}_0)) - (h_\delta(\bar{x}_T, \bar{y}_T) - h_\delta^*(\bar{x}_T))]}{\eta_x T}
$$

$$
+ \frac{1}{\delta} \frac{\mathbb{E}[(g(\bar{x}_0, \bar{z}_0) - g^*(\bar{x}_0)) - (g(\bar{x}_T, \bar{z}_T) - g^*(\bar{x}_T))]}{\eta_x T}
$$

$$
+ O\left(\frac{\lambda}{(1-\lambda)T}\right) \frac{1}{\sqrt{K}} \sum_{k=1}^{K} \mathbb{E}[\|\nabla_1 f^{(k)}(x_0^{(k)}, y_0^{(k)})\|] + O\left(\frac{\lambda}{(1-\lambda)T} \frac{1}{\delta}\right) \frac{1}{\sqrt{K}} \sum_{k=1}^{K} \mathbb{E}[\|\nabla_1 g^{(k)}(x_0^{(k)}, y_0^{(k)})\|]
$$

$$
+ O\left(\frac{\lambda}{(1-\lambda)T} \frac{1}{\delta}\right) \frac{1}{\sqrt{K}} \sum_{k=1}^{K} \mathbb{E}[\|\nabla_1 g^{(k)}(x_0^{(k)}, z_0^{(k)})\|] + O\left(\frac{\lambda \kappa}{(1-\lambda)T}\right) \frac{1}{\sqrt{K}} \sum_{k=1}^{K} \mathbb{E}[\|\nabla_2 f^{(k)}(x_0^{(k)}, y_0^{(k)})\|]
$$

$$
+ O\left(\frac{\lambda \kappa}{(1-\lambda)T} \frac{1}{\delta}\right) \frac{1}{\sqrt{K}} \sum_{k=1}^{K} \mathbb{E}[\|\nabla_2 g^{(k)}(x_0^{(k)}, y_0^{(k)})\|] + O\left(\frac{\lambda \kappa}{(1-\lambda)T} \frac{1}{\delta}\right) \frac{1}{\sqrt{K}} \sum_{k=1}^{K} \mathbb{E}[\|\nabla_2 g^{(k)}(x_0^{(k)}, z_0^{(k)})\|]
$$

$$
+ O\left(\eta_x \kappa^3 \ell\right) + O\left(\eta_x \frac{\kappa^2 \ell}{\delta}\right) + O\left(\eta_x \frac{\kappa^2 \ell}{\delta} \frac{\lambda \sqrt{K}}{(1-\lambda)^{5/2}}\right)
$$

$$
+ O\left(\frac{1}{\delta} \frac{\gamma_x^{1-1/s}}{K^{1-1/s}} \sigma\right) + O\left(\frac{\kappa}{\delta} \frac{\gamma_y^{1-1/s}}{K^{1-1/s}} \sigma\right) + O\left(\frac{\kappa}{\delta} \frac{\gamma_z^{1-1/s}}{K^{1-1/s}} \sigma\right)
$$

$$
+ O\left(\frac{\eta_x}{(1-\lambda)\sqrt{\gamma_x K}} \frac{\kappa \ell}{\delta}\right) + O\left(\frac{\eta_x}{(1-\lambda)\sqrt{\gamma_y K}} \frac{\kappa^2 \ell}{\delta}\right) + O\left(\frac{\eta_x}{(1-\lambda)\sqrt{\gamma_z K}} \frac{\kappa^2 \ell}{\delta}\right)
$$

$$
+ O\left(\frac{1}{\delta} \frac{1}{\gamma_x T} \frac{1}{B_0^{1-1/s}} \sigma\right) + O\left(\frac{\kappa}{\delta} \frac{1}{\gamma_y T} \frac{1}{B_0^{1-1/s}} \sigma\right) + O\left(\frac{\kappa}{\delta} \frac{1}{\gamma_z T} \frac{1}{B_0^{1-1/s}} \sigma\right)
$$

$$
+ O\left(\frac{1}{\delta} \frac{\gamma_x \lambda \sqrt{K}}{(1-\lambda)^{3/2}} \sigma\right) + O\left(\frac{\kappa}{\delta} \frac{\gamma_y \lambda \sqrt{K}}{(1-\lambda)^{3/2}} \sigma\right) + O\left(\frac{\kappa}{\delta} \frac{\gamma_z \lambda \sqrt{K}}{(1-\lambda)^{3/2}} \sigma\right)
$$

$$
+ O\left(\frac{\eta_x \sqrt{\gamma_x} \lambda}{(1-\lambda)^{5/2}} \frac{\kappa \ell}{\delta}\right) + O\left(\frac{\eta_x \sqrt{\gamma_y} \lambda}{(1-\lambda)^{5/2}} \frac{\kappa^2 \ell}{\delta}\right) + O\left(\frac{\eta_x \sqrt{\gamma_z} \lambda}{(1-\lambda)^{5/2}} \frac{\kappa^2 \ell}{\delta}\right)
$$

$$
+ O\left(\frac{\kappa}{\delta} \frac{\lambda}{(1-\lambda)^{3/2} T} \frac{\sqrt{K}}{B_0^{1-1/s}} \sigma\right). \tag{123}
$$

We set

$$
\delta = O\left(\frac{1}{\kappa^3 \ell} \frac{1}{K^{\frac{s-1}{2s+1}} T^{\frac{s-1}{2s+1}}}\right),
$$

$$
\gamma_x = \gamma_y = \gamma_z = O\left(\frac{K^{\frac{1}{2s+1}}}{T^{\frac{2s}{2s+1}} \sigma^{\frac{3s}{(2s+1)(s-1)}}}\right),
$$

$$
\eta_x = O\left(\frac{1-\lambda}{\kappa^5 \ell} \frac{K^{\frac{1}{2s+1}}}{T^{\frac{2s}{2s+1}} \sigma^{\frac{4-s}{2(2s+1)(s-1)}}}\right),
$$

$$
B_0 = O\left(K^{\frac{2s}{2s+1}} T^{\frac{2s}{2s+1}} \sigma^{\frac{s(4s-1)}{(2s+1)(s-1)^2}}\right). \tag{124}
$$

Then, we can obtain

$$
O\left(\frac{1}{\delta} \frac{\gamma_x^{1-1/s}}{K^{1-1/s}} \sigma\right) = O\left(\frac{\kappa^3 \ell}{1} \frac{K^{\frac{s-1}{2s+1}} T^{\frac{s-1}{2s+1}}}{1} \frac{K^{\frac{s-1}{s(2s+1)}}}{T^{\frac{2s}{2s+1} \times \frac{s-1}{s}} \sigma^{\frac{3s}{(2s+1)(s-1)} \times \frac{s-1}{s}}} \frac{1}{K^{1-1/s}} \sigma\right)
$$

$$= O\left(\frac{\kappa^3 \ell \sigma^{\frac{2s-2}{2s+1}}}{K^{\frac{s-1}{2s+1}} T^{\frac{s-1}{2s+1}}}\right),$$

$$O\left(\frac{\eta_x}{(1-\lambda)\sqrt{\gamma_x K}}\frac{\kappa\ell}{\delta}\right) = O\left(\frac{1-\lambda}{\kappa^5 \ell}\frac{K^{\frac{1}{2s+1}}}{T^{\frac{2s}{2s+1}}\sigma^{\frac{4-s}{2(2s+1)(s-1)}}}\frac{T^{\frac{s}{2s+1}}\sigma^{\frac{3s}{2(2s+1)(s-1)}}}{K^{\frac{1}{2(2s+1)}}}\frac{1}{(1-\lambda)K^{\frac{1}{2}}}\kappa^4\ell^2 K^{\frac{s-1}{2s+1}}T^{\frac{s-1}{2s+1}}\right)$$

$$= O\left(\frac{\ell}{\kappa}\frac{\sigma^{\frac{2}{2s+1}}}{K^{\frac{1}{2s+1}}T^{\frac{1}{2s+1}}}\right),$$

$$O\left(\frac{1}{\delta}\frac{1}{\gamma_x T}\frac{1}{B_0^{1-1/s}}\sigma\right) = O\left(\kappa^3\ell K^{\frac{s-1}{2s+1}}T^{\frac{s-1}{2s+1}}\frac{T^{\frac{2s}{2s+1}}\sigma^{\frac{3s}{(2s+1)(s-1)}}}{K^{\frac{1}{2s+1}}}\frac{1}{T}\frac{1}{K^{\frac{2s}{2s+1}\times\frac{s-1}{s}}T^{\frac{2s}{2s+1}\times\frac{s-1}{s}}\sigma^{\frac{s-1}{s}\times\frac{s(4s-1)}{(2s+1)(s-1)^2}}}\sigma\right)$$

$$= O\left(\frac{\kappa^3\ell\sigma^{\frac{2s}{2s+1}}}{K^{\frac{s}{2s+1}}T^{\frac{s}{2s+1}}}\right),$$

$$O\left(\frac{1}{\eta_x T}\right) = O\left(\frac{\kappa^5\ell}{1-\lambda}\frac{T^{\frac{2s}{2s+1}}\sigma^{\frac{4-s}{2(2s+1)(s-1)}}}{K^{\frac{1}{2s+1}}}\frac{1}{T}\right)$$

$$= O\left(\frac{\kappa^5\ell}{1-\lambda}\frac{\sigma^{\frac{4-s}{2(2s+1)(s-1)}}}{K^{\frac{1}{2s+1}}T^{\frac{1}{2s+1}}}\right). \tag{125}$$

Because $s \in (1,2]$ and $\bar{x}_0 = x_0, \bar{y}_0 = y_0, \bar{z}_0 = z_0$, it is easy to verify that the following terms marked by blue are high-order terms compared to $\frac{1}{T^{\frac{s-1}{2s+1}}}$:

$$\frac{1}{T}\sum_{t=0}^{T-1}\mathbb{E}[\|\nabla\Phi(\bar{x}_t)\|] \leq \frac{\mathbb{E}[\Phi(x_0) - \Phi(x^*)]}{\eta_x T} + 2\frac{1}{T}\sum_{t=0}^{T-1}\mathbb{E}[\|\nabla\Phi(\bar{x}_t) - \nabla\Phi_\delta(\bar{x}_t)\|]$$

$$+ \frac{1}{\delta}\frac{\mathbb{E}[h_\delta(x_0, y_0) - h_\delta^*(x_0)]}{\eta_x T} + \frac{1}{\delta}\frac{\mathbb{E}[g(x_0, z_0) - g^*(x_0)]}{\eta_x T}$$

$$+ O\left(\frac{1}{\delta}\frac{\gamma_x^{1-1/s}}{K^{1-1/s}}\sigma\right) + O\left(\frac{\kappa}{\delta}\frac{\gamma_y^{1-1/s}}{K^{1-1/s}}\sigma\right) + O\left(\frac{\kappa}{\delta}\frac{\gamma_z^{1-1/s}}{K^{1-1/s}}\sigma\right)$$

$$+ O\left(\frac{\eta_x}{(1-\lambda)\sqrt{\gamma_x K}}\frac{\kappa\ell}{\delta}\right) + O\left(\frac{\eta_x}{(1-\lambda)\sqrt{\gamma_y K}}\frac{\kappa^2\ell}{\delta}\right) + O\left(\frac{\eta_x}{(1-\lambda)\sqrt{\gamma_z K}}\frac{\kappa^2\ell}{\delta}\right)$$

$$+ O\left(\frac{1}{\delta}\frac{1}{\gamma_x T}\frac{1}{B_0^{1-1/s}}\sigma\right) + O\left(\frac{\kappa}{\delta}\frac{1}{\gamma_y T}\frac{1}{B_0^{1-1/s}}\sigma\right) + O\left(\frac{\kappa}{\delta}\frac{1}{\gamma_z T}\frac{1}{B_0^{1-1/s}}\sigma\right)$$

$$+ O\left(\frac{1}{\delta}\frac{\gamma_x\lambda\sqrt{K}}{(1-\lambda)^{3/2}}\sigma\right) + O\left(\frac{\kappa}{\delta}\frac{\gamma_y\lambda\sqrt{K}}{(1-\lambda)^{3/2}}\sigma\right) + O\left(\frac{\kappa}{\delta}\frac{\gamma_z\lambda\sqrt{K}}{(1-\lambda)^{3/2}}\sigma\right)$$

$$+ O\left(\frac{\eta_x\sqrt{\gamma_x}\lambda}{(1-\lambda)^{5/2}}\frac{\kappa\ell}{\delta}\right) + O\left(\frac{\eta_x\sqrt{\gamma_y}\lambda}{(1-\lambda)^{5/2}}\frac{\kappa^2\ell}{\delta}\right) + O\left(\frac{\eta_x\sqrt{\gamma_z}\lambda}{(1-\lambda)^{5/2}}\frac{\kappa^2\ell}{\delta}\right)$$

$$+ O\left(\frac{\kappa}{\delta}\frac{\lambda}{(1-\lambda)^{3/2}T}\frac{\sqrt{K}}{B_0^{1-1/s}}\sigma\right) + O\left(\eta_x\kappa^3\ell\right) + O\left(\eta_x\frac{\kappa^2\ell}{\delta}\right) + O\left(\eta_x\frac{\kappa^2\ell}{\delta}\frac{\lambda\sqrt{K}}{(1-\lambda)^{5/2}}\right)$$

$$+ O\left(\frac{\lambda}{(1-\lambda)T}\right)\frac{1}{\sqrt{K}}\sum_{k=1}^{K}\mathbb{E}[\|\nabla_1 f^{(k)}(x_0^{(k)}, y_0^{(k)})\|] + O\left(\frac{\lambda}{(1-\lambda)T}\frac{1}{\delta}\right)\frac{1}{\sqrt{K}}\sum_{k=1}^{K}\mathbb{E}[\|\nabla_1 g^{(k)}(x_0^{(k)}, y_0^{(k)})\|]$$

$$+ O\left(\frac{\lambda}{(1-\lambda)T}\frac{1}{\delta}\right)\frac{1}{\sqrt{K}}\sum_{k=1}^{K}\mathbb{E}[\|\nabla_1 g^{(k)}(x_0^{(k)}, z_0^{(k)})\|] + O\left(\frac{\lambda\kappa}{(1-\lambda)T}\right)\frac{1}{\sqrt{K}}\sum_{k=1}^{K}\mathbb{E}[\|\nabla_2 f^{(k)}(x_0^{(k)}, y_0^{(k)})\|]$$

$$+ O\left(\frac{\lambda\kappa}{(1-\lambda)T}\frac{1}{\delta}\right)\frac{1}{\sqrt{K}}\sum_{k=1}^{K}\mathbb{E}[\|\nabla_2 g^{(k)}(x_0^{(k)}, y_0^{(k)})\|] + O\left(\frac{\lambda\kappa}{(1-\lambda)T}\frac{1}{\delta}\right)\frac{1}{\sqrt{K}}\sum_{k=1}^{K}\mathbb{E}[\|\nabla_2 g^{(k)}(x_0^{(k)}, z_0^{(k)})\|].$$

$$\tag{126}$$

On the one hand, both $\frac{1}{\delta}\mathbb{E}[h_\delta(x_0, y_0) - h_\delta(x_0, y_\delta^*(x_0))]$ and $\frac{1}{\delta}\mathbb{E}[g(x_0, z_0) - g(x_0, y^*(x_0))]$ are affected by $\frac{1}{\delta}$, to avoid the degeneration of the convergence rate, we can provide good initial points $(x_0, y_0)$ and $(x_0, z_0)$ such that $\mathbb{E}[h_\delta(x_0, y_0) - h_\delta(x_0, y_\delta^*(x_0))] \leq \delta$ and $\mathbb{E}[g(x_0, z_0) - g(x_0, y^*(x_0))] \leq \delta$ can mitigate the adverse affect from $\frac{1}{\delta}$. Since both $h_\delta(x, y)$ and $g(x, z)$ satisfy the $\mu$-PL condition with respect to the second variable, we can use a gradient descent method to obtain such solutions, which has a linear convergence rate and therefore does not affect the other terms in Eq. (126). On the other hand, we have $\mathbb{E}[\|\nabla\Phi(\bar{x}_t) - \nabla\Phi_\delta(\bar{x}_t)\|] \leq O(\delta\ell\kappa^3) = O\left(\frac{1}{\kappa^3\ell}\frac{1}{K^{\frac{s-1}{2s+1}}T^{\frac{s-1}{2s+1}}}\ell\kappa^3\right) = O\left(\frac{1}{K^{\frac{s-1}{2s+1}}T^{\frac{s-1}{2s+1}}}\right)$. As a result, we can obtain

$$\frac{1}{T}\sum_{t=0}^{T-1}\mathbb{E}[\|\nabla\Phi(\bar{x}_t)\|] \leq O\left(\frac{\kappa^5\ell}{1-\lambda}\frac{\sigma^{\frac{4-s}{2(2s+1)(s-1)}}}{K^{\frac{1}{2s+1}}T^{\frac{1}{2s+1}}}\right) + O\left(\frac{1}{K^{\frac{s-1}{2s+1}}T^{\frac{s-1}{2s+1}}}\right)$$
$$+ O\left(\frac{\kappa^4\ell\sigma^{\frac{2s-2}{2s+1}}}{K^{\frac{s-1}{2s+1}}T^{\frac{s-1}{2s+1}}}\right) + O\left(\frac{\ell\sigma^{\frac{2}{2s+1}}}{K^{\frac{1}{2s+1}}T^{\frac{1}{2s+1}}}\right). \tag{127}$$

$\square$