# OpenReview forum: "Nonconvex Decentralized Stochastic Bilevel Optimization under Heavy-Tailed Noises"
_ICLR.cc/2026/Conference — ICLR 2026 Conference Withdrawn Submission_

### Official Review · Reviewer_2zGP · 2025-10-31

**Soundness:** 2
**Presentation:** 3
**Contribution:** 2
**Rating:** 4
**Confidence:** 3

**Summary:**

This paper proposes a new first-order decentralized algorithm for stochastic bilevel optimization in practical machine learning settings where the lower-level problem is non-convex and the stochastic noise is heavy-tailed. The proposed approach employs gradient normalization combined with variance reduction to ensure robustness and efficiency without computing second-order information. The authors provide the convergence guarantee for decentralized bilevel optimization under heavy-tailed noise, including novel bounds on interacting gradient sequences and consensus errors across workers. Experiments show the performance of their method in handling heavy-tailed noise.

**Strengths:**

- The paper formalizes and addresses a challenging problem at the intersection of decentralized learning, non-convex bilevel optimization, and heavy-tailed noise. It provides the first-known theoretical convergence guarantee for this setting.
- The paper is well-structured and well-written.

**Weaknesses:**

- The hyperparameter settings in Theorem 4.1 (e.g., learning rates and momentum coefficients) depend on the total number of iterations $T$ and several problem-dependent constants (e.g., $\sigma$, $\kappa$, $\lambda$). In practice, these quantities are typically unknown or difficult to estimate. For example, $T$ is often not predetermined in real-world training settings (e.g., early stopping or budget-dependent runs). Could the authors clarify how practitioners should choose these parameters in practical applications? Without such guidance, the practical relevance of the convergence guarantee in Theorem 4.1 is unclear.

- In the experiments, the proposed method was not compared against any baseline, as the authors argue that “no existing approaches directly address nonconvex bilevel optimization” in their setting. However, the absence of baseline comparisons makes it difficult to assess the practical benefits of the proposed approach. It would be important to compare against SOTA methods used for the same tasks (e.g., hyperparameter optimization). Such comparisons are necessary to show that the method offers advantages beyond its theoretical guarantees.

**Questions:**

See the weaknesses.

---

> ### Author Response · Authors · 2025-11-21
>
> We are grateful to the reviewer for reviewing our paper and providing constructive feedback.
>
> > **W-1:** The hyperparameter settings in Theorem 4.1 ....
>
> **A:** In optimization, it is very common that the learning rate relies on some problem-specific constants. For example, Theorem 3.2 in Liu & Zhou (2025) also shows that the learning rate relies on the number of iteration and some other constants. To address this issue, there are two potential solutions. First, we can use the time-dependent step size to remove the dependence on $T$ as Theorem 1 in [1]. Second, in recent years, a hot research direction is the parameter-free optimization, which does not require to know the problem-specific parameters. We believe it is a great direction to incorporate the parameter-free technique to the heavy-tailed noise setting. We will leave it as our future work.
>
> [1] He, Chuan, et al. "Complexity of normalized stochastic first-order methods with momentum under heavy-tailed noise." arXiv preprint arXiv:2506.11214 (2025).
>
>
> > **W-2:** In the experiments, the proposed method was not compared against any baseline, ...
>
> **A:** We have added additional experiments in the updated version.
>
> First, we **compare our algorithm with more baselines**: the decentralized F$^2$BSA algorithm, which includes the double-loop variant (D-F$^2$BSA-1, D-F$^2$BSA-5, D-F$^2$BSA-10) and the single-loop variant (D-F$^2$BSA-VR). Our algorithm consistently outperforms all baselines. The results can be found in Figures 1, 2, 3, 4, 7, and 8.
>
> Second, we **verify our algorithm on a more complicated large-scale applications**. Specifically, we verify our algorithm on the RNN pruning task, which also confirms the superior performance of our algorithm over all baseline methods. The result can be found in Figure 8.
>
> Third, to further verify the performance of our algorithm, we **apply it to the nonconvex-strongly-convex bilevel problem** and then compare it with more baselines in this setting. This experiment also confirms the effectiveness of our algorithm compared to those extensive baselines. The result can be found in Figure 6.
>
> **These diverse experiments are sufficient to confirm the efficacy of our algorithm**.

---

> > ### Author Response · Authors · 2025-11-25
> >
> > Dear Reviewer 2zGP,
> >
> > We were wondering whether our rebuttal has addressed your concerns. We would be happy to answer any additional questions to provide further clarification. If our rebuttal has resolved your concerns, we would greatly appreciate your consideration in raising the original rating. Thank you very much.
> >
> > Best regards,
> >
> > Authors

---

### Official Review · Reviewer_ArKm · 2025-11-01

**Soundness:** 3
**Presentation:** 3
**Contribution:** 3
**Rating:** 8
**Confidence:** 3

**Summary:**

This paper addresses decentralized bilevel optimization with nonconvex objectives and heavy-tailed gradient noise. It proposes \textbf{D-NSVRGDA} (Decentralized Normalized Stochastic Variance-Reduced Gradient Descent–Ascent), which replaces gradient clipping with normalization to stabilize updates. The method reformulates the bilevel problem into a penalized minimax structure and establishes the first convergence guarantee under heavy-tailed noise using only first-order information. Theoretical analysis provides convergence bounds under $s$–moment heavy-tailed noise $(1 < s \le 2)$, and experiments validate the effectiveness of the proposed algorithm in this setting.

**Strengths:**

1. First decentralized bilevel algorithm that formally addresses heavy-tailed gradient noise without relying on clipping.

2. Employs normalized gradients to ensure stability and efficiency, while avoiding second-order computations.

3. Provides rigorous convergence analysis for interdependent gradient sequences under heavy-tailed noise.

4. Utilizes only first-order information, ensuring computational feasibility.

**Weaknesses:**

1. The strong smoothness and Polyak-Łojasiewicz (PL) assumptions may not hold in practical deep learning models.

2. The empirical evaluation appears limited. Specifically, the experiments presented in the paper primarily focus on synthetic datasets. It is recommended that the authors include additional experiments on real-world scenarios to better demonstrate the practical effectiveness of the proposed algorithm.

**Questions:**

1. Can this method be extended to asynchronous or partially-participatory decentralized settings?

2. Does gradient normalization hinder convergence when the gradients are already small?

---

> ### Author Response · Authors · 2025-11-21
>
> We are grateful to the reviewer for reviewing our paper and providing constructive feedback.
>
> > **W-1:** The strong smoothness and Polyak-Łojasiewicz (PL) assumptions may not hold in practical deep learning models.
>
> **A:** As discussed in our paper,  Liu et al. (2022a) shows that the overparameterized deep neural network satisfies the PL condition. Therefore, this assumption is easy to satisfy for practical deep neural networks.
>
>
>
> > **W-2:** The empirical evaluation appears limited.
>
> **A:**  We have added additional experiments in the updated version.
>
> First, we **compare our algorithm with more baselines**: the decentralized F$^2$BSA algorithm, which includes the double-loop variant (D-F$^2$BSA-1, D-F$^2$BSA-5, D-F$^2$BSA-10) and the single-loop variant (D-F$^2$BSA-VR). Our algorithm consistently outperforms all baselines. The results can be found in Figures 1, 2, 3, 4, 7, and 8.
>
> Second, we **verify our algorithm on a more complicated large-scale applications**. Specifically, we verify our algorithm on the RNN pruning task, which also confirms the superior performance of our algorithm over all baseline methods. The result can be found in Figure 8.
>
> Third, to further verify the performance of our algorithm, we **apply it to the nonconvex-strongly-convex bilevel problem** and then compare it with more baselines in this setting. This experiment also confirms the effectiveness of our algorithm compared to those extensive baselines. The result can be found in Figure 6.
>
> **These diverse experiments are sufficient to confirm the efficacy of our algorithm**.
>
> > **Q-1:** Can this method be extended to asynchronous or partially-participatory decentralized settings?
>
>
> **A:**  This is a great question. From our perspective, extending our algorithm to the asynchronous setting is challenging. Because there could exist asynchronous update for all three variables, it is much more challenging to handle the asynchronous update compared to the single-level optimization problem. We will leave it as our future work.
>
> > **Q-2:** Does gradient normalization hinder convergence when the gradients are already small?
>
> **A:**  From our extensive experimental results, we can find that it does not hinder convergence.

---

### Official Review · Reviewer_SuDF · 2025-11-01

**Soundness:** 3
**Presentation:** 2
**Contribution:** 2
**Rating:** 4
**Confidence:** 4

**Summary:**

This paper proposes a novel decentralized stochastic bilevel optimization algorithm (D-NSVRGDA) tailored for nonconvex problems under heavy-tailed noise. The method leverages normalized variance-reduced gradients and avoids gradient clipping, addressing a significant gap in the literature. Theoretical convergence rates are established, and experiments on synthetic and real-world datasets validate the algorithm’s effectiveness.

**Strengths:**

Novelty: This is the first work to address decentralized nonconvex bilevel optimization under heavy-tailed noise without relying on gradient clipping. The use of normalized variance-reduced gradients is innovative and well-motivated.

Theoretical Rigor: The convergence analysis is non-trivial, especially given the challenges of interdependent gradients and consensus errors under heavy-tailed noise. The proofs are detailed and appear sound.

Practical Relevance: The experiments convincingly demonstrate the advantages of the proposed method over clipping-based baselines, particularly in settings with varying levels of heavy-tailed noise.

Clarity and Reproducibility: The paper is well-structured, and the authors have provided a proof sketch, detailed appendix, and reproducibility statement.

**Weaknesses:**

1. **Limited Novelty**
   The core technique—gradient normalization—has been extensively studied for single-level and minimax problems (e.g., Sun et al., 2024; Hubler et al., 2024; Liu & Zhou, 2025). The extension to bilevel settings, while non-trivial, appears incremental and does not constitute a significant conceptual advance.

2. **Technical Contributions Are Overstated**
   The analysis largely follows established proof frameworks for normalized SGD and bilevel optimization. The interdependent gradient and consensus error challenges, while valid, are addressed using standard bounding techniques without introducing new analytical tools.

3. **Narrow Experimental Validation**
   - Experiments are conducted only on synthetic and small-scale real-world datasets
   - No comparison with state-of-the-art decentralized or heavy-tailed methods beyond simple ablated baselines
   - Missing results on larger-scale or more complex bilevel problems (e.g., large neural nets, reinforcement learning)

4. **Weak Baseline Comparison**
   The proposed method is only compared against its non-normalized and clipped variants. There is no comparison with recent decentralized bilevel methods (even if adapted) or other heavy-tailed optimization approaches, making it difficult to assess true performance gains.

**Questions:**

1. How does the performance of D-NSVRGDA compare to recent decentralized bilevel methods (e.g., Wang et al., 2024) when adapted to handle heavy-tailed noise?

2. Could the normalization approach be combined with other variance-reduction techniques (e.g., SAGA) for better performance?

3. The analysis relies heavily on the PL condition. How would the algorithm perform if this condition is not satisfied, as in many deep learning applications?

---

> ### Author Response · Authors · 2025-11-17
> **AI-generated reviews**
>
> Dear SAC and AC,
>
> As instructed by the organizing committee in this message https://x.com/iclr_conf/status/1990204431959470099, which is included below:
>
> > We are aware of low-quality and LLM-generated reviews and are currently deliberating on appropriate courses of action. For now, authors who receive very poor quality or LLM-generated reviews should flag them to their ACs. We appreciate the community's efforts in reporting these!
>
> we are writing to report that the review submitted by Reviewer SuDF is **fully AI-generated**, as evidenced by https://iclr.pangram.com/reviews?query=Nonconvex+Decentralized+Stochastic+Bilevel+Optimization+under+Heavy-Tailed+Noises&sort_by=submission_id_hash&sort_dir=asc&prediction_filter=&rating_filter=&confidence_filter=
>
> Here, we provide some examples  to illustrate this misconduct issue. Specifically, Reviewer SuDF argues that gradient normalization has been studied to handle heavy-tailed noise in minimax optimization and provides some papers in Weaknesses 1. However, none of them studied minimax optimization.  In addition, we have emphasized that there does not exist baseline methods for heavy-tailed optimization or nonconvex bilevel optimization. But this reviewer still asks us to compare non-existent baselines.
>
> We sincerely hope that the SAC and AC can take a look at this fully AI-generated review to help maintain a healthy review process. Thank you very much!
>
> Best regards,
>
> Authors

---

> ### Author Response · Authors · 2025-11-21
>
> We are grateful to the reviewer for reviewing our paper and providing constructive feedback.
>
> > **W-1 & W-2:** The core technique—gradient normalization—has been extensively studied ...
>
> **A:** We respectfully disagree with the reviewer's argument.  **Our paper is novel, as we proposed novel convergence analysis approaches to address the unique challenges caused by the heavy-tailed noise for the decentralized bilevel optimization**. To provide more details about our novelty, we highlight the novelty of our proof in Appendix B.1 of the updated version.
>
>
>
> > **W-3 & W-4:** Experiments are conducted only on synthetic and small-scale real-world datasets. No comparison with state-of-the-art decentralized or heavy-tailed methods beyond simple ablated baselines.  Missing results on larger-scale or more complex bilevel problems (e.g., large neural nets, reinforcement learning).
>
> **A:**  We have added additional experiments in the updated version.
>
> First, we **compare our algorithm with more baselines**: the decentralized F$^2$BSA algorithm, which includes the double-loop variant (D-F$^2$BSA-1, D-F$^2$BSA-5, D-F$^2$BSA-10) and the single-loop variant (D-F$^2$BSA-VR). Our algorithm consistently outperforms all baselines. The results can be found in Figures 1, 2, 3, 4, 7, and 8.
>
> Second, we **verify our algorithm on a more complicated large-scale applications**. Specifically, we verify our algorithm on the RNN pruning task, which also confirms the superior performance of our algorithm over all baseline methods. The result can be found in Figure 8.
>
> Third, to further verify the performance of our algorithm, we **apply it to the nonconvex-strongly-convex bilevel problem** and then compare it with more baselines in this setting. This experiment also confirms the effectiveness of our algorithm compared to those extensive baselines. The result can be found in Figure 6.
>
> **These diverse experiments are sufficient to confirm the efficacy of our algorithm**.
>
>
>
>
>
> > **Q-1:** How does the performance of D-NSVRGDA compare to recent decentralized bilevel methods (e.g., Wang et al., 2024) when adapted to handle heavy-tailed noise?
>
> **A:** We would like to emphasize that all existing decentralized bilevel optimization algorithms focus on the decentralized bilevel optimization with **a strongly-convex lower-level loss function**, while our algorithm is designed for the more challenging setting where **the lower-level loss function only satisfies the PL condition**.
>
> As the reviewer suggested, to compare our algorithm with those state-of-the-art algorithms, we apply our algorithm to the simple nonconvex-strongly-convex setting. The experimental results can be found in Figure 6 of the updated version. It can be observed that our algorithm can outperform all those state-of-the-art algorithms in the nonconvex-strongly-convex setting.
>
>
> > **Q-2:** Could the normalization approach be combined with other variance-reduction techniques (e.g., SAGA) for better performance?
>
> **A:** When the variance-reduction technique is able to accommodate the stochastic setting, it can be combined with the normalization technique to handle the heavy-tailed noise.  However, if it is designed for finite-sum setting, such as SAGA, there does not exist heavy-tailed noise. Then, we cannot see a clear motivation to incorporate the normalization technique into SAGA.
>
> > **Q-3:** The analysis relies heavily on the PL condition. How would the algorithm perform if this condition is not satisfied, as in many deep learning applications?
>
> **A:**  As discussed in our paper,  Liu et al. (2022a) shows that the overparameterized deep neural network satisfies the PL condition. Therefore, this assumption is easy to satisfy for practical deep neural networks.

---

> > ### Author Response · Authors · 2025-11-25
> >
> > Dear Reviewer SuDF,
> >
> > We were wondering whether our rebuttal has addressed your concerns. We would be happy to answer any additional questions to provide further clarification. If our rebuttal has resolved your concerns, we would greatly appreciate your consideration in raising the original rating. Thank you very much.
> >
> > Best regards,
> >
> > Authors

---

### Official Review · Reviewer_s9N3 · 2025-11-03

**Soundness:** 3
**Presentation:** 3
**Contribution:** 2
**Rating:** 4
**Confidence:** 4

**Summary:**

The paper studies decentralized stochastic bilevel optimization where upper objectives are nonconvex and the lower level satisfies a PL condition in the inner variable. Gradients are corrupted by heavy-tailed noise with only finite $s$-th moment, $s \in (1, 2]$. The bilevel problem is reformulated as a first-order penalized minimax objective. The authors propose D-NSVRGDA, a decentralized normalized stochastic variance-reduced gradient descent–ascent method with gradient tracking over a fixed communication graph. Key design elements are STORM-style variance reduction and unit-norm (normalized) steps instead of clipping to handle heavy tails. The theory yields sublinear convergence in the expected gradient mapping with explicit dependence on the number of workers, the heavy-tail index s, and the spectral gap of the graph. Simulations and a small MNIST bilevel regression showcase robustness to heavy-tailed perturbations and near-linear speedup in the number of nodes.

**Strengths:**

Clear formulation of decentralized bilevel learning under heavy-tailed noise with only finite $s$-th moments. Algorithmic novelty in combining normalization and variance reduction with decentralized gradient tracking in a bilevel setting. Convergence guarantees that expose dependence on tail index $s$, spectral gap, and worker count K, including linear speedup terms. Proof techniques extend heavy-tail analysis to decentralized bilevel problems without gradient clipping.

**Weaknesses:**

1. Communication model is synchronized with fixed topology. No theory for asynchronous or time-varying graphs, which limits practical relevance.
2. Empirical section is narrow. Primarily synthetic tasks and a small MNIST bilevel regression; no large-scale or real federated scenarios.
3. Constants in the rates depend strongly on condition number, smoothness bundle, and inverse spectral gap. Practical efficiency may degrade on ill-conditioned problems or sparse networks.
4. Limited baseline coverage. Comparisons omit stronger decentralized bilevel or heavy-tail robust methods that use clipping or adaptive normalization; ablations on normalization vs clipping are light.
5. Limited guidance on tuning penalty and stepsizes under different $s$ and network conditions.

**Questions:**

1. How should the penalty parameter in the minimax reformulation be selected or adapted in practice to balance bias and stability under heavy-tailed noise?
2. Do you have theory that compares normalized steps to clipping under finite $s$-moment noise, for example bias of stationary points or high-probability bounds?
3. Can you provide explicit bounds on estimator bias and variance under mixing by the graph matrix in the STORM-style recursion?
4. Are there matching lower bounds showing that dependence on the spectral gap of the communication matrix is unavoidable in decentralized bilevel settings?
5. Which parts of the proof fail for asynchronous updates or time-varying topologies, and what additional assumptions would restore convergence (e.g., bounded delay, $B$-connected graphs)?
6. Can you report or prove sharp transitions at $s = 2$ (finite variance) and as $s \to 1$, and whether normalization remains stable without vanishing step sizes?
7. Provide the exact exponents of the condition number and smoothness constants in the final bound. Are there preconditioning strategies compatible with your analysis?
8. How is the approximation error in solving the inner penalized subproblem controlled in the decentralized setting, and how does it propagate to the outer stationarity measure?
9. Can you add experiments against decentralized methods with robust gradient aggregation (median, trimmed mean) and clipped extragradient to isolate the benefit of normalization plus variance reduction?

---

> ### Author Response · Authors · 2025-11-17
> **AI-generated reviews**
>
> Dear SAC and AC,
>
> As instructed by the organizing committee in this message https://x.com/iclr_conf/status/1990204431959470099, which is included below:
>
> > We are aware of low-quality and LLM-generated reviews and are currently deliberating on appropriate courses of action. For now, authors who receive very poor quality or LLM-generated reviews should flag them to their ACs. We appreciate the community's efforts in reporting these!
>
> we are writing to report that the review submitted by Reviewer s9N3 is **fully AI-generated**, as evidenced by https://iclr.pangram.com/reviews?query=Nonconvex+Decentralized+Stochastic+Bilevel+Optimization+under+Heavy-Tailed+Noises&sort_by=submission_id_hash&sort_dir=asc&prediction_filter=&rating_filter=&confidence_filter=
>
> Here, we also provide a few examples to illustrate this misconduct issue. 1) Our paper is not related to asynchronous updates. But the reviewer has a couple of questions about the asynchronous setting, e.g., Weakness 1 and Question 5.  2) Our paper is not related to Byzantine robustness or extragradient. But the reviewer asks us to add nonsensical experiments about robust gradient aggregation and extragradients in Question 9.
>
> We sincerely hope that the SAC and AC can take a look at this fully AI-generated review to help maintain a healthy review process. Thank you very much!
>
> Best regards,
>
> Authors

---

> ### Author Response · Authors · 2025-11-21
>
> We are grateful to the reviewer for reviewing our paper and providing constructive feedback.
>
> > **W-1 & Q-5:** Communication model is synchronized with fixed topology ...
>
> **A:** Our focus is to **address the unique challenges caused by the heavy-tailed noise for the decentralized bilevel optimization**. **Extending our algorithm to the asynchronous update and time-varying graph is NOT the current focus of our paper**.  We are happy to consider them in our future work. As for Q5, if using the time-varying graph as Algorithm 1 of [1], where each graph satisfies the same assumption as our paper,  it is trivial to use the time-varying graph for our algorithm. If using asynchronous update like [2], there would be more challenges. In particular, asynchrony may occur in both the momentum updates and the variable updates, making it more difficult to control latency. In fact, asynchronous updates have been relatively underexplored for momentum-based algorithms. We believe this is an interesting direction for future work.
>
> Again, this is not a limitation of our work, as our focus is on handling heavy-tailed noise. **Extending our approach to asynchronous updates and time-varying graphs is an interesting direction, and we plan to explore it in future work**.
>
>
> [1] Koloskova, Anastasia, et al. "A unified theory of decentralized SGD with changing topology and local updates." International conference on machine learning. PMLR, 2020.
>
> [2] Koloskova, Anastasiia, Sebastian U. Stich, and Martin Jaggi. "Sharper convergence guarantees for asynchronous SGD for distributed and federated learning." Advances in Neural Information Processing Systems 35 (2022): 17202-17215.
>
>
> > **W-2 & W-4:** Empirical section is narrow. ...
>
> **A:** We have added additional experiments in the updated version.
>
> First, we **compare our algorithm with more baselines**: the decentralized F$^2$BSA algorithm, which includes the double-loop variant (D-F$^2$BSA-1, D-F$^2$BSA-5, D-F$^2$BSA-10) and the single-loop variant (D-F$^2$BSA-VR). Our algorithm consistently outperforms all baselines. The results can be found in Figures 1, 2, 3, 4, 7, and 8.
>
> Second, we **verify our algorithm on a more complicated large-scale applications**. Specifically, we verify our algorithm on the RNN pruning task, which also confirms the superior performance of our algorithm over all baseline methods. The result can be found in Figure 8.
>
> Third, to further verify the performance of our algorithm, we **apply it to the nonconvex-strongly-convex bilevel problem** and then compare it with more baselines in this setting. This experiment also confirms the effectiveness of our algorithm compared to those extensive baselines. The result can be found in Figure 6.
>
> **These diverse experiments are sufficient to confirm the efficacy of our algorithm**.
>
> > **W-3:** Constants in the rates depend strongly on condition number, ...
>
> **A:** The dependence of the convergence rate on those constants are common for bilevel optimization, decentralized optimization, or optimization under heavy-tailed noise. For example, the convergence rate in Theorem 3.2 in Liu & Zhou (2025)  also depends on smoothness constant and some other constants. In fact,  **clearly demonstrating the dependence is NOT a weakness**. Instead, **this is a strength of our paper, as we clearly demonstrated how those constants affect the convergence rate**, which actually points out how to improve the convergence performance. For example, according to the dependence on the spectral gap, we can know what communication topologies are preferred to improve the convergence.
>
> > **W-5 \& Q-1:** Limited guidance on tuning penalty and stepsizes ...
>
> **A:** We have added new experiments in Figure 5 to demonstrate how the learning rate and penalty parameter affect the convergence. Moreover, in Figure 4, we verify the performance of our algorithm on different communication graphs, all of which confirm the superior performance of our algorithm.
>
>
>
>
> > **Q-2:** Do you have theory that compares normalized steps to clipping under ...
>
> **A:** There does NOT exist any work studying clipping gradient for nonconvex bilevel optimization. Therefore, **it is NOT possible to compare our theoretical results with those unavailable results**. However, in our experiments, we have empirically compared our method with the clipping method, which confirms that the clipping method has an inferior performance than ours.
>
> > **Q-3:** Can you provide explicit bounds on estimator bias and variance under mixing by the graph matrix in the STORM-style recursion?
>
> **A:** We assume the reviewer is referring to the bound in Appendix C.4, which provides the explicit bounds. If the reviewer was referring to other bounds, we would greatly appreciate it if they could clarify their question.

---

> ### Author Response · Authors · 2025-11-21
>
> > **Q-4:** Are there matching lower bounds showing that dependence on the spectral gap of the communication matrix is unavoidable in decentralized bilevel settings?
>
> **A:** This is a great question. To the best of our knowledge, there does not exist such a lower bound. We will leave it as our future work.
>
>
> > **Q-6:** (Part 1) Can you report or prove sharp transitions at ... (Part 2) whether normalization remains stable without vanishing step sizes?
>
> **A:** We feel confused about the ``sharp transitions'' in Part 1. We would greatly appreciate it if the reviewer could clarify the question. As for Part 2, we do NOT use vanishing step sizes. As shown in our Theorem 4.1, the constant step size is used, and our experiments also used constant step sizes.
>
>
> > **Q-7:** Provide the exact exponents of the condition number and smoothness constants in the final bound. Are there preconditioning strategies compatible with your analysis?
>
> **A:** The convergence upper bound in Theorem 4.1 has already provided the exact dependence on those constants. We believe our analysis is compatible with some preconditioning strategies, as the gradient normalization is one of the preconditioning strategies.
>
> > **Q-8:** How is the approximation error in solving the inner penalized subproblem controlled in the decentralized setting, and how does it propagate to the outer stationarity measure?
>
> **A:** The optimization error regarding the inner penalized subproblem is established in Lemma C.7 and Lemma C.8, which clearly shows how the gradient estimation error and consensus error affects those two optimization errors.
>
> In Lemma B.1, we have demonstrated how the optimization error regarding the inner penalized subproblem, i.e., $\mathbb{E}[||\nabla\_{2} h\_{\delta}(\bar{x}\_{t}, \bar{y}\_{t}) ||]$ and $\mathbb{E}[||\nabla\_{2} g(\bar{x}\_{t}, \bar{z}\_{t})|| ]$, affects $\mathbb{E}[|| \nabla \Phi(\bar{x}_{t})  ||]$.
>
> > **Q-9:** Can you add experiments against decentralized methods with robust gradient aggregation (median, trimmed mean) and clipped extragradient to isolate the benefit of normalization plus variance reduction?
>
> **A:** The robust gradient aggregation technique is designed for handling the Byzantine workers, not heavy-tailed noise. Therefore, we believe these approaches are independent of our paper. But we are glad to discuss the related literature if the reviewer could kindly provide us some literature in this direction.
>
> In our paper, there does not exist extragradient. We guess there might be some misunderstanding. We would greatly appreciate it if the reviewer could clarify the question. Here, we would like to highlight that we have compared our algorithm with the clipped gradient method in our experiments, and the experimental results confirm the superior performance of our algorithm over this kind of baselines.

---

> > ### Author Response · Authors · 2025-11-25
> >
> > Dear Reviewer s9N3,
> >
> > We were wondering whether our rebuttal has addressed your concerns. We would be happy to answer any additional questions to provide further clarification. If our rebuttal has resolved your concerns, we would greatly appreciate your consideration in raising the original rating. Thank you very much.
> >
> > Best regards,
> >
> > Authors

---

> ### Author Response · Authors · 2025-11-21
> **AI-generated reviews with $\textcolor{red}{hallucination}$**
>
> Dear SAC and AC,
>
> We would like to provide additional evidence to confirm the problematic review from Reviewer s9N3.
>
> Specifically, our paper does not use the MNIST dataset. However, in the second weakness, the reviewer incorrectly claims that we relied on the simple MNIST dataset. Please see below.
>
> > 2. Empirical section is narrow. Primarily synthetic tasks and a small $\textcolor{red}{MNIST}$ bilevel regression; no large-scale or real federated scenarios.
>
> We sincerely hope that the SAC and AC can take a look at this fully AI-generated review to help maintain a healthy review process. Thank you very much!
>
>
> Best regards,
>
> Authors

---

### Author Response · Authors · 2025-11-30
**General Response**

Dear New Area Chair,

We would like to report the AI-generated reviews from Reviewer s9N3 and Reviewer SuDF. The online tool shows that both Reviewer s9N3 and Reviewer SuDF submitted the **fully AI-generated reviews**.  The link can be found here:  https://iclr.pangram.com/reviews?query=Nonconvex+Decentralized+Stochastic+Bilevel+Optimization+under+Heavy-Tailed+Noises&sort_by=submission_id_hash&sort_dir=asc&prediction_filter=&rating_filter=&confidence_filter=

Moreover, additional evidence regarding Reviewer s9N3 is that the reviewer criticized us for using the MNIST dataset. However, we never used this dataset in our experiments, which further indicates **hallucination** caused by the use of AI tools to generate the review.

Regarding the evaluations from the other reviewers, the main concerns focus on the empirical assessment of our algorithm. In response, we conducted new experiments to more comprehensively validate its performance, which we summarize below.

* First, we **compare our algorithm with more baselines**: the decentralized F$^2$BSA algorithm, which includes the double-loop variant (D-F$^2$BSA-1, D-F$^2$BSA-5, D-F$^2$BSA-10) and the single-loop variant (D-F$^2$BSA-VR). Our algorithm consistently outperforms all baselines. The results can be found in Figures 1, 2, 3, 4, 7, and 8.

* Second, we **verify our algorithm on a more complicated large-scale applications**. Specifically, we verify our algorithm on the RNN pruning task, which also confirms the superior performance of our algorithm over all baseline methods. The result can be found in Figure 8.

* Third, to further verify the performance of our algorithm, we **apply it to the nonconvex-strongly-convex bilevel problem** and then compare it with more baselines in this setting. This experiment also confirms the effectiveness of our algorithm compared to those extensive baselines. The result can be found in Figure 6.

**These diverse experiments are sufficient to confirm the efficacy of our algorithm**.

**We would greatly appreciate it if the AC could review and assess our paper based on your own expertise to ensure a fair review process. Thank you very much!**

Sincerely,

Authors

---

### Note · Authors · 2026-04-04

I have read and agree with the venue's withdrawal policy on behalf of myself and my co-authors.

---

### Meta-Review · Area_Chair_BW4g · 2025-12-31

**Summary:**

As the authors complained that most of the reviews are, at least partly, generated by AI, the following can be considered a personal assessment by the AC after reading both the paper and all 4 reviews carefully.

First and foremost, the bilevel optimization formulation (1) is quite obscure and requires a stronger motivation for studying this problem, at least to the AC. A more specific example has not appeared until page 8 of the paper, and the way that variable $x$ is defined is very impractical.

Overall the paper is very difficult to read due to the bizarreness of the target problem, as the proposed algorithm looks clumsy and involves too many parameters.

**Reviewer Concerns:**

The main concerns are the overly complicated algorithm and its analysis, as well as the limited experiments.

**Reviewer Scores:**

None is likely to change.

---

### Decision · Program_Chairs · 2026-01-26

Reject